# Understanding the theoretical properties of projected Bellman equation, linear Q-learning, and approximate value iteration

## Abstract

In this paper, we study the theoretical properties of the projected Bellman equation (PBE) and two algorithms to solve this equation: linear Q-learning and approximate value iteration (AVI). We consider two sufficient conditions for the existence of a solution to PBE : strictly negatively row dominating diagonal (SNRDD) assumption and a condition motivated by the convergence of AVI. The SNRDD assumption also ensures the convergence of linear Q-learning, and its relationship with the convergence of AVI is examined. Lastly, several interesting observations on the solution of PBE are provided when using $\epsilon$-greedy policy.

## 1 Introduction

Reinforcement learning (RL) has achieved significant success, exemplified by the deep Q-network (DQN) (Mnih et al., 2015). This success can be largely attributed to two algorithms: Q-learning (Watkins and Dayan, 1992) and the approximate value iteration (AVI) (Bertsekas, 2011). Understanding the behavior of these algorithms has been a central focus of extensive research.

Q-learning, initially developed by Watkins and Dayan (1992) in a tabular setup where $Q$-values are stored for every state-action pair, has since been the subject of considerable investigation. Both asymptotic and non-asymptotic analysis of the algorithm have been thoroughly explored in works such as (Szepesvári, 1997; Borkar and Meyn, 2000; Even-Dar and Mansour, 2003; Lee and He, 2020b; Chen et al., 2022; Li et al., 2024; Lee, 2024), to list a few.

Moving beyond the tabular setup, function approximation is commonly used to address the problem of large state-action spaces in practical scenarios. Specifically, we focus on the simplest form of approximation: the linear function approximation scheme. However, introducing function approximation brings several challenges. In the case of Q-learning with linear function approximation—referred to as linear Q-learning—two major issues arise: 1) the existence of a solution to the projected Bellman equation (PBE) that the algorithm aims to solve, and 2) the stability of the algorithm. While recent works have explored these challenges (Melo et al., 2008; Meyn, 2024), there remains significant opportunity for further advancing our understanding in this area.

Meanwhile, value iteration is one of the simplest algorithms in RL when the model is known. By incorporating linear function approximation into the value iteration framework, the approximate value iteration (AVI) scheme has been widely used Munos (2007); Mann and Mannor (2014). AVI also seeks to solve the PBE as linear Q-learning does. Nonetheless, it is not well understood, when the AVI algorithm converges while linear Q-learning does not, and vice versa.

Overall, the theoretical understanding of PBE and its related algorithms, specifically linear Q-learning and AVI, are not well-understood. This motivates our study, and the purpose of this paper is to extend our knowledge on these subjects. The main contributions are outlined in the following:

1. A sufficient condition for existence and uniqueness of a solution to PBE:

   - We provide a thorough investigation of the existence and uniqueness of a solution to PBE under the assumption of a matrix having strictly negatively row dominating diagonals (SNRDD assumption), which is new in the literature. This assumption includes a wide class of settings:

tabular and linear function approximation (with regularization). Moreover, our analysis considers various behavior and target policy scenarios including continuous or Lispchitz policies.

- A sufficient condition, derived from the AVI framework, is provided. We then explore its relationship to the SNRDD assumption, demonstrating that while the two are generally different, they can coincide under specific additional conditions.

2. We provide a new convergence proofs for a family of Q-learning algorithms and AVI algorithm, respectively. Furthermore, we provide examples where AVI converges while linear Q-learning does not, and vice-versa. This provides novel insights on the relationship of convergence behavior of linear Q-learning and AVI.

- The proof of Q-learning relies on ODE arguments based on contraction theory (Lohmiller and Slotine, 1998) and the SNRDD assumption. This covers asynchronous tabular Q-learning, linear Q-learning with SNRDD assumption, and regularized Q-learning (Lim and Lee, 2024). It provides a novel unified understanding in proving convergence of both linear and tabular Q-learning using a fixed behavior policy. Regarding regularized Q-learning, the existing assumptions on positiveness and orthogonality of feature matrix are relaxed. Furthermore, we identify the one-sided Lipschitz condition of linear Q-learning and show a condition on regularization coefficient $\eta$ that does not depend on the knowledge of model parameters.

- We provide an example showing that, even though the SNRDD assumption ensures the convergence of Q-learning and the existence of a solution to the PBE, the resulting solution may still lead to a sub-optimal policy.

- The convergence of AVI follows from the condition that guarantees existence and uniqueness of a solution to PBE.

3. Lastly, we provide two examples explaining the theoretical properties of solutions to PBE when $\epsilon$-greedy policy is used, which is not covered by previous analysis due to its discontinuity. The first example shows that depending on the value of $\epsilon$, there is a chance of non-existence or multiplicity of the solution even though SNRDD condition is met. The second example shows a pathological phenomenon using $\epsilon$-greedy policy that increasing the parameter $\epsilon$ can ensure a solution to PBE that yields an optimal policy, to which Q-learning cannot converge. These examples show the hardness of analysis when using the $\epsilon$-greedy behavior policy.

**Related Works:**

Linear function approximation has been a useful tool to provide insight on the behavior of RL algorithms. Parr et al. (2008) studies learning a feature matrix in a model-based manner, i.e., requires a $\mathbb{R}^{|\mathcal{S}| \times |\mathcal{S}|}$ space memory. The main focus of Parr et al. (2008) is on the analysis under the policy evaluation scheme rather than considering policy improvement setting. In contrast, we focus on model-free learning setting and policy improvement scheme. Baird et al. (1995) analyzed residual algorithms that rely on two independent samples per iteration, in contrast to our algorithms, which do not require such a sampling structure. Tsitsiklis and Van Roy (1996) considered function approximation under policy evaluation scheme and highlighted its divergence issue whereas we consider policy control scheme where the target policy is iteratively updated.

Melo et al. (2008); Chen et al. (2022) studied the convergence of linear Q-learning with additional assumptions that might not be satisfied in the tabular setting. Meyn (2024); Liu et al. (2025) considered using a version of $\epsilon$-softmax behavior policy, the so-called tamed-Gibbs policy, and established results that there exists a solution of PBE, and the learning parameters of Q-learning remain bounded. Nonetheless, the tamed-Gibbs policy requires several specific design choices. In contrast, we consider a different scenario and proof approach: existence of the solution is explored for continuous or lipschitz policy under the assumption of SNRDD. In proving the convergence of Q-learning, we consider an arbitrary fixed behavior policy, which is idealistic but different scenario, and this naturally extends the proof idea of Q-learning in the tabular setup. The proof relies on contraction theory (Lohmiller and Slotine, 1998), and its connection offers new insights.

Lim and Lee (2024) studied Q-learning with an additional term that serves a similar role to $l_2$-regularization, referred to as regularized Q-learning. Under additional assumptions on the feature matrix, this ensures convergence to a unique point. Zhang et al. (2021) studied Q-learning using target network, projection and regularization. We show that target network, projection or any additional assumptions on feature matrix are not required to prove the convergence of regularized Q-learning.

Several studies have proposed variations of linear Q-learning (Chen et al., 2023; Maei et al., 2010; Devraj and Meyn, 2017; Carvalho et al., 2020) which are summarized in the Appendix Section 15. Although these methods ensure boundedness or convergence, the exact points to which the algorithm converges remain not well understood.

The AVI scheme has been widely studied to tackle the challenges posed by large state-action spaces (Bertsekas, 2016). Recent research has provided insights into the convergence properties of AVI, highlighting its close connection to algorithms that employ target network updates—a methodology inspired by the success of DQN. Lee and He (2020a) explored Q-learning in a tabular setting, while Asadi et al. (2023); Fellows et al. (2023); Che et al. (2024) investigated temporal difference (TD) learning with target network updates, demonstrating the crucial connection with AVI.

A few works tried to understand AVI scheme and TD-learning in a unified perspective. Guo and Hu (2022) proposed a convex program test approach for value iteration and TD-learning but requires different test for each algorithm. Wu et al. (2025) provided an understanding of TD-learning and AVI from the matrix splitting technique (Berman and Plemmons, 1994). In contrast, our work focuses on Q-learning, which presents unique challenge due to switching of the policies and non-linearity of the max-operator, making the standard TD-learning analysis techniques insufficient.

Pathological behaviors regarding the solution of PBE, e.g, the non-existence or multiplicity of solutions, which can lead to suboptimal policies has been well-known in the literature (De Farias and Van Roy, 2000; Bertsekas, 2011; Young and Sutton, 2020). This becomes more complex when we use $\epsilon$-greedy policy.[1] Lu et al. (2018) provided an example that for a certain regime of $\epsilon$, Q-learning can yield a sub-optimal policy compared to possible ones that can be represented by the linear feature while the optimal policy is not realizable. Covering a different scenario, we provide an example that the number of solutions depends on the choice of $\epsilon$, and depending on $\epsilon$, there is a solution of PBE induces optimal policy but to which Q-learning cannot converge.

## 2 PRELIMINARIES

### 2.1 MARKOV DECISION PROCESS (MDP)

MDP consists of five tuples $(\mathcal{S}, \mathcal{A}, \gamma, \mathcal{P}, r)$. $\mathcal{S} := [|\mathcal{S}|]$ and $\mathcal{A} := [|\mathcal{A}|]$, where $[n] := \{1, 2, \ldots, n\}$ for $n \in \mathbb{N}$, are finite state and action spaces, respectively. $\gamma \in (0, 1)$ is the discount factor. $\mathcal{P} : \mathcal{S} \times \mathcal{A} \to \Delta^{\mathcal{S}}$ is the Markov kernel where $\Delta^{\mathcal{S}}$ denotes a probability distribution over the set $\mathcal{S}$. $r : \mathcal{S} \times \mathcal{A} \times \mathcal{S} \to \mathbb{R}$ is the reward function, which we assume to be bounded. An agent at state $s \in \mathcal{S}$ selects an action $a \sim \pi(\cdot \mid s)$ following a policy $\pi : \mathcal{S} \to \Delta^{\mathcal{A}}$. Then, transition occurs to next state $s' \sim \mathcal{P}(\cdot \mid s, a)$ and the agent receives reward $r(s, a, s')$. The $Q$-function induced by policy $\pi$ is defined as $Q^{\pi}(s, a) = \mathbb{E}\left[\sum_{k=0}^{\infty} \gamma^k r(S_k, A_k, S_{k+1}) \mid (S_0, A_0) = (s, a), \pi\right]$, where $\{(S_k, A_k) \in \mathcal{S} \times \mathcal{A}\}_{k=0}^{\infty}$ are a sequence of random variables following the policy $\pi$. The goal is to find an optimal policy $\pi^*$ such that $\pi^* = \arg\max_{\pi \in \Omega} \mathbb{E}\left[\sum_{k=0}^{\infty} \gamma^k r(S_k, A_k, S_{k+1}) \mid \pi\right]$ where $\Omega$ is the set of all deterministic policies. We denote $Q^*$ as the optimal $Q$-function, which is the $Q$-function induced by the optimal policy $\pi^*$. The optimal $Q$-function satisfies the Bellman optimality equation : $\boldsymbol{R} + \gamma \boldsymbol{P} \boldsymbol{Q}^* = \boldsymbol{Q}^*$ where $\boldsymbol{R} \in \mathbb{R}^{|\mathcal{S}||\mathcal{A}|}$ is a vector such that $[\boldsymbol{R}]_{(s-1)|\mathcal{A}|+a} = \mathbb{E}\left[r(s, a, s') \mid (s, a)\right]$, $\boldsymbol{P} \in \mathbb{R}^{|\mathcal{S}||\mathcal{A}| \times |\mathcal{S}|}$ is transition matrix such that $[\boldsymbol{P}]_{(s-1)|\mathcal{A}|+a, s'} = \mathcal{P}(s' \mid s, a)$, and $\boldsymbol{Q}^* \in \mathbb{R}^{|\mathcal{S}||\mathcal{A}|}$ is a vector such that $[\boldsymbol{Q}^*]_{(s-1)|\mathcal{A}|+a} = Q^*(s, a)$, where for $\boldsymbol{v} \in \mathbb{R}^n$ and $i \in [n]$, $[\boldsymbol{v}]_i$ denotes $i$-th element of $\boldsymbol{v}$, and $[\boldsymbol{A}]_{i,j}$ for $\boldsymbol{A} \in \mathbb{R}^{n \times m}$ denotes the element in the $i$-th row and $j$-th column of $\boldsymbol{A}$.

### 2.2 LINEAR FUNCTION APPROXIMATION OF $Q$-FUNCTION

Consider a set of features $\{\boldsymbol{\phi}(s, a) \in \mathbb{R}^p\}_{(s,a) \in \mathcal{S} \times \mathcal{A}}$, where $p \in \mathbb{N}$ is the feature dimension. We approximate the $Q$-function , $Q^{\pi}(s, a) \approx \boldsymbol{\phi}(s, a)^{\top} \boldsymbol{\theta}$ where $\boldsymbol{\theta} \in \mathbb{R}^p$ is the learnable parameter. The $Q$-function may not be exactly represented by the feature, therefore we consider the following projected version of Bellman optimality equation (Sutton et al., 2008), which is motivated from

---

[1]See Appendix 15 for more detail.

solving $\min_{\boldsymbol{\theta}\in\mathbb{R}^p} \frac{1}{2}\|\boldsymbol{y} - \boldsymbol{\Phi}\boldsymbol{\theta}\|^2_{\boldsymbol{D}_{\nu_{\boldsymbol{\theta}}}}$ where $\boldsymbol{y} = \boldsymbol{\Phi}(\boldsymbol{\Phi}^\top \boldsymbol{D}_{\nu_{\boldsymbol{\theta}}}\boldsymbol{\Phi})^{-1}\boldsymbol{\Phi}^\top \boldsymbol{D}_{\nu_{\boldsymbol{\theta}}}(\boldsymbol{R} + \gamma\boldsymbol{P}\boldsymbol{\Pi}_{\pi_{\boldsymbol{\theta}}}\boldsymbol{\Phi}\boldsymbol{\theta})$:

$$\boldsymbol{F}(\boldsymbol{\theta}, \pi_{\boldsymbol{\theta}}, \nu_{\boldsymbol{\theta}}) := \boldsymbol{\Phi}^\top \boldsymbol{D}_{\nu_{\boldsymbol{\theta}}}\boldsymbol{R} + \boldsymbol{T}(\boldsymbol{\theta}, \pi_{\boldsymbol{\theta}}, \nu_{\boldsymbol{\theta}})\boldsymbol{\theta} = \boldsymbol{0}, \tag{1}$$

$$\boldsymbol{T}(\boldsymbol{\theta}, \pi_{\boldsymbol{\theta}}, \nu_{\boldsymbol{\theta}}) := \gamma\boldsymbol{\Phi}^\top \boldsymbol{D}_{\nu_{\boldsymbol{\theta}}}\boldsymbol{P}\boldsymbol{\Pi}_{\pi_{\boldsymbol{\theta}}}\boldsymbol{\Phi} - \boldsymbol{\Phi}^\top \boldsymbol{D}_{\nu_{\boldsymbol{\theta}}}\boldsymbol{\Phi}. \tag{2}$$

where the sampling distribution $\nu_{\boldsymbol{\theta}} \in \Delta^{\mathcal{A}}$ and the target policy $\pi_{\boldsymbol{\theta}} : \mathcal{S} \to \Delta^{\mathcal{A}}$ are parameterized by $\boldsymbol{\theta} \in \mathbb{R}^p$, the matrix $\boldsymbol{\Phi} \in \mathbb{R}^{|\mathcal{S}||\mathcal{A}|\times p}$ has its $(s-1)|\mathcal{A}| + a$-th row corresponding to the vector $\boldsymbol{\phi}(s,a)^\top$, and the matrix $\boldsymbol{\Pi}_\pi \in \mathbb{R}^{|\mathcal{S}|\times|\mathcal{S}||\mathcal{A}|}$ has the $s$-th row vector given by $(\boldsymbol{e}_s \otimes \boldsymbol{\pi}(s))^\top$, where $\boldsymbol{\pi}(s) \in \mathbb{R}^{|\mathcal{A}|}$ satisfies $[\boldsymbol{\pi}(s)]_a = \pi(a \mid s)$ and $\boldsymbol{e}_s$ is the unit vector with a value of one at the $s$-th position and zeros elsewhere . The diagonal matrix $\boldsymbol{D}_{\nu_{\boldsymbol{\theta}}} \in \mathbb{R}^{|\mathcal{S}||\mathcal{A}|\times|\mathcal{S}||\mathcal{A}|}$ has $(s-1)|\mathcal{A}| + a$-th diagonal entry as $\nu_{\boldsymbol{\theta}}(s,a)$. $\nu_{\boldsymbol{\theta}}$ can be set as the stationary distribution induced by Markov chain using a behavior policy $\beta_{\boldsymbol{\theta}} : \mathcal{S} \to \Delta^{\mathcal{A}}$, which we denote as $\mu_{\beta_{\boldsymbol{\theta}}}$. We assume it to be unique and existent throughout the paper, which is standard in the literature (Meyn, 2024; Liu et al., 2025):

**Assumption 2.1.** *Every element in the closure of $\{\boldsymbol{P}\boldsymbol{\Pi}_{\beta_{\boldsymbol{\theta}}} : \boldsymbol{\theta} \in \mathbb{R}^p\}$ induces an irreducible and aperiodic Markov chain.*

Note that $\nu_{\boldsymbol{\theta}}$ in (1) can be also set as some arbitrary fixed probability distribution $d \in \Delta^{\mathcal{S}\times\mathcal{A}}$ such that $d(s,a) > 0$ for all $(s,a) \in \mathcal{S}\times\mathcal{A}$ when we can sample state action pair from a fixed distribution, for example using a experience replay buffer (Lin, 1992).

Meanwhile, the solution to (1) may not exist. To ensure the existence of a solution, we can add an additional term $\eta\boldsymbol{\theta}$ (for some positive real number $\eta$) to (1), which can be interpreted as the regularized PBE (3) (Zhang et al., 2021; Lim and Lee, 2024).

$$\boldsymbol{F}_\eta(\boldsymbol{\theta}, \pi_{\boldsymbol{\theta}}, \nu_{\boldsymbol{\theta}}) := \boldsymbol{\Phi}^\top \boldsymbol{D}_{\nu_{\boldsymbol{\theta}}}\boldsymbol{R} + \boldsymbol{T}(\boldsymbol{\theta}, \pi_{\boldsymbol{\theta}}, \nu_{\boldsymbol{\theta}})\boldsymbol{\theta} - \eta\boldsymbol{\theta} = \boldsymbol{0}. \tag{3}$$

## 3 PROJECTED BELLMAN EQUATION

In this section, we discuss the existence and uniqueness of solution of PBE in (1). It is known that the solution of (1) might not exist or there might be multiple depending on the choice of behavior policy and target policy (De Farias and Van Roy, 2000; Bertsekas, 2011). Section 3.1 considers a condition using SNRDD and Section 3.2 provides a condition motivated from the AVI algorithm. The relationship between these two conditions is thoroughly examined in Section 3.3.

### 3.1 SNRDD GUARANTEES EXISTENCE AND UNIQUENESS OF SOLUTION TO (1)

Let us introduce a condition that guarantees the existence and uniqueness of the solution of (1). The key concept we leverage is the strictly negatively row dominating diagonal (SNRDD) condition:

**Definition 3.1** (Molchanov and Pyatnitskiy (1989)). *A matrix $\boldsymbol{A} \in \mathbb{R}^{n\times n}$ is said to have strictly negatively row dominating diagonal if $S_i(\boldsymbol{A}) := [\boldsymbol{A}]_{i,i} + \sum_{j\in[n]\setminus\{i\}}|[\boldsymbol{A}]_{i,j}| < 0$ for all $i \in [n]$.*

For simplicity, we will call a matrix $\boldsymbol{A}$ is SNRDD if it satisfies Definition 3.1. The above condition has been widely used in determining the stability of a dynamical system (Molchanov and Pyatnitskiy, 1989) or analysis of fixed point problem (Davydov et al., 2024a), which is summarized in Appendix Section 9. We explore the solution of PBE with this assumption and consider various behavior and target policy scenarios. Now, let us consider a parameterized form of SNRDD, for $\boldsymbol{M}_{\boldsymbol{\theta}} \in \mathbb{R}^{p\times p}$, a matrix dependent on $\boldsymbol{\theta}$, and for some set $\mathcal{D} \subseteq \mathbb{R}^p$:

$$\sup_{\boldsymbol{\theta}\in\mathcal{D}}\max_{i\in[p]} S_i(\boldsymbol{M}_{\boldsymbol{\theta}}) < 0. \tag{4}$$

where $S_i$ is defined in Definition 3.1, and we call the above inequality as *condition (4) with $(\mathcal{D}, \boldsymbol{M}_{\boldsymbol{\theta}})$*.

Depending on the choice of behavior and target policy, the existence of solution to PBE differs. A policy $\pi_{\boldsymbol{\theta}}$ is said to be continuous if it is continuous with respect to $\boldsymbol{\theta}$, and Lipschitz if $|\pi_{\boldsymbol{\theta}}(a \mid s) - \pi_{\tilde{\boldsymbol{\theta}}}(a \mid s)| \leq L\|\boldsymbol{\theta} - \tilde{\boldsymbol{\theta}}\|$ for some norm $\|\cdot\|$ and a positive real number $L$. Typical examples of Lipschitz policies are the greedy policy and the $\epsilon$-softmax policy, as discussed in Appendix 10.3.

**Theorem 3.2.** *1. Assume that both the behavior policy, $\beta_{\boldsymbol{\theta}}$, and the target policy, $\pi_{\boldsymbol{\theta}}$, are continuous. Suppose the parameterized SNRDD condition in (4) holds with $(\mathbb{R}^p, \boldsymbol{T}(\boldsymbol{\theta}, \pi_{\boldsymbol{\theta}}, \mu_{\beta_{\boldsymbol{\theta}}}))$. Then, a solution of $\boldsymbol{F}(\boldsymbol{\theta}, \pi_{\boldsymbol{\theta}}, \mu_{\beta_{\boldsymbol{\theta}}}) = \boldsymbol{0}$ defined in (1) exists.*

2. *Suppose* $||\boldsymbol{\Phi}^\top(\boldsymbol{D}_{\mu_{\beta_{\boldsymbol{\theta}}}} - \boldsymbol{D}_{\mu_{\beta_{\boldsymbol{\theta}'}}})\boldsymbol{R}||_\infty \leq l||\boldsymbol{\theta} - \boldsymbol{\theta}'||_\infty$ *for* $\boldsymbol{\theta}, \boldsymbol{\theta}' \in \mathbb{R}^p$ *and* $l < |\sup_{\boldsymbol{\theta} \in \mathcal{D}} \max_{i \in [p]} S_i(\boldsymbol{T}(\boldsymbol{\theta}, \pi_{\boldsymbol{\theta}}, \mu_{\beta_{\boldsymbol{\theta}}})|$, *and the condition in (4) holds with* $(\mathcal{D}, \boldsymbol{T}(\boldsymbol{\theta}, \pi_{\boldsymbol{\theta}}, \mu_{\beta_{\boldsymbol{\theta}}}))$ *where* $\mathcal{D}$ *is the set of all differentiable points of* $\boldsymbol{F}(\boldsymbol{\theta}, \pi_{\boldsymbol{\theta}}, \mu_{\beta_{\boldsymbol{\theta}}})$. *Then, a solution of* $\boldsymbol{F}(\boldsymbol{\theta}, \pi_{\boldsymbol{\theta}}, \mu_{\beta_{\boldsymbol{\theta}}}) = \boldsymbol{0}$ *exists and is unique.*

The proof, given in Appendix 12.1, uses standard methods of fixed point theory (Brouwer, 1911; Banach, 1922). The first condition in the second item naturally holds when $\beta_{\boldsymbol{\theta}}$ is a fixed policy.

**Remark 3.3.** *De Farias and Van Roy (2000) proved the existence of the solution when the behavior and target policy are identical (the on-policy case), and they are continuous. In contrast, we allow scenarios under different behavior and target policy, i.e., the off-policy case. Meyn (2024) proved that using a particular type of $\epsilon$-softmax policy, so-called $(\epsilon, \kappa_0)$-tamed Gibbs policy (detailed in Appendix 10.3), ensures the existence of a solution of PBE. This covers different scenario from ours as using a $(\epsilon, \kappa_0)$-tamed Gibbs policy, does not necessarily imply SNRDD condition. Moreover, it requires knowledge of the model parameters of MDP, i.e., $\lambda_{\min}(\boldsymbol{\Phi}^\top \boldsymbol{D}_{\mu_{\beta_{\boldsymbol{\theta}}}} \boldsymbol{\Phi})$.*

**Remark 3.4.** *For a Lipschitz target policy $\pi_{\boldsymbol{\theta}}$ and behavior policy $\beta_{\boldsymbol{\theta}}$, $\boldsymbol{F}(\boldsymbol{\theta}, \pi_{\boldsymbol{\theta}}, \mu_{\beta_{\boldsymbol{\theta}}})$ is a locally Lipschitz function (defined in Definition 10.1 in the Appendix), which is differentiable almost everywhere by Rademacher's theorem (Evans, 2018).*

**Remark 3.5.** *The condition in (4) holds with $(\mathbb{R}^p, \boldsymbol{T}(\boldsymbol{\theta}, \pi_{\boldsymbol{\theta}}, \mu_{\beta_{\boldsymbol{\theta}}}))$ when $\boldsymbol{\Phi} = \boldsymbol{I}$ where $\boldsymbol{I}$ is a $|\mathcal{S}||\mathcal{A}| \times |\mathcal{S}||\mathcal{A}|$ identity matrix, and behavior policy satisfies the condition $\inf_{\boldsymbol{\theta} \in \mathbb{R}^p} \min_{(s,a) \in \mathcal{S} \times \mathcal{A}} \mu_{\beta_{\boldsymbol{\theta}}}(s, a) > 0$. This corresponds to the tabular setup of PBE,*

Considering the solution of PBE, as the feature dimension $p$ increases, it becomes more challenging to satisfy condition (4) due to the growing column size. One simple way to address this issue is to consider a matrix with additional scaled identity matrix, i.e., $\boldsymbol{T}(\boldsymbol{\theta}, \pi_{\boldsymbol{\theta}}, \mu_{\beta_{\boldsymbol{\theta}}}) - \eta\boldsymbol{I}$ for $\eta > 0$. This yields the regularized version of PBE given in (3), and the same arguments in Theorem 3.2 hold for the solution to (3). The SNRDD assumption can be satisfied with the following choice of $\eta$:

**Lemma 3.6.** *If $\eta > \sup_{\boldsymbol{\theta} \in \mathbb{R}^p} \max_{i \in [p]} S_i(\boldsymbol{T}(\boldsymbol{\theta}, \pi_{\boldsymbol{\theta}}, \mu_{\beta_{\boldsymbol{\theta}}}))$, (4) holds with $(\mathbb{R}^p, \boldsymbol{T}(\boldsymbol{\theta}, \pi_{\boldsymbol{\theta}}, \mu_{\beta_{\boldsymbol{\theta}}}) - \eta\boldsymbol{I})$.*

**Remark 3.7.** *When feature scaling is used, $||\phi(s, a)||_\infty < 1/\sqrt{p}$ for all $(s, a) \in \mathcal{S} \times \mathcal{A}$, then $\eta > 3$ is sufficient to meet the above condition. The proof is given in Lemma 12.1 in Appendix 12. We note that this condition does not depend on any model parameters of the MDP, for example $\lambda_{\min}(\boldsymbol{\Phi}^\top \boldsymbol{D}_{\mu_{\beta_{\boldsymbol{\theta}}}} \boldsymbol{\Phi})$.*

**Remark 3.8.** *The SNRDD condition was also considered in Lim and Lee (2024) but only in terms of convergence of regularized Q-learning but not existence of solution, and it requires additional assumptions including positiveness and orthogonality on the feature matrix. In Section 4, we show that only SNRDD condition is required for proving the convergence of regularized Q-learning.*

**Remark 3.9.** *For (3), when $\boldsymbol{\Phi} = \boldsymbol{I}$, then $\eta > 0$ implies using a smaller discount factor, $\gamma$ (Chen et al., 2023). Nonetheless, the interpretation is more complex when $\boldsymbol{\Phi} \neq \boldsymbol{I}$, and algorithms to sovle (3) has been widely used in practice (Farebrother et al., 2018; Cobbe et al., 2019).*

### 3.2 AVI AND EXISTENCE AND UNIQUENESS OF SOLUTION OF (1)

Meanwhile, let us investigate another sufficient condition to guarantee the existence of solution of PBE in (1), which is motivated from the AVI algorithm. We can re-write (1) as

$$\boldsymbol{\theta} = (\boldsymbol{\Phi}^\top \boldsymbol{D}_{\mu_{\beta_{\boldsymbol{\theta}}}} \boldsymbol{\Phi})^{-1}(\gamma\boldsymbol{\Phi}^\top \boldsymbol{D}_{\mu_{\beta_{\boldsymbol{\theta}}}} \boldsymbol{P}\boldsymbol{\Pi}_{\pi_{\boldsymbol{\theta}}} \boldsymbol{\Phi}\boldsymbol{\theta} + \boldsymbol{\Phi}^\top \boldsymbol{D}_{\mu_{\beta_{\boldsymbol{\theta}}}} \boldsymbol{R}) \quad (5)$$

assuming invertibility of $\boldsymbol{\Phi}^\top \boldsymbol{D}_{\mu_{\beta_{\boldsymbol{\theta}}}} \boldsymbol{\Phi}$. Therefore, a closely related condition to guarantee the existence and uniqueness of the solution to (1) is that for $\mathcal{D} \subseteq \mathbb{R}^p$, a set to be defined further, one of the following two conditions hold:

$$\begin{cases} \sup_{\boldsymbol{\theta} \in \mathcal{D}} \gamma\|\boldsymbol{\Phi}(\boldsymbol{\Phi}^\top \boldsymbol{D}_{\mu_{\beta_{\boldsymbol{\Phi}\boldsymbol{\theta}}}} \boldsymbol{\Phi})^{-1}\boldsymbol{\Phi}^\top \boldsymbol{D}_{\mu_{\beta_{\boldsymbol{\Phi}\boldsymbol{\theta}}}} \boldsymbol{P}\boldsymbol{\Pi}_{\pi_{\boldsymbol{\Phi}\boldsymbol{\theta}}}\|_\infty < 1, & (6) \\ \sup_{\boldsymbol{\theta} \in \mathcal{D}} \gamma\|(\boldsymbol{\Phi}^\top \boldsymbol{D}_{\mu_{\beta_{\boldsymbol{\theta}}}} \boldsymbol{\Phi})^{-1}\boldsymbol{\Phi}^\top \boldsymbol{D}_{\mu_{\beta_{\boldsymbol{\theta}}}} \boldsymbol{P}\boldsymbol{\Pi}_{\pi_{\boldsymbol{\theta}}} \boldsymbol{\Phi}\|_\infty < 1. & (7) \end{cases}$$

Note that the policies in (6) are dependent on $\boldsymbol{\Phi}\boldsymbol{\theta}$. As in Section 3, the following results can be derived using standard fixed point theory arguments, and the proof is deferred to Appendix 12.2.

**Theorem 3.10.** *1. Suppose $\beta_{\boldsymbol{\theta}}$ and $\pi_{\boldsymbol{\theta}}$ are continuous. Moreover, assume that either (6) or (7) holds with $\mathcal{D} = \mathbb{R}^p$, and $0 < \inf_{\boldsymbol{\theta} \in \mathcal{D}} \lambda_{\min}(\boldsymbol{\Phi}^\top \boldsymbol{D}_{\mu_{\beta_{\boldsymbol{\theta}}}} \boldsymbol{\Phi})$. Then, a solution of (1) exists.*

*2. Suppose a fixed behavior policy $\beta$ is used and $\pi_{\boldsymbol{\theta}}$ is Lipschitz. Moreover, assume that either (6) or (7) holds with $\mathcal{D}$ being all the differentiable points of $\boldsymbol{F}(\boldsymbol{\theta}, \pi_{\boldsymbol{\theta}}, \beta)$, and $0 < \inf_{\boldsymbol{\theta} \in \mathcal{D}} \lambda_{\min}(\boldsymbol{\Phi}^\top \boldsymbol{D}_{\mu_\beta} \boldsymbol{\Phi})$. Then, a solution of (1) exists and is unique.*

**Remark 3.11.** *One can replace the infinity norm with joint spectral radius, which is defined as, given a set of square matrices $\{\boldsymbol{A}_i \in \mathbb{R}^{n \times n}\}_{i=1}^m$, $\rho(\boldsymbol{A}_1, \cdots, \boldsymbol{A}_m) = \lim_{k \to \infty} \max_{\sigma \in \{1,2,\ldots,m\}^k} \|\boldsymbol{A}_{\sigma_k} \cdots \boldsymbol{A}_{\sigma_2} \boldsymbol{A}_{\sigma_1}\|^{1/k}$. If $\rho(\boldsymbol{A}_1, \ldots, \boldsymbol{A}_m) < 1$, there exists a norm $\|\cdot\|$ such that $\|\boldsymbol{A}_i\| < 1$ for all $i \in [m]$ (Rota and Strang, 1960). Therefore, we can replace the infinity norm with this common norm in (6) or (7).*

*It is important to note that each matrix $\boldsymbol{A}_i$ having a spectral radius smaller than one — the maximum absolute value of its eigenvalues — does not imply Theorem 3.10. This is because it does not guarantee the existence of a common norm $\|\cdot\|$ such that $\|\boldsymbol{A}_i\| < 1$ (Jungers, 2009).*

**Remark 3.12.** *It is challenging to ensure when (6) or (7) hold in practice. Alternatively, one may consider a form motivated from the regularized PBE in (3) by replacing $(\boldsymbol{\Phi}^\top \boldsymbol{D}_{\mu_{\beta_{\boldsymbol{\theta}}}} \boldsymbol{\Phi})^{-1}$ with $(\boldsymbol{\Phi}^\top \boldsymbol{D}_{\mu_{\beta_{\boldsymbol{\theta}}}} \boldsymbol{\Phi} + \eta \boldsymbol{I})^{-1}$, and ensure the solution of (3).*

*Zhang et al. (2021) showed the existence of a solution to (3), regularized PBE, whereas extension of Theorem 3.10 with regularization can guarantee uniqueness. Lim and Lee (2024) showed the uniqueness of the solution but we sharpen the bound from $\gamma \|\boldsymbol{\Phi}(\boldsymbol{\Phi}^\top \boldsymbol{D}_{\mu_{\beta_{\boldsymbol{\Phi}\boldsymbol{\theta}}}} \boldsymbol{\Phi} + \eta \boldsymbol{I})^{-1} \boldsymbol{\Phi} \boldsymbol{D}_{\mu_{\beta_{\boldsymbol{\Phi}\boldsymbol{\theta}}}}\|_\infty < 1$ to $\gamma \|\boldsymbol{\Phi}(\boldsymbol{\Phi}^\top \boldsymbol{D}_{\mu_{\beta_{\boldsymbol{\Phi}\boldsymbol{\theta}}}} \boldsymbol{\Phi} + \eta \boldsymbol{I})^{-1} \boldsymbol{\Phi} \boldsymbol{D}_{\mu_{\beta_{\boldsymbol{\Phi}\boldsymbol{\theta}}}} \boldsymbol{P} \boldsymbol{\Pi}_{\pi_{\boldsymbol{\Phi}\boldsymbol{\theta}}}\|_\infty < 1$ from (6). This follows from the application of a version of mean value theorem in Lemma 10.5 in the Appendix 10.1.*

## 3.3 DISCUSSION ON THE CONDITION (4) AND (7)

Letting $\boldsymbol{M}_{\boldsymbol{\theta}} = \boldsymbol{T}(\boldsymbol{\theta}, \pi_{\boldsymbol{\theta}}, \mu_{\beta_{\boldsymbol{\theta}}})$ in (4), we now examine when the conditions (4) and (7) imply each other. While either condition guarantees the existence of a solution of PBE, they are closely tied to the convergence of Q-learning and the AVI, respectively, which we defer the discussion to Section 5. Below, we present a result on the relationship between conditions (4) and (7).

**Proposition 3.13.** *If $\inf_{\boldsymbol{\theta} \in \mathcal{D}} \lambda_{\min}(\boldsymbol{\Phi}^\top \boldsymbol{D}_{\mu_{\beta_{\boldsymbol{\theta}}}} \boldsymbol{\Phi}) > 0$ for some $\mathcal{D} \subseteq \mathbb{R}^p$, the following holds:*

*1) Suppose (4) holds with $(\mathcal{D}, \boldsymbol{T}(\boldsymbol{\theta}, \pi_{\boldsymbol{\theta}}, \mu_{\beta_{\boldsymbol{\theta}}}))$, and $\boldsymbol{\Phi}^\top \boldsymbol{D}_{\mu_{\beta_{\boldsymbol{\theta}}}} \boldsymbol{P} \boldsymbol{\Pi}_{\pi_{\boldsymbol{\theta}}} \boldsymbol{\Phi}$ has non-negative diagonal elements for all $\boldsymbol{x} \in \mathcal{D}$. Then, (7) holds with $\mathcal{D}$.*

*2) Suppose (7) holds with $\mathcal{D}$. Then, (4) holds with $(\mathcal{D}, \boldsymbol{T}(\boldsymbol{\theta}, \pi_{\boldsymbol{\theta}}, \mu_{\beta_{\boldsymbol{\theta}}}))$.*

The proof is given in Appendix 12.3. If $\boldsymbol{\Phi}^\top \boldsymbol{D}_{\mu_{\beta_{\boldsymbol{\theta}}}} \boldsymbol{\Phi}$ is a diagonal matrix, and diagonal elements of $\boldsymbol{\Phi}^\top \boldsymbol{D}_{\mu_{\beta_{\boldsymbol{\theta}}}} \boldsymbol{P} \boldsymbol{\Pi}_{\pi_{\boldsymbol{\theta}}} \boldsymbol{\Phi}$ are non-negative, then the conditions (4) and (7) are equivalent. The diagonal elements of $\boldsymbol{\Phi}^\top \boldsymbol{D}_{\mu_{\beta_{\boldsymbol{\theta}}}} \boldsymbol{P} \boldsymbol{\Pi}_{\pi_{\boldsymbol{\theta}}} \boldsymbol{\Phi}$ can be non-negative if each entry of $\boldsymbol{\Phi}$ has non-negative values.

We note that condition (6) also guarantees solution existence, though its relationship with condition (4) is difficult to characterize. As these conditions are linked to convergence of AVI and Q-learning respectively, in Section 5, we present an example where only one of the conditions–either (4) or (6)– is met, leading to the convergence of its corresponding algorithm, while the other fails. Moreover, it is not clear whether we can construct such example with condition (7), requiring further research.

## 4 CONVERGENCE OF Q-LEARNING

In this section, we briefly review the Q-learning algorithm, and prove its convergence by ordinary differential equation (ODE) analysis using the parameterized SNRDD condition in (4). We consider an i.i.d. sampling model from an arbitrary fixed distribution $d \in \Delta^{\mathcal{S} \times \mathcal{A}}$. The analysis can be easily extended to the Markovian observation model observing a single trajectory, using the arguments in Liu et al. (2024). Upon observing $(s_k, a_k) \sim d(\cdot)$, $s'_k \sim \mathcal{P}(\cdot \mid s_k, a_k)$ and $r_k := r(s_k, a_k, s'_k)$ independently for every $k$-th iteration, the parameter of (regularized) Q-learning using step-size

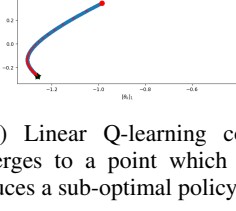

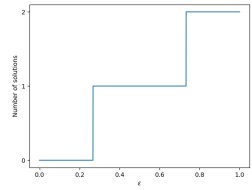

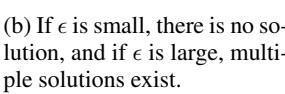

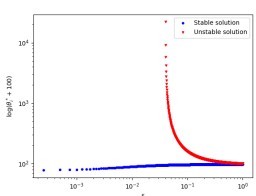

(a) Linear Q-learning converges to a point which induces a sub-optimal policy.

(b) If $\epsilon$ is small, there is no solution, and if $\epsilon$ is large, multiple solutions exist.

(c) (Example 14.2) Increasing $\epsilon$ adds an unstable solution. $r_1 = -0.1$ and $r_2 = -0.78$.

Figure 1: The first and second figure show Example 13.3 and 14.1 in Appendix 13, respectively. In the last figure, stable and unstable refers to whether $\boldsymbol{T}(\boldsymbol{\theta}, \pi_{\boldsymbol{\theta}}, \mu_{\beta_{\boldsymbol{\theta}}})$ is a Hurwitz matrix at each point.

$\alpha_k \in (0,1)$ satisfying $\sum_{k \in \mathbb{N}} \alpha_k = \infty, \sum_{k \in \mathbb{N}} \alpha_k^2 < \infty$, and $\eta \in \mathbb{R}$, is updated as follows:

$$\boldsymbol{\theta}_{k+1} = \boldsymbol{\theta}_k + \alpha_k \boldsymbol{\phi}(s_k, a_k)(r_k + \gamma \max_{a \in \mathcal{A}} \boldsymbol{\phi}^\top(s_k', a)\boldsymbol{\theta}_k - \boldsymbol{\phi}(s_k, a_k)^\top \boldsymbol{\theta}_k - \eta \boldsymbol{\theta}_k), \quad \boldsymbol{\theta}_0 \in \mathbb{R}^p \quad (8)$$

### 4.1 STOCHASTIC APPROXIMATION AND ODE APPROACH

Q-learning can be understood as a stochastic approximation scheme (Robbins and Monro, 1951):

$$\boldsymbol{x}_{k+1} = \boldsymbol{x}_k + \alpha_k(\boldsymbol{A}_{\sigma(\boldsymbol{x}_k)}\boldsymbol{x}_k + \boldsymbol{b} + \boldsymbol{\epsilon}_{k+1}), \quad \boldsymbol{x}_0 \in \mathbb{R}^p \quad (9)$$

where $\sigma : \mathbb{R}^p \to \mathcal{M}$ is a switching signal, $\mathcal{M} := \{1, 2, \ldots, |\mathcal{M}|\}$ is the set of modes, $\boldsymbol{b}$ is a constant vector and $\{\boldsymbol{A}_m : m \in \mathcal{M}\}$ are square matrices. $\boldsymbol{\epsilon}_k$ is Martingale-difference sequence defined in Definition 11.1. The almost sure convergence of (9) is closely related to its ODE counterpart:

$$\dot{\boldsymbol{x}}_t = \boldsymbol{A}_{\sigma(\boldsymbol{x}_t)}\boldsymbol{x}_t + \boldsymbol{b}, \quad \boldsymbol{x}_0 \in \mathbb{R}^p, t \geq 0 \quad (10)$$

where $\frac{d}{dt}\boldsymbol{x}_t = \dot{\boldsymbol{x}}_t$. Loosely speaking, the asymptotic behavior of $\boldsymbol{x}_k$ in (9) is governed by its corresponding ODE if it admits a globally asymptotically stable equilibrium point. An equilibrium point is a vector $\boldsymbol{x}^* \in \mathbb{R}^p$ that satisfies $\boldsymbol{A}_{\sigma(\boldsymbol{x}^*)}\boldsymbol{x}^* + \boldsymbol{b} = \boldsymbol{0}$ and global asymptotic stability means that the solutions $\boldsymbol{x}_t$ converge to $\boldsymbol{x}^*$ regardless of the initial condition $\boldsymbol{x}_0$. A detailed argument is given by Borkar and Meyn Theorem (Borkar and Meyn, 2000) provided in Appendix 11. A key concept in verifying a globally asymptotically stable equilibrium point is the so-called one-sided Lipschitzness:

**Definition 4.1** (One-sided Lipschitz, Definition 3.2 in Bullo (2024)). *For $\mathcal{D} \subseteq \mathbb{R}^p$, if $\boldsymbol{f} : \mathcal{D} \to \mathbb{R}^p$ satisfies the following, it is called one-sided Lipschitz with constant $b$: $[\boldsymbol{f}(\boldsymbol{x}) - \boldsymbol{f}(\boldsymbol{y})]_i[\boldsymbol{x} - \boldsymbol{y}]_i \leq b \|\boldsymbol{x} - \boldsymbol{y}\|_\infty^2$ where $i \in \mathcal{I}_\infty(\boldsymbol{x} - \boldsymbol{y}) := \{j \in [p] : |[\boldsymbol{x} - \boldsymbol{y}]_j| = \|\boldsymbol{x} - \boldsymbol{y}\|_\infty\}$ and $\boldsymbol{x}, \boldsymbol{y} \in \mathbb{R}^p$.*

In fact, for a locally Lipschitz function $\boldsymbol{f} : \mathbb{R}^p \to \mathbb{R}^p$, Definition 3.1 holding for $\nabla \boldsymbol{f}(\boldsymbol{x})$–the gradient at differentiable point of $\boldsymbol{f}$–at all such points is equivalent to the one-sided Lipschitz condition (Davydov et al., 2024a) (see Lemma 10.8 in Appendix 10.1).

If $\boldsymbol{f}$ is one-sided Lipschitz with a negative constant, then every pair of trajectories of (10) are contracting, i.e., for solutions $\boldsymbol{x}_t$ and $\boldsymbol{y}_t$ with different initial conditions $\boldsymbol{x}_0$ and $\boldsymbol{y}_0$, we have $\|\boldsymbol{x}_t - \boldsymbol{y}_t\|_\infty \to 0$. This is known as contraction theory (Lohmiller and Slotine, 1998), and if a unique equilibrium exists, all trajectories converge to it, and it is globally asymptotically stable.

**Lemma 4.2** (Theorem 3.9 in Bullo (2024)). *Suppose the condition in Definition 4.1 holds for $\boldsymbol{f}(\boldsymbol{x}) := \boldsymbol{A}_{\sigma(\boldsymbol{x})}\boldsymbol{x} + \boldsymbol{b}$ for $\mathbb{R}^p$ with some $c < 0$ and $\boldsymbol{f}$ is a Lipschitz function. Then, there exists a unique $\boldsymbol{x}^* \in \mathbb{R}^p$ such that $\boldsymbol{A}_{\sigma(\boldsymbol{x}^*)}\boldsymbol{x}^* + \boldsymbol{b} = \boldsymbol{0}$ which is globally asymptotically stable.*

### 4.2 ANALYSIS OF Q-LEARNING ALGORITHMS

Now, using the ODE arguments introduced in the previous section, we prove that ODE counterparts of a family of Q-learning algorithms admit a globally asymptotically stable equilibrium point. Let us consider the following ODE counterpart of the Q-learning algorithm in (8):

$$\dot{\boldsymbol{\theta}}_t = \boldsymbol{\Phi}^\top \boldsymbol{D}_d \boldsymbol{R} + \gamma \boldsymbol{\Phi}^\top \boldsymbol{D}_d \boldsymbol{P} \boldsymbol{\Pi}_{\pi_{\boldsymbol{\theta}_t}^g} \boldsymbol{\Phi} \boldsymbol{\theta}_t - (\boldsymbol{\Phi}^\top \boldsymbol{D}_d \boldsymbol{\Phi} + \eta \boldsymbol{I})\boldsymbol{\theta}_t, \quad \boldsymbol{\theta}_0 \in \mathbb{R}^p, \ t \geq 0,$$

where $\pi_{\boldsymbol{\theta}}^g$ denotes the greedy policy over $\boldsymbol{\Phi\theta}$, i.e., $\pi_{\boldsymbol{\theta}}^g(s) = \arg\max_{a \in \mathcal{A}} \boldsymbol{\phi}(s,a)^\top \boldsymbol{\theta}$, and a fixed tie-breaking rule is applied when it is not a singleton set. When $\eta = 0$, it coincides with the update rule of linear Q-learning, and if additionally $\boldsymbol{\Phi} = \boldsymbol{I}$, the algorithm reduces to asynchronous tabular Q-learning. If $\eta > 0$, the algorithm becomes regularized Q-learning. For each case, we can verify that the one-sided Lipschitz condition in Definition 4.1 holds under the SNRDD condition in (4) (which is necessary and sufficient condition for a locally Lipschitz map (Davydov et al., 2024a)):

**Lemma 4.3.** *The following holds depending on the choice of $\boldsymbol{\Phi}$ and $\eta$:*

*1) Let $\eta = 0$ and $\boldsymbol{\Phi} = \boldsymbol{I}$. For $\boldsymbol{Q} \in \mathbb{R}^{|\mathcal{S}||\mathcal{A}|}$, let $\boldsymbol{F}_{\mathrm{AsyncQ}}(\boldsymbol{Q}) = \boldsymbol{F}(\boldsymbol{Q}, \pi_{\boldsymbol{Q}}^g, d)$. Then, $\boldsymbol{F}_{\mathrm{AsyncQ}}(\boldsymbol{Q})$ is one-sided Lipschitz with constant $(\gamma - 1)d_{\min}$ where $d_{\min} := \min_{(s,a) \in \mathcal{S} \times \mathcal{A}} d(s,a)$.*

*2) Let $\eta = 0$ and suppose (4) holds with $(\mathcal{D}_{\boldsymbol{F}_{\mathrm{linear}}}, \boldsymbol{T}(\boldsymbol{\theta}, \pi_{\boldsymbol{\theta}}^g, d))$ where $\mathcal{D}_{\boldsymbol{F}_{\mathrm{linear}}}$ is the set of differentiable points of $\boldsymbol{F}_{\mathrm{linear}}(\boldsymbol{\theta}) := \boldsymbol{F}(\boldsymbol{\theta}, \pi_{\boldsymbol{\theta}}^g, d)$. Then, $\boldsymbol{F}_{\mathrm{linear}}(\boldsymbol{\theta})$ is one-sided Lipschitz with constant $a_{\min} := \sup_{\boldsymbol{\theta} \in \mathcal{D}_{\boldsymbol{F}_{\mathrm{linear}}}} \max_{i \in [p]} S_i(\boldsymbol{T}(\boldsymbol{\theta}, \pi_{\boldsymbol{\theta}}^g, d))$ which is defined in Definition 3.1.*

*3) Let $\eta > \sup_{\boldsymbol{\theta} \in \mathcal{D}_{\boldsymbol{F}_{\mathrm{linear}}}} \max_{i \in [p]} S_i(\boldsymbol{T}(\boldsymbol{\theta}, \pi_{\boldsymbol{\theta}}^g, d))$, and denote $\boldsymbol{F}_{\mathrm{Reg}}(\boldsymbol{\theta}) := \boldsymbol{F}_\eta(\boldsymbol{\theta}, \pi_{\boldsymbol{\theta}}^g, d)$. Then, $\boldsymbol{F}_{\mathrm{Reg}}(\boldsymbol{\theta})$ is one-sided Lipschitz with constant $-\eta + a_{\min}$.*

The proof is given in Appendix Section 12.4. From Theorem 3.2, the SNRDD condition ensures the uniqueness and existence of a solution, which corresponds to the globally asymptotically stable equilibrium point of the ODE counterpart of each Q-learning algorithms by Lemma 4.2. This yields the following result of which the proof is given in Appendix Section 12.5:

**Proposition 4.4.** *1) (Asynchronous tabular Q-learning) Let $\boldsymbol{\Phi} = \boldsymbol{I}$ and $\eta = 0$ in (8). Then, $\boldsymbol{\theta}_k$ in (8) converges to a solution of $\boldsymbol{F}(\boldsymbol{\theta}, \pi_{\boldsymbol{\theta}}^g, d) = \boldsymbol{0}$ which is unique, with probability one.*

*2) (Linear Q-learning) Let $\eta = 0$ in (8). Suppose the parameterized SNRDD condition (4) holds with $(\mathcal{D}_{\boldsymbol{F}_{\mathrm{linear}}}, \boldsymbol{T}(\boldsymbol{\theta}, \pi_{\boldsymbol{\theta}}^g, d))$ where $\mathcal{D}_{\boldsymbol{F}_{\mathrm{linear}}}$ is the set of differentiable points of $\boldsymbol{F}(\boldsymbol{\theta}, \pi_{\boldsymbol{\theta}}^g, d)$. Then, $\boldsymbol{\theta}_k$ in (8) converges to the unique solution of $\boldsymbol{F}(\boldsymbol{\theta}, \pi_{\boldsymbol{\theta}}^g, d) = \boldsymbol{0}$ with probability one.*

*3) (Regularized Q-learning) Let $\eta$ satisfy the condition in (4) with $(\mathcal{D}_{\boldsymbol{F}_{\mathrm{linear}}}, \boldsymbol{T}(\boldsymbol{\theta}, \pi_{\boldsymbol{\theta}}^g, d) - \eta \boldsymbol{I})$. Then, $\boldsymbol{\theta}_k$ in (8) converges to the unique solution of $\boldsymbol{F}_\eta(\boldsymbol{\theta}, \pi_{\boldsymbol{\theta}}^g, d) = \boldsymbol{0}$ with probability one.*

**Remark 4.5.** *SNRDD condition can be both applied to prove convergence of linear and tabular Q-learning, providing a unified understanding of Q-learning algorithms. For regularized Q-learning, we relax the assumptions on positiveness and orthogonality of feature matrix (Lim and Lee, 2024). Our approach is based on contraction theory, whereas Lim and Lee (2024) adopts a switched-system framework. Moreover, we do not require target network update or projection as in Zhang et al. (2021).*

**Remark 4.6.** *Melo and Ribeiro (2007) investigates convergence of Q-learning and imposes a condition on the feature function, requiring $\|\boldsymbol{\phi}(s,a)\|_\infty \leq 1$; however, as noted in its errata, the proof under this condition is incomplete. In a follow-up work, Melo et al. (2008) adopts a stronger assumption, $\|\boldsymbol{\phi}(s,a)\|_\infty = 1$, which is stronger than the condition used in our analysis, i.e., if $\|\boldsymbol{\phi}(s,a)\|_\infty = 1$, then SNRDD condition is satisfied.*

**Remark 4.7** (Convergence to sub-optimal policy). *Even though Q-learning can converge to a unique point, there is no guarantee that this point induces the optimal policy. Moreover, suppose there exist multiple solutions of PBE. $\boldsymbol{T}(\boldsymbol{\theta}, \pi_{\boldsymbol{\theta}}^g, d)$ at the solution of PBE can be SNRDD which yields a sub-optimal policy compared to the others. Then, the Q-learning algorithm may converge to this solution. A simple example is given in Example 13.3 in the Appendix 13, and its trajectories are shown in Figure 1a. This complements the observation by Gopalan and Thoppe (2024), which empirically showed that linear Q-learning can converge to the worst policy.*

## 5 APPROXIMATE VALUE ITERATION AND Q-LEARNING

In this section, we analyze the convergence of the AVI algorithm and present examples where Q-learning converges while AVI does not, and vice versa. The convergence of AVI is known to play a key role in algorithm with target network updates (Lee and He, 2020a; Chen et al., 2023). Nonetheless, its relation with Q-learning has not been well understood.

An iterative method to solve (5), the so-called AVI algorithm (De Farias and Van Roy, 2000), is

$$\boldsymbol{\theta}_{k+1} = (\boldsymbol{\Phi}^\top \boldsymbol{D}_d \boldsymbol{\Phi})^{-1} \boldsymbol{\Phi}^\top \boldsymbol{D}_d (\boldsymbol{R} + \gamma \boldsymbol{P} \boldsymbol{\Pi}_{\pi_{\boldsymbol{\theta}_k}^g} \boldsymbol{\Phi}\boldsymbol{\theta}_k), \quad \boldsymbol{\theta}_0 \in \mathbb{R}^p, \; k \in \mathbb{N}. \tag{11}$$

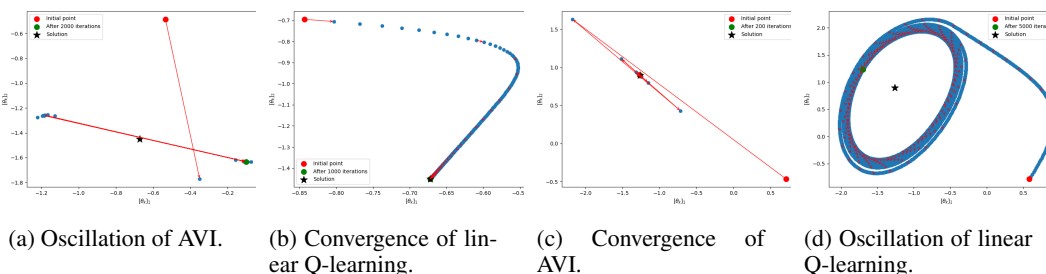

(a) Oscillation of AVI.     (b) Convergence of linear Q-learning.     (c) Convergence of AVI.     (d) Oscillation of linear Q-learning.

Figure 2: The first two and last two figures show experimental results on Example 13.1 and 13.2, respectively. For reproducibility, the experiments are done with an expected update version of Q-learning provided in Algorithm 2 in Appendix Section 17.

One can easily check that the condition in (6) or (7) ensures the convergence of (11), which is given in Lemma 12.2 in the Appendix. Now, our focus is on the relation between the convergence of AVI and Q-learning. Proposition 3.13 states a condition when both algorithms converge. Our interest is in the case when one algorithm converges while the other does not. In particular, we consider the case when the condition (4) is met but the spectral radius of the matrix $\gamma(\mathbf{\Phi}^\top \boldsymbol{D}_d \mathbf{\Phi})^{-1}(\mathbf{\Phi}^\top \boldsymbol{D}_d \boldsymbol{P} \mathbf{\Pi}_{\pi_{\boldsymbol{\theta}}^g} \mathbf{\Phi})$ at the solution is larger than one, i.e., Q-learning converges but AVI does not. [2] Likewise, we provide an example for the opposite direction, AVI converges but Q-learning does not. In this case, condition (6) is met but $\boldsymbol{T}(\boldsymbol{\theta}, \pi_{\boldsymbol{\theta}}^g, d)$ at the solution is not a Hurwitz matrix. The examples are provided in Example 13.1 and 13.2 in the Appendix 13, and the experimental results are plotted in Figure 2.

**Remark 5.1.** *For TD-learning ($|\mathcal{A}| = 1$), the spectral radius of $\gamma(\mathbf{\Phi}^\top \boldsymbol{D}_d \mathbf{\Phi})^{-1}(\mathbf{\Phi}^\top \boldsymbol{D}_d \boldsymbol{P} \mathbf{\Pi}_\pi \mathbf{\Phi})$ being smaller than one is sufficient to guarantee the convergence of AVI. Moreover, the convergence of linear Q-learning can be checked whether the matrix $\boldsymbol{T}(\boldsymbol{\theta}, \pi, d)$ is Hurwitz, i.e., the real part of the eigenvalues are all negative. Using these conditions, Wu et al. (2025) provided an example that TD-learning converges but AVI does not, and vice-versa. In contrast, we provide examples for the case when $|\mathcal{A}| \geq 2$. The spectral radius condition and Hurwitz conditions do not imply convergence of AVI and Q-learning, respectively. Moreover, SNRDD matrix is a Hurwitz matrix but the reverse does not necessarily hold, which is provided in Lemma 10.13 in the Appendix. Therefore, our examples cover different scenarios from the example in Wu et al. (2025).*

## 6 PATHOLOGICAL BEHAVIOR USING $\epsilon$-GREEDY BEHAVIOR POLICY

In this section, we examine the case when $\epsilon$-greedy policy is used, which was not addressed in the previous analysis. The first example illustrates a problem arising from this discontinuity. In particular, even though SNRDD condition is met, is shows that previous results do not extends to the $\epsilon$-greedy behavior policiy, showing its hardness in the analysis. The second example highlights a specific phenomenon resulting from the use of the $\epsilon$-greedy policy.

**Change of number of solutions:** In this example, there is a critical value for $\epsilon$, at which the number of solutions to equation (1) changes. $\epsilon$-greedy policy is used for both behavior and target policy. Consider an MDP with $|\mathcal{S}| = 1, |\mathcal{A}| = 2, p = 1, \gamma = 0.99$, and $r(1, 1, 1) = 0.5$ and $r(1, 2, 1) = 0.48$, illustrated in Example 14.1 in Appendix 14. Depending on $\epsilon$, there are three distinct regions: no solution, unique solution, and multiple solutions (Figure 1b). A critical value separates these regimes, with the number of solutions changing when this threshold is crossed. In particular, as the SNRDD condition is met, the non-existence of the solution is due to using a discontinuous policy.

Bertsekas (2011); Young and Sutton (2020) provided examples that the number of solutions changes depending on the value of transition probability or reward. Our example differs as the change is determined by the value of $\epsilon$, which reflects the degree of exploration.

---

[2]To be precise, the expected version of Q-learning does not converge to a solution at which $\boldsymbol{F}(\boldsymbol{\theta}, \pi_{\boldsymbol{\theta}}^g, \mu_{\beta_{\boldsymbol{\theta}}})$ in (1) is differentiable and $\boldsymbol{T}(\boldsymbol{\theta}, \pi_{\boldsymbol{\theta}}^g, \mu_{\beta_{\boldsymbol{\theta}}})$ is not a Hurwitz matrix. The stochastic counterpart closely follows the behavior of expected update version provided in Algorithm 2 in Appendix 17.

**Emergence of solution that yields optimal policy but to which Q-learning cannot converge:** We provide an example showing that increasing $\epsilon$ introduces a solution that induces the optimal policy but to which Q-learning cannot converge, which is illustrated in Figure 1c. Consider an MDP with $|\mathcal{S}| = 1, |\mathcal{A}| = 2$, and $\phi(1,1) = x, \phi(1,2) = y$, and the behavior and target policy are $\epsilon$-greedy and greedy policy, respectively. The reward $r(1,1,1) = r_1$ and $r(1,2,1) = r_2$ are negative constants, as illustrated in Example 14.2 in Appendix 14. There are two possible deterministic policies, say $\pi_1$ and $\pi_2$. Depending on the choice of $r_1$ and $r_2$ ($r_1 < r_2$ or $r_2 < r_2$), the optimal policy can be either $\pi_1$ or $\pi_2$. When $r_1 = -0.1 < r_2 = -0.78$, for $\epsilon < \epsilon^* \approx 0.1$, there exists a unique solution to PBE. This induces a sub-optimal policy, and Q-learning converges to this solution. For $\epsilon > \epsilon^*$, there exist two solutions: one leading to a sub-optimal policy and the other to the optimal one. However, Q-learning cannot converge to the optimal solution because $F(\theta, \pi_\theta, \mu_{\beta_\theta})$ in (1) is differentiable and $T(\theta, \pi_\theta, \mu_{\beta_\theta})$ is not Hurwitz at the corresponding point.

Young and Sutton (2020) provided an example that Q-learning can converge to a point that induces sub-optimal policy depending on the ordering of the reward (but not dependent on $\epsilon$). Lu et al. (2018) showed that for a certain regime of $\epsilon$, Q-learning can yield a sub-optimal policy compared to possible policies that can be represented by the linear feature while the optimal one is not realizable (detailed in Appendix 15). Our example shows a different scenario that we can tune $\epsilon$ to ensure a solution of PBE that induces optimal policy, but Q-learning cannot converge to this solution.

## 7    Conclusion

In this paper, we have studied PBE through the lens of SNRDD assumption and condition motivated from the AVI scheme. In this context, we also studied the relationship between the convergence of Q-learning and AVI. Moreover, to extend the understanding of solution to PBE, we provided examples showing pathological phenomena when using $\epsilon$-greedy policy. Future studies would include extending the analysis to non-linear function approximation case. We believe that our analysis can be naturally extended to this setting, following the approach in Xu and Gu (2020), which investigates the convergence of Q-learning with neural networks. Specifically, Xu and Gu (2020) considers a projection onto a linear subspace of the form $Q(\theta_0) + \nabla Q(\theta_0)^\top (\theta - \theta_0)$ where $Q(\cdot)$ is the function approximator and $\theta_0$ denotes the initialization point. Then, the analysis in Xu and Gu (2020) relies on Melo's condition from Melo et al. (2008), originally used to prove convergence under linear function approximation. By replacing Melo's condition with our SNRDD assumption, we believe that our theoretical results can similarly be extended to the non-linear setting.

Moreover, our theoretical findings also indicate where practical improvements may be pursued. Since AVI can be viewed as a primitive form of deep Q-networks (DQN), the fact that we identify regimes in which Q-learning converges while AVI fails highlights meaningful structural differences between the two. This observation encourages further investigation into Q-learning–type updates, consistent with recent efforts to revisit DQN without target networks—essentially reverting to a form of Q-learning (Gallici et al., 2025; Vasan et al., 2024).

In Section 6, we have provided examples using $\epsilon$-greedy policy showing difficulty on the extension of our analysis. Nonetheless, we believe that the boundedness of the iterate can be established under the SNRDD condition, following the approach in Gopalan and Thoppe (2023).

Lastly, upon assuming the existence of the fixed point, we believe that our convergence analysis can be relaxed to local points by replacing the set of differentiable points $\mathcal{D}$ with a neighbourhood around the fixed point.

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

## 8 NOTATIONS AND ORGANIZATION

**Notations:** $\mathbb{R}$ : set of real numbers; $\mathbb{R}_+$: set of non-negative real numbers; $\mathbb{R}^d$ : set of real-valued $d$-dimensional vectors, $\mathbb{R}^{m \times n}$ : set of real-valued $m \times n$-dimensional matrices; $\mathbb{C}$ : set of complex numbers ; $[n]$ for $n \in \mathbb{N}$ : $\{1, 2, \ldots, n\}$; $[\boldsymbol{v}]_i$ for $\boldsymbol{v} \in \mathbb{R}^n$ and $i \in [n]$ : $i$-th element of $\boldsymbol{v}$; $[\boldsymbol{A}]_{i,j}$ for $\boldsymbol{A} \in \mathbb{R}^{n \times m}$: the element in the $i$-th row and $j$-th column of the matrix $\boldsymbol{A}$; $\|\boldsymbol{A}\|_\infty$ for $\boldsymbol{A} \in \mathbb{R}^{m \times n}$: infinity norm of matrix $\boldsymbol{A}$, i.e., $\max_{1 \le i \le m} \sum_{j=1}^n |[\boldsymbol{A}]_{i,j}|$; $\Delta^{\mathcal{D}}$ for some set $\mathcal{D}$ : a probability distribution over the set $\mathcal{D}$; $\lambda_{\min}(\boldsymbol{A})$ for $\boldsymbol{A} \in \mathbb{R}^{n \times n}$ : minimum eigenvalue of $\boldsymbol{A}$; $\|\boldsymbol{x}\|_{\boldsymbol{A}}$ for $\boldsymbol{x} \in \mathbb{R}^p$ and a positive semi-definite matrix $\boldsymbol{A} \in \mathbb{R}^{p \times p}$ : $\sqrt{\boldsymbol{x}^\top \boldsymbol{A} \boldsymbol{x}}$;

**Organization:** In Section 3, conditions for existence of a solution to PBE is discussed. Section 4 and 5 provides convergence result for Q-learning and AVI, respectively. Lastly in Section 6, we discuss the properties of the solution to PBE when an $\epsilon$-greedy policy is adopted.

The Appendix Section 9 reviews the fixed point theory and provides foundational result for studying the solution of PBE. In Appendix Section 10, additional technical details are provided. A brief review on stochastic approximation is given in Appendix Section 11. The proofs omitted from the main manuscript are given in Appendix Section 12. The MDP examples used in the main manuscript are provided in Appendix Section 13 and 14. The omitted related works and the pseudo-codes in the main manuscript are provided in Appendix Section 15 and 17, respectively.

## 9 FIXED POINT PROBLEM

In this section, we present an analysis of existence and uniqueness of the solution to a specific equation. The results will be applied to the study of the solution to the projected Bellman equation (PBE) in Section 3. Our goal is to solve the following equation:

$$h(\boldsymbol{x}) := \boldsymbol{A}_{\boldsymbol{x}} \boldsymbol{x} + \boldsymbol{b}_{\boldsymbol{x}} = \boldsymbol{0}, \tag{12}$$

where $\boldsymbol{A}_{\boldsymbol{x}} \in \mathbb{R}^{p \times p}$ and $\boldsymbol{b}_{\boldsymbol{x}} \in \mathbb{R}^p$ are a matrix and a vector that depend on $\boldsymbol{x} \in \mathbb{R}^p$, respectively. When there are only finitely many possible choices of $\boldsymbol{A}_{\boldsymbol{x}}$ and $\boldsymbol{b}_{\boldsymbol{x}}$, it is called a switched affine system or a piecewise affine system.

Finding a solution of (12) can be re-casted into a fixed point problem: $\boldsymbol{x} + \alpha h(\boldsymbol{x}) = \boldsymbol{x}$ for some $\alpha \in \mathbb{R}$. The study of fixed-point problems has been extensively explored in the literature, with foundational contributions from pioneering works such as Brouwer (1911) and Banach (1922).

**Lemma 9.1** (Brouwer's fixed point theorem (Brouwer, 1911)). *Let $\mathcal{B}_R := \{\boldsymbol{x} \in \mathbb{R}^n : \|\boldsymbol{x}\| < R\}$ be an open ball in $\mathbb{R}^n$ centered at the origin and of radius $R$ with some norm $\|\cdot\|$. If $\boldsymbol{f} : \mathcal{B}_R \to \mathcal{B}_R$ is a continuous function, then, $\boldsymbol{f}$ has a fixed point, i.e., a solution of $\boldsymbol{f}(\boldsymbol{x}) = \boldsymbol{x}$.*

**Lemma 9.2** (Banach fixed point theorem (Banach, 1922)). *Consider a mapping $\boldsymbol{f} : \mathbb{R}^n \to \mathbb{R}^n$. Suppose there exists a norm $\|\cdot\|$ such that $\|\boldsymbol{f}(\boldsymbol{x}) - \boldsymbol{f}(\boldsymbol{y})\| < C \|\boldsymbol{x} - \boldsymbol{y}\|$ where $C \in (0, 1)$. Then, there exists a unique point $\boldsymbol{x}^* \in \mathbb{R}^n$ such that $\boldsymbol{f}(\boldsymbol{x}^*) = \boldsymbol{x}^*$.*

A common method for verifying the existence of a fixed point is to check if the matrix $\boldsymbol{A}_{\boldsymbol{x}}$ satisfies a specific condition. We focus on a matrix with a strictly negatively row-dominant diagonal (SNRDD) introduced in Definition 3.1:

$$\sup_{\boldsymbol{x} \in \mathcal{D}} \max_{i \in [p]} S_i(\boldsymbol{A}_{\boldsymbol{x}}) < 0, \tag{13}$$

where $S_i(\boldsymbol{A}_{\boldsymbol{x}})$ is defined in Definition 3.1, and $\mathcal{D} \subset \mathbb{R}^p$ will be formally defined later. This concept has been widely used in the literature of fixed point problem as well as in various system analyses (Molchanov and Pyatnitskiy, 1989; Davydov et al., 2024a).

Now, let us present a simple result that follows from standard argument of Brouwer's fixed point theorem given in Lemma 9.1 in Appendix 10:

**Lemma 9.3.** *Suppose (13) holds with $\mathcal{D} = \mathbb{R}^p$, and $\sup_{\boldsymbol{x} \in \mathbb{R}^p} \|\boldsymbol{b}_{\boldsymbol{x}}\|_\infty < \infty$. Furthermore, if the function $h$ in (12) is continuous, then a solution of (12) exists.*

*Proof.* For simplicity of the proof, denote $c := \sup_{\boldsymbol{x} \in \mathbb{R}^p} \max_{i \in [p]} S_i(\boldsymbol{A}_{\boldsymbol{x}})$, which is a negative constant. Consider the following map : $\bar{h}(\boldsymbol{x}) = \boldsymbol{x} + \alpha h(\boldsymbol{x})$ for small enough $\alpha$ such that $0 < \alpha <$

$\frac{1}{\sup_{\boldsymbol{x}\in\mathbb{R}^p}\|\boldsymbol{A_x}\|_\infty}$. Then, we have,

$$\|\boldsymbol{I} + \alpha\boldsymbol{A_x}\|_\infty = \max_{1\le i\le p}|1 + \alpha[\boldsymbol{A_x}]_{i,i}| + \alpha\sum_{j\in\{1,2,\dots,p\}\setminus\{i\}}|[\boldsymbol{A_x}]_{i,j}|$$

$$= \max_{1\le i\le p}1 + \alpha\left([\boldsymbol{A_x}]_{i,i} + \sum_{j\in\{1,2,\dots,p\}\setminus\{i\}}|[\boldsymbol{A_x}]_{i,j}|\right)$$

$$\le 1 + c\alpha.$$

The second equality follows from the fact that $0 < 1 + \alpha[\boldsymbol{A_x}]_{i,i}$ from the choice of $\alpha$. The last inequality follows from the definition of $S_i(\boldsymbol{A_x})$ in Definition 3.1 and denoting $c := \sup_{\boldsymbol{x}\in\mathbb{R}^p}\max_{i\in[p]}S_i(\boldsymbol{A_x})$.

Then, for $\boldsymbol{x}$ such that $\|\boldsymbol{x}\|_\infty \le \frac{\sup_{\boldsymbol{x}\in\mathbb{R}^p}\|\boldsymbol{b_x}\|_\infty}{|c|}$,

$$\left\|\bar{\boldsymbol{h}}(\boldsymbol{x})\right\|_\infty \le \|(\boldsymbol{I} + \alpha\boldsymbol{A_x})\boldsymbol{x}\|_\infty + \alpha\|\boldsymbol{b_x}\|_\infty$$

$$\le (1 + c\alpha)\|\boldsymbol{x}\|_\infty + \alpha\|\boldsymbol{b_x}\|_\infty$$

$$\le \frac{\sup_{\boldsymbol{x}\in\mathbb{R}^p}\|\boldsymbol{b_x}\|_\infty}{|c|}.$$

Therefore, $\bar{\boldsymbol{h}}$ is a self-map, i.e, the set $\left\{\boldsymbol{x} : \|\boldsymbol{x}\|_\infty \le \frac{\sup_{\boldsymbol{x}\in\mathbb{R}^p}\|\boldsymbol{b_x}\|_\infty}{|c|}\right\}$ is mapped into itself by $\bar{\boldsymbol{h}}$. Moreover, $\bar{\boldsymbol{h}}$ is a continuous function from the assumption that $\boldsymbol{h}$ is a continuous function. Now, we can apply Brouwer's fixed point theorem in Lemma 9.1 in the Appendix and the existence of a fixed point of the map $\bar{\boldsymbol{h}}$ follows. □

**Remark 9.4.** *Without continuity, characterizing the existence of a solution becomes challenging. An example where the condition (13) is satisfied, yet no solution exists or multiple solutions arise, is provided in Section 6.*

If we assume a slightly more contingent assumption that $\boldsymbol{f}(\boldsymbol{x}) = \boldsymbol{A_x}\boldsymbol{x}$ in (12) is a locally Lipschitz map (where the definition is given in Definition 10.1 in Appendix 10), then we can guarantee a stronger result, i.e., the uniqueness of the solution(which can be also viewed as a result from Davydov et al. (2024b) satisfying one-sided Lipschitz condition), and can be derived using a version of Lebourg's mean value theorem as in Theorem 17 in Davydov et al. (2024b):

**Lemma 9.5.** *Suppose $\|\boldsymbol{b_x} - \boldsymbol{b_{x'}}\|_\infty \le l\|\boldsymbol{x} - \boldsymbol{x'}\|_\infty$ where $l < |\sup_{\boldsymbol{x}\in\mathcal{D_f}}\max_{i\in[p]}S_i(\boldsymbol{A_x})|$ and $\boldsymbol{f}(\boldsymbol{x}) = \boldsymbol{A_x}\boldsymbol{x}$ and $\mathcal{D_f}$ is the set of differentiable points of $\boldsymbol{f}$. Moreover, suppose $\boldsymbol{f}$ is a locally Lipschitz mapping, with condition (13) holding with $\mathcal{D_f}$. Then, there exists a unique point $\boldsymbol{x}^* \in \mathbb{R}^p$ such that $\boldsymbol{A_{x^*}}\boldsymbol{x}^* + \boldsymbol{b_{x^*}} = \boldsymbol{0}_p$ where $\boldsymbol{0}_p$ is a zero vector in $\mathbb{R}^p$.*

*Proof.* Let $\bar{\boldsymbol{h}}(\boldsymbol{x}) = \boldsymbol{x} + \alpha(\boldsymbol{A_x} + \boldsymbol{b_x})$ and denote $c = \sup_{\boldsymbol{x}\in\mathcal{D_f}}\max_{i\in[p]}S_i(\boldsymbol{A_x})$ which is a negative constant. Then, we have

$$\left\|\bar{\boldsymbol{h}}(\boldsymbol{x}) - \bar{\boldsymbol{h}}(\boldsymbol{y})\right\|_\infty = \|\boldsymbol{x} - \boldsymbol{y} + \alpha(\boldsymbol{A_x}\boldsymbol{x} - \boldsymbol{A_y}\boldsymbol{y}) + \alpha(\boldsymbol{b_x} - \boldsymbol{b_y})\|_\infty$$

$$\le (1 + \alpha c)\|\boldsymbol{x} - \boldsymbol{y}\|_\infty + \alpha\|\boldsymbol{b_x} - \boldsymbol{b_y}\|_\infty$$

$$\le (1 + \alpha c)\|\boldsymbol{x} - \boldsymbol{y}\|_\infty + \alpha l\|\boldsymbol{x} - \boldsymbol{y}\|_\infty$$

$$\le (1 + \alpha(c + l))\|\boldsymbol{x} - \boldsymbol{y}\|_\infty.$$

The second line follows from application of Lebourg's mean value theorem in Lemma 10.6 of the Appendix. As $l < |c|$ from the assumption, choosing $\alpha < \frac{1}{|c+l|}$, the proof is completed by the Banach fixed point theorem in Lemma 9.2. □

**Remark 9.6.** *The continuity of a function does not necessarily imply Lipschitzness. However, if a function is Lipschitz, it is necessarily continuous.*

**Remark 9.7.** *A locally Lipschitz function is differentiable almost everywhere by Rademacher's theorem in Lemma 10.2 in Appendix 10.*

Now, let us focus on a slightly different condition to study the solution of (12). For some $C_{\boldsymbol{x}} \in \mathbb{R}^{p \times p}$ that is invertible, we can re-write the equation in (12) by

$$\tilde{\boldsymbol{h}}(\boldsymbol{x}) := C_{\boldsymbol{x}}^{-1}(A_{\boldsymbol{x}} + C_{\boldsymbol{x}})\boldsymbol{x} + C_{\boldsymbol{x}}^{-1}\boldsymbol{b}_{\boldsymbol{x}} - \boldsymbol{x} = \boldsymbol{0}. \tag{14}$$

When $A_{\boldsymbol{x}}$, $\boldsymbol{b}_{\boldsymbol{x}}$, and $C_{\boldsymbol{x}}$ are constant matrices and vectors, i.e., $A_{\boldsymbol{x}} = A$, $\boldsymbol{b}_{\boldsymbol{x}} = \boldsymbol{b}$, and $C_{\boldsymbol{x}} = C$ for some $A, C \in \mathbb{R}^{p \times p}, \boldsymbol{b} \in \mathbb{R}^p$, the system simplifies to a linear form. In this scenario, the reformulation in (14) is widely recognized as matrix splitting (Berman and Plemmons, 1994), a method extensively studied for analyzing the convergence of linear systems such as

$$\boldsymbol{x}_{k+1} = C^{-1}(A + C)\boldsymbol{x}_k, \quad \boldsymbol{x}_0 \in \mathbb{R}^p.$$

The convergence of these systems is determined by the spectral radius of $C^{-1}A$. However, when these matrices depend on $\boldsymbol{x}$, the spectral radius of each $C_{\boldsymbol{x}}^{-1}A_{\boldsymbol{x}}$ becomes insufficient to ensure the existence of solutions or the stability of dynamical systems described by (14), specifically

$$\boldsymbol{x}_{k+1} = C_{\boldsymbol{x}_k}^{-1}(A_{\boldsymbol{x}_k} + C_{\boldsymbol{x}_k})\boldsymbol{x}_k + C_{\boldsymbol{x}_k}^{-1}\boldsymbol{b}_{\boldsymbol{x}_k}, \quad \boldsymbol{x}_0 \in \mathbb{R}^p.$$

Consequently, an alternative condition must be considered to address these challenges. In particular, we consider the following condition for some real number $c^*$:

$$\left\|C_{\boldsymbol{x}}^{-1}(A_{\boldsymbol{x}} + C_{\boldsymbol{x}})\right\|_\infty \leq c^* < 1, \quad \forall \boldsymbol{x} \in \mathcal{D} \tag{15}$$

for some set $\mathcal{D} \subset \mathbb{R}^p$, which will be clarified further. Now, we have the result for the existence of a solution to (12):

**Lemma 9.8.** *Suppose (15) holds with $\mathcal{D} = \mathbb{R}^p$, and $\sup_{\boldsymbol{x} \in \mathbb{R}^p} \left\|C_{\boldsymbol{x}}^{-1}\boldsymbol{b}_{\boldsymbol{x}}\right\|_\infty < \infty$. If $\tilde{\boldsymbol{h}}$ in (14) is a continuous function, then there exists a solution of (12).*

*Proof.* For simplicity of the notation, let us denote $c = 1 - \sup_{\boldsymbol{x} \in \mathbb{R}^p} \left\|C_{\boldsymbol{x}}^{-1}(A_{\boldsymbol{x}} + C_{\boldsymbol{x}})\right\|_\infty$, and let $\bar{\boldsymbol{h}}(\boldsymbol{x}) = \boldsymbol{x} + \alpha\tilde{\boldsymbol{h}}(\boldsymbol{x})$ where $0 < \alpha < \frac{1}{1 - \sup_{\boldsymbol{x} \in \mathbb{R}^p} \left\|C_{\boldsymbol{x}}^{-1}(A_{\boldsymbol{x}} + C_{\boldsymbol{x}})\right\|_\infty}$. Now, we have the following bound

$$\left\|(1 - \alpha)\boldsymbol{I} + \alpha C_{\boldsymbol{x}}^{-1}(A_{\boldsymbol{x}} + C_{\boldsymbol{x}})\right\|_\infty$$

$$= \max_{i \in [p]} \left| |(1 - \alpha) + \alpha[C_{\boldsymbol{x}}^{-1}(A_{\boldsymbol{x}} + C_{\boldsymbol{x}})]_{i,i}| + \alpha \sum_{j \neq i}[[C_{\boldsymbol{x}}^{-1}(A_{\boldsymbol{x}} + C_{\boldsymbol{x}})]_{ii}]_{i,j} \right|$$

$$\leq 1 - \alpha + \alpha \left\|C_{\boldsymbol{x}}^{-1}(A_{\boldsymbol{x}} + C_{\boldsymbol{x}})\right\|_\infty$$

$$< 1 - \alpha c \tag{16}$$

where the last two inequalities follows from the choice of $\alpha$.

Then, we have

$$\left\|\bar{\boldsymbol{h}}(\boldsymbol{x})\right\|_\infty = \left\|\left((1 - \alpha)\boldsymbol{I} + \alpha C_{\boldsymbol{x}}^{-1}(A_{\boldsymbol{x}} + C_{\boldsymbol{x}})\right)\boldsymbol{x} + \alpha C_{\boldsymbol{x}}^{-1}\boldsymbol{b}_{\boldsymbol{x}}\right\|_\infty$$

$$\leq \left\|(1 - \alpha)\boldsymbol{I} + \alpha C_{\boldsymbol{x}}^{-1}(A_{\boldsymbol{x}} + C_{\boldsymbol{x}})\right\|_\infty \left\|\boldsymbol{x}\right\|_\infty + \alpha \left\|C_{\boldsymbol{x}}^{-1}\boldsymbol{b}_{\boldsymbol{x}}\right\|_\infty$$

$$\leq (1 - \alpha c) \left\|\boldsymbol{x}\right\|_\infty + \alpha \left\|C_{\boldsymbol{x}}^{-1}\boldsymbol{b}_{\boldsymbol{x}}\right\|_\infty.$$

The last inequality follows from (16).

For $\boldsymbol{x}$ such that $\left\|\boldsymbol{x}\right\|_\infty \leq \frac{\sup_{\boldsymbol{x} \in \mathbb{R}^p} \left\|C_{\boldsymbol{x}}^{-1}\boldsymbol{b}_{\boldsymbol{x}}\right\|_\infty}{c}$, we have $\left\|\bar{\boldsymbol{h}}(\boldsymbol{x})\right\|_\infty \leq \frac{\sup_{\boldsymbol{x} \in \mathbb{R}^p} \left\|C_{\boldsymbol{x}}^{-1}\boldsymbol{b}_{\boldsymbol{x}}\right\|_\infty}{c}$. Therefore, $\bar{\boldsymbol{h}}$ is a self-map, and a continuous function from the assumption that $\tilde{\boldsymbol{h}}$ is a continuous function. Now, applying Brouwer's fixed point theorem in Lemma 9.1 in the Appendix, there exists a solution of (12). $\square$

The uniqueness of the solution can be guaranteed with additional assumption of local Lipshictzness of $\tilde{\boldsymbol{h}}$:

**Lemma 9.9.** *Suppose $C_{\boldsymbol{x}}$ and $\boldsymbol{b}_{\boldsymbol{x}}$ are bounded constant matrix and vector, respectively. If $\tilde{\boldsymbol{h}}$ is a locally Lipschitz function and $\sup_{\boldsymbol{x} \in \mathcal{D}_{\tilde{h}}} \left\|C_{\boldsymbol{x}}^{-1}(A_{\boldsymbol{x}} + C_{\boldsymbol{x}})\right\|_\infty < 1$ where $\mathcal{D}_{\tilde{h}}$ is the set of differentiable points of $\tilde{\boldsymbol{h}}$, then a solution of (12) exists and is unique.*

*Proof.* For simplicity, let us denote $c := \sup_{\boldsymbol{x} \in \mathcal{D}_{\tilde{h}}} \left\| \boldsymbol{C}_{\boldsymbol{x}}^{-1}(\boldsymbol{A}_{\boldsymbol{x}} + \boldsymbol{C}_{\boldsymbol{x}}) \right\|_{\infty} < 1$.

By a version of Lebourg's mean value theorem in Lemma 10.6 in Appendix Section 10.1, we have

$$\left\| \boldsymbol{C}_{\boldsymbol{x}}^{-1}(\boldsymbol{A}_{\boldsymbol{x}} + \boldsymbol{C}_{\boldsymbol{x}})^{-1}\boldsymbol{x} - \boldsymbol{C}_{\boldsymbol{y}}^{-1}(\boldsymbol{A}_{\boldsymbol{y}} + \boldsymbol{C}_{\boldsymbol{y}})^{-1}\boldsymbol{y} \right\|_{\infty} \leq c \left\| \boldsymbol{x} - \boldsymbol{y} \right\|_{\infty}.$$

The Banach fixed point theorem in Lemma 9.2 in the Appendix ensures the uniqueness and existence of the solution. $\square$

**Remark 9.10.** *Suppose there exists a norm $\|\cdot\|$ such that $\sup_{\boldsymbol{x} \in \mathbb{R}^p} \left\| \boldsymbol{C}_{\boldsymbol{x}}^{-1}(\boldsymbol{A}_{\boldsymbol{x}} + \boldsymbol{C}_{\boldsymbol{x}}) \right\| < 1$, which can be guaranteed if the joint spectral radius is smaller than one (Rota and Strang, 1960) and $\{\boldsymbol{C}_{\boldsymbol{x}}^{-1}(\boldsymbol{A}_{\boldsymbol{x}} + \boldsymbol{C}_{\boldsymbol{x}}) : \boldsymbol{x} \in \mathbb{R}^p\}$ is a finite set. Then, we can replace the infinity norm in (15) with this common norm. We refer to Definition 10.9 and Lemma 10.10 in the Appendix 10 for further details.*

### 9.1 DISCUSSION ON THE CONDITION (13) AND (15)

From the above discussion, we can see that the conditions (13) and (15) are important in guaranteeing the existence of a fixed point. Let us discuss the relationship between the two conditions:

**Proposition 9.11.** *Suppose for all $\boldsymbol{x} \in \mathcal{D}$, $\boldsymbol{C}_{\boldsymbol{x}}$ is a diagonal matrix such that $0 < \sigma_{\min} < \lambda_{\min}(\boldsymbol{C}_{\boldsymbol{x}})$ and $\lambda_{\max}(\boldsymbol{C}_{\boldsymbol{x}}) < \sigma_{\max} < \infty$ for some positive constants $\sigma_{\min}$ and $\sigma_{\max}$.*

*1. Suppose (13) holds, and $\boldsymbol{A}_{\boldsymbol{x}} + \boldsymbol{C}_{\boldsymbol{x}}$ has non-negative diagonal elements for all $\boldsymbol{x} \in \mathcal{D}$, then, (15) holds.*

*2. If (15) holds, then (13) holds.*

*Proof.* From (13), for some $\kappa > 0$, we have the following:

$$
\begin{aligned}
-\kappa > &[\boldsymbol{A}_{\boldsymbol{x}}]_{i,i} + \sum_{j \in [p] \setminus \{i\}} |[\boldsymbol{A}_{\boldsymbol{x}}]_{i,j}| \\
= &- [\boldsymbol{C}_{\boldsymbol{x}}]_{i,i} + [\boldsymbol{A}_{\boldsymbol{x}} + \boldsymbol{C}_{\boldsymbol{x}}]_{i,i} + \sum_{j \in [p] \setminus \{i\}} |[\boldsymbol{A}_{\boldsymbol{x}} + \boldsymbol{C}_{\boldsymbol{x}}]_{i,j}| \\
= &[\boldsymbol{C}_{\boldsymbol{x}}]_{i,i} \left( -1 + [\boldsymbol{C}_{\boldsymbol{x}}]_{i,i}^{-1} \sum_{j=1}^{p} |[\boldsymbol{A}_{\boldsymbol{x}} + \boldsymbol{C}_{\boldsymbol{x}}]_{i,j}| \right).
\end{aligned}
$$

The first equality follows since $\boldsymbol{C}_{\boldsymbol{x}}$ is a diagonal matrix. The last equality follows from the fact that the diagonal elements for $\boldsymbol{A}_{\boldsymbol{x}} + \boldsymbol{C}_{\boldsymbol{x}}$ are non-negative. As $[\boldsymbol{C}_{\boldsymbol{x}}]_{i,i} > 0$, we have the following result:

$$0 < \frac{\kappa}{\sigma_{\max}} \leq \sup_{\boldsymbol{x} \in \mathcal{D}} \left( 1 - \max_{i \in [p]} [\boldsymbol{C}_{\boldsymbol{x}}]_{i,i}^{-1} \sum_{j=1}^{p} |[\boldsymbol{A}_{\boldsymbol{x}} + \boldsymbol{C}_{\boldsymbol{x}}]_{i,j}| \right) = 1 - \sup_{\boldsymbol{x} \in \mathcal{D}} \left\| \boldsymbol{C}_{\boldsymbol{x}}^{-1}(\boldsymbol{A}_{\boldsymbol{x}} + \boldsymbol{C}_{\boldsymbol{x}}) \right\|_{\infty}.$$

This proves the first statement.

Now, let us prove the second statement. Note that from the condition (15), for some $\omega > 0$,

$$
\begin{aligned}
-\omega > &\max_{i \in [p]} \frac{\sum_{j=1}^{p} |[\boldsymbol{A}_{\boldsymbol{x}} + \boldsymbol{C}_{\boldsymbol{x}}]_{i,j}|}{[\boldsymbol{C}_{\boldsymbol{x}}]_{i,i}} - 1 \\
= &\max_{i \in [p]} \frac{1}{[\boldsymbol{C}_{\boldsymbol{x}}]_{i,i}} \left( \sum_{j=1}^{p} |[\boldsymbol{A}_{\boldsymbol{x}} + \boldsymbol{C}_{\boldsymbol{x}}]_{i,j}| - [\boldsymbol{C}_{\boldsymbol{x}}]_{i,i} \right) \\
\geq &\max_{i \in [p]} \frac{1}{\sigma_{\min}} \left( \sum_{j=1}^{p} |[\boldsymbol{A}_{\boldsymbol{x}} + \boldsymbol{C}_{\boldsymbol{x}}]_{i,j}| - [\boldsymbol{C}_{\boldsymbol{x}}]_{i,i} \right).
\end{aligned}
$$

The first inequality follows from the assumption in (15), and the last inequality follows from the condition $\max_{i \in [p]} [\boldsymbol{C}_{\boldsymbol{x}}]_{i,i} > \sigma_{\min}$.

Now, we can check that

$$-\sigma_{\min}\omega > \sum_{j=1}^{p} |[\boldsymbol{A_x} + \boldsymbol{C_x}]_{i,j}| - [\boldsymbol{C_x}]_{i,i} \geq \sum_{j\in[p]\setminus\{i\}} |[\boldsymbol{A_x}]_{i,j}| + [\boldsymbol{A_x}]_{i,i}.$$

where the last inequality follows from the fact that $\boldsymbol{C_x}$ is a diagonal matrix.

Therefore, from the definition of $S_i(\cdot)$ in Definition 3.1, taking supremum and maximum over the above inequality, we get

$$\sup_{\boldsymbol{x}\in\mathcal{D}} \max_{i\in[p]} S_i(\boldsymbol{A_x}) \leq -\sigma_{\min}w < 0,$$

which proves the second statement.

$\square$

## 10 Auxiliary Details

### 10.1 Fine properties of function

**Definition 10.1** (Locally Lipschitz function (Clarke, 1981)). *A function $\boldsymbol{f} : \mathbb{R}^n \to \mathbb{R}^n$ is said to be locally Lipschitz, if for $\boldsymbol{x} \in \mathbb{R}^n$, there exists a constant $L$ an $\delta$ such that*

$$\|\boldsymbol{x} - \boldsymbol{x}_0\| < \delta \Rightarrow \|\boldsymbol{f}(\boldsymbol{x}) - \boldsymbol{f}(\boldsymbol{x}_0)\| \leq L \|\boldsymbol{x} - \boldsymbol{x}_0\|$$

**Lemma 10.2** (Rademacher's theorem, page 810 in Evans (2018)). *Let $\boldsymbol{f} : \mathbb{R}^n \to \mathbb{R}^n$. If $\boldsymbol{f}$ is a locally Lispcthiz function, then $\boldsymbol{f}$ is differentiable almost everywhere.*

**Definition 10.3** (Generalized directional derivative (Clarke, 1981)). *The generalized directional derivative of the locally Lipschitz function $f : \mathbb{R}^n \to \mathbb{R}$ at the point $\boldsymbol{u} \in \mathbb{R}^n$ in the direction $\boldsymbol{v} \in \mathbb{R}^n$ is defined by*

$$f^o(\boldsymbol{u};\boldsymbol{v}) = \limsup_{\boldsymbol{w}\to\boldsymbol{u}, t\to 0+} \frac{f(\boldsymbol{w} + t\boldsymbol{v}) - f(\boldsymbol{w})}{t}.$$

**Definition 10.4** (Clarke subdifferential, page 54 in Clarke (1981)). *Let $f : \mathbb{R}^n \to \mathbb{R}$ be a locally Lipschitz function. The Clarke subdifferential $\partial_C f(\boldsymbol{u})$ of $f$ at a point $\boldsymbol{u} \in \mathbb{R}^n$ is defined as the following:*

$$\partial_C f(\boldsymbol{u}) = \left\{ \boldsymbol{v} \in \mathbb{R}^n : \boldsymbol{v}^\top \boldsymbol{y} \leq f^o(\boldsymbol{u};\boldsymbol{v}), \ \forall \boldsymbol{y} \in \mathbb{R}^n \right\}.$$

*When $f$ is locally Lipschitz, then*

$$\partial_C f(\boldsymbol{u}) = \mathrm{conv}\left\{ \lim_{i\to\infty} \nabla f(\boldsymbol{x}_i) : \boldsymbol{x}_i \in \mathcal{D}_f \text{ such that } \boldsymbol{x}_i \to \boldsymbol{v} \text{ and } \lim_{i\to\infty} \nabla f(\boldsymbol{x}_i) \text{ exists} \right\}$$

*where $\mathrm{conv}(A)$ denotes convex hull of a set $A$, and $\mathcal{D}_f$ is the differentiable points of $f$ and $\boldsymbol{x}_i$ is a converging sequence to $\boldsymbol{v}$.*

**Lemma 10.5** (Lebourg's mean value theorem, Theorem 2.4 in Clarke et al. (2008)). *Consider a locally Lipschitz function $f : \mathbb{R}^n \to \mathbb{R}$. For $\boldsymbol{x}, \boldsymbol{y} \in \mathbb{R}^n$, there exists $\boldsymbol{v} \in \{t\boldsymbol{x} + (1-t)\boldsymbol{y} : t \in [0,1]\}$ such that*

$$f(\boldsymbol{x}) - f(\boldsymbol{x}) = \boldsymbol{z}^\top (\boldsymbol{x} - \boldsymbol{y})$$

*where*

$$\boldsymbol{z} \in \partial f_C(\boldsymbol{v}) = \mathrm{conv}\left\{ \lim_{i\to\infty} \nabla f(\boldsymbol{x}_i) : \boldsymbol{x}_i \in \mathcal{D}_f \text{ such that } \boldsymbol{x}_i \to \boldsymbol{v} \text{ and } \lim_{i\to\infty} \nabla f(\boldsymbol{x}_i) \text{ exists} \right\}.$$

*$\mathcal{D}_f$ is the differentiable points of $f$ and $\{\boldsymbol{x}_i \in \mathcal{D}_f\}_{i=1}^\infty$ is a converging sequence to $\boldsymbol{v}$.*

**Lemma 10.6.** *Suppose $\boldsymbol{f} : \mathbb{R}^n \to \mathbb{R}^n$ is a locally Lipscthiz function and $\|\nabla \boldsymbol{f}(\boldsymbol{x})\|_\infty \leq f_{\max}$ for all the differentiable points $\boldsymbol{x}$ for some positive real number $f_{\max}$. Then, the following holds:*

$$\|\boldsymbol{f}(\boldsymbol{x}) - \boldsymbol{f}(\boldsymbol{y})\|_\infty \leq f_{\max} \|\boldsymbol{x} - \boldsymbol{y}\|_\infty.$$

*Proof.* Consider $\boldsymbol{e}_i^\top \boldsymbol{f}(\boldsymbol{x})$ for some $i \in [n]$. By Lebourg's mean value theorem in Lemma 10.5, we have, for a basis vector in $\mathbb{R}^n$ whose $i$-th coordinate is one,

$$\boldsymbol{e}_i^\top (\boldsymbol{f}(\boldsymbol{x}) - \boldsymbol{f}(\boldsymbol{y})) = \boldsymbol{a}_i^\top (\boldsymbol{x} - \boldsymbol{y}) \tag{17}$$

for $\boldsymbol{a}_i \in \mathrm{conv}\{\lim_{k \to \infty} \nabla \boldsymbol{f}(\boldsymbol{x}_k)^\top \boldsymbol{e}_i : \boldsymbol{x}_k \to \boldsymbol{v}, \boldsymbol{x}_k \in \mathcal{D}_{\boldsymbol{f}}\}$ where $\boldsymbol{v} \in \{t\boldsymbol{x} + (1-t)\boldsymbol{y} : t \in [0,1]\}$. We can find such $\boldsymbol{a}_i$ for all $i \in [n]$, and we have

$$\boldsymbol{f}(\boldsymbol{x}) - \boldsymbol{f}(\boldsymbol{y}) = \begin{bmatrix} \boldsymbol{a}_1^\top \\ \vdots \\ \boldsymbol{a}_n^\top \end{bmatrix} (\boldsymbol{x} - \boldsymbol{y}).$$

Taking the infinity norm on both sides, we get

$$\|\boldsymbol{f}(\boldsymbol{x}) - \boldsymbol{f}(\boldsymbol{y})\|_\infty \leq \max_{i \in [n]} \left\|\boldsymbol{a}_i^\top\right\|_\infty \|\boldsymbol{x} - \boldsymbol{y}\|_\infty$$

$$\leq \left\|\sum_{j=1}^q \lambda_j \hat{\boldsymbol{f}}_j\right\|_\infty \|\boldsymbol{x} - \boldsymbol{y}\|_\infty \tag{18}$$

where $\sum_{j=1}^q \lambda_j = 1$. For $j \in [q]$, $\lambda_j \geq 0$, and $\hat{\boldsymbol{f}}_j = \lim_{k \to \infty} \boldsymbol{e}_i^\top \nabla \boldsymbol{f}(\boldsymbol{x}_k^j)$ for some sequence $\{\boldsymbol{x}_k^j \in \mathcal{D}_f\}_{k=1}^\infty$. Note that we have

$$\left\|\sum_{j=1}^q \lambda_j \hat{\boldsymbol{f}}_j\right\|_\infty = \lim_{k \to \infty} \left\|\sum_{j=1}^q \boldsymbol{e}_i^\top \nabla \boldsymbol{f}(\boldsymbol{x}_k^j)\right\|_\infty$$

$$\leq \lim_{k \to \infty} \sum_{j=1}^n \lambda_j \left\|\nabla \boldsymbol{f}(\boldsymbol{x}_k^j)\right\|_\infty$$

$$\leq \sum_{j=1}^n \lambda_j f_{\max}$$

$$= f_{\max}.$$

Applying this result to (18) yields the desired result. $\qquad\square$

**Lemma 10.7.** *A function $\boldsymbol{f} : \mathbb{R}^n \to \mathbb{R}^n$ is locally Lipschitz if and only if $\boldsymbol{f}$ is Lipschitz on every compact subset $K \subset \mathbb{R}^n$.*

*Proof.* The necessity part is an immediate consequence of the definition of local Lipschitz continuity. We now establish the converse implication. Suppose the local Lipschitzness holds with some norm $\|\cdot\|$ and $\boldsymbol{f}$ is not Lipschitz on some compact set $K$. Then, there exists some $\boldsymbol{x}, \boldsymbol{y} \in K$ such that $\frac{\|\boldsymbol{f}(\boldsymbol{x}) - \boldsymbol{f}(\boldsymbol{y})\|}{\|\boldsymbol{x} - \boldsymbol{y}\|} > C$ for any $C \geq 0$. Therefore, there exist a sequence $\{(\boldsymbol{x}_n, \boldsymbol{y}_n) \in K \times K\}_{n=1}^\infty$ such that $\frac{\|\boldsymbol{f}(\boldsymbol{x}_n) - \boldsymbol{f}(\boldsymbol{y}_n)\|}{\|\boldsymbol{x}_n - \boldsymbol{y}_n\|} \to \infty$. From the compactness of $K$, there exist a convergent subsequence $\{(\boldsymbol{x}_{k_n}, \boldsymbol{y}_{k_n})\}_{n=1}^\infty$ converging to $(\tilde{\boldsymbol{x}}, \tilde{\boldsymbol{y}})$. Moreover, as continuous function is bounded on compact set, we should have $\|\boldsymbol{x}_{k_n} - \boldsymbol{y}_{k_n}\| \to 0$. This contradicts the fact that $\boldsymbol{f}$ is locally Lipschitz at $\tilde{\boldsymbol{x}}$, and this proves the reverse direction. $\qquad\square$

**Lemma 10.8** (Lemma 7 in Davydov et al. (2024a)). *Suppose the map $f$ is locally Lipschitz. Then, the following two conditions are equivalent*

$$\max_{i \in [p]} S_i(\nabla f(\boldsymbol{x})) \leq -c \text{ for } \boldsymbol{x} \in \mathcal{D}_f \iff f \text{ is one-sided Lipschitz with constant } -c \text{ in } \mathcal{D}_f$$

*where $\mathcal{D}_f$ is the set of differentiable points of $f$ and $S_i(\cdot)$ is defind in Definition 3.1.*

By Rademacher theorem, a Lipschitz continuous function is differentiable almost everywhere.

## 10.2 MATRIX PROPERTIES

Now, let us briefly explain the concept of joint spectral radius (Rota and Strang, 1960), which is defined as follows:

**Definition 10.9** (Joint spectral radius (Rota and Strang, 1960))**.**

Given a set of matrix $\{\boldsymbol{A}_i \in \mathbb{R}^{n \times n}\}_{i=1}^m$, the joint spectral radius is defined as

$$\rho(\boldsymbol{A}_1, \cdots, \boldsymbol{A}_m) = \lim_{k \to \infty} \max_{\sigma \in \{1,2,\ldots,m\}^k} \|\boldsymbol{A}_{\sigma_k} \cdots \boldsymbol{A}_{\sigma_2} \boldsymbol{A}_{\sigma_1}\|^{1/k}.$$

**Lemma 10.10** (Rota and Strang (1960))**.** *Given a set of matrix $\{\boldsymbol{A}_i \in \mathbb{R}^{n \times n}\}_{i=1}^m$, if the joint spectral radius is smaller than one, then there exists a norm $\|\cdot\|$ such that $\|\boldsymbol{A}_i\| < 1$ for all $i \in [m]$.*

**Lemma 10.11** (Gerschgorin circle theorem (Horn and Johnson, 2012))**.** *Let $\boldsymbol{A} \in \mathbb{R}^{n \times n}$ and $R_i(\boldsymbol{A}) = \sum\limits_{j \in [n] \setminus \{i\}} [\boldsymbol{A}]_{i,j}$. Consider the Gerschgorin circles*

$$\{z \in \mathbb{C}| : |z - [\boldsymbol{A}]_{i,i}| \le R_i(\boldsymbol{A})\}, \quad i = 1, \ldots, n.$$

*The eigenvalues of $\boldsymbol{A}$ are in the union of Gerschgorin discs*

$$G(\boldsymbol{A}) = \cup_{i=1}^n \{z \in \mathbb{C} : |z - [\boldsymbol{A}]_{i,i}| \le R_i(\boldsymbol{A})\}.$$

**Definition 10.12** (Hurwtiz matrix (Khalil and Grizzle, 2002))**.** *A matrix $\boldsymbol{A} \in \mathbb{R}^{n \times n}$ is said to be a Hurwitz matrix if all of its eigenvalues has negative real part.*

**Lemma 10.13.** *An SNRDD matrix is a Hurwitz matrix.*

*Proof.* The proof directly follows from Gerschgorin circle theorem in Lemma 10.11. □

## 10.3 TYPES OF POLICIES AND MARKOV CHAIN

$\epsilon$**-greedy policy:** Let $\mathcal{A}^* = arg\max_{a \in \mathcal{A}} \boldsymbol{\phi}(s,a)^\top \boldsymbol{\theta}$.

$$\pi_{\boldsymbol{\theta}}^\epsilon(a \mid s) = \begin{cases} \frac{1}{|\mathcal{A}^*|} - \frac{\epsilon}{|\mathcal{A}^*|} & \text{if} \quad a \in \mathcal{A}^* \\ \frac{\epsilon}{|\mathcal{A}| - |\mathcal{A}^*|} & \text{if} \quad a \notin \mathcal{A}^* \end{cases}$$

$\epsilon$**-softmax policy** : Given a positive real number, $\tau$, which is so-called a temperature parameter, the $\epsilon$-softmax policy policy is defined as

$$\pi_{\boldsymbol{\theta}}(a \mid s) = \frac{\exp(\tau \boldsymbol{\phi}(s,a)^\top \boldsymbol{\theta})}{\sum_{u \in \mathcal{A}} \exp(\tau \boldsymbol{\phi}(s,u)^\top \boldsymbol{\theta})}, \quad \forall (s,a) \in \mathcal{S} \times \mathcal{A}$$

**Tamed Gibbs Policy (Meyn, 2024)** A $(\epsilon, \kappa_0)$-tamed Gibbs policy defined as $\pi_{\boldsymbol{\theta}}(a \mid s) = \frac{\exp(-\tau_{\boldsymbol{\theta}} \boldsymbol{\phi}(s,a)^\top \boldsymbol{\theta})}{\sum_{u \in \mathcal{A}} \exp(-\tau_{\boldsymbol{\theta}} \boldsymbol{\phi}(s,u)^\top \boldsymbol{\theta})}$ where

$$\tau_{\boldsymbol{\theta}}(a \mid s) = \begin{cases} \frac{\kappa_0}{\|\boldsymbol{\theta}\|_2} & \|\boldsymbol{\theta}\|_2 \ge 1 \\ \frac{\kappa_0}{2} & \text{else} \end{cases}.$$

The following lemma is from Perkins and Precup (2002):

**Lemma 10.14.** *If the behavior policy $\beta_{\boldsymbol{\theta}}$ satisfying Assumption 2.1 is locally Lipschitz, then, its corresponding stationary distribution $\mu_{\beta_{\boldsymbol{\theta}}}$ is also locally Lipschitz.*

*Proof.* For simplicity of the proof, let $\boldsymbol{P}_{\beta_{\boldsymbol{\theta}}} \in \mathbb{R}^{|\mathcal{S}||\mathcal{A}| \times |\mathcal{S}||\mathcal{A}|}$ and $\boldsymbol{\mu}_{\beta_{\boldsymbol{\theta}}} \in \mathbb{R}^{|\mathcal{S}||\mathcal{A}|}$ such that

$$[\boldsymbol{P}_{\beta_{\boldsymbol{\theta}}}]_{(s-1)|\mathcal{A}|+a, (x-1)|\mathcal{A}|+u} = \mathcal{P}(x \mid s, a)\beta_{\boldsymbol{\theta}}(u \mid x), \quad [\boldsymbol{\mu}_{\beta_{\boldsymbol{\theta}}}]_{(s-1)|\mathcal{A}|+a} = \mu_{\beta_{\boldsymbol{\theta}}}(a \mid s).$$

From local lipschitszness of $\beta_{\boldsymbol{\theta}}$, there exists $\delta$ and $L$ such that for $\boldsymbol{\theta}'$ satisfying $|\beta_{\boldsymbol{\theta}}(a \mid s) - \beta_{\boldsymbol{\theta}'}(a \mid s)|$ then, $|| \le L \|\boldsymbol{\theta} - \boldsymbol{\theta}'\|$ for some norm $\|\cdot\|$.

Now, note that the following holds (Seneta, 1993):

$$\boldsymbol{\mu}_{\beta_{\boldsymbol{\theta}'}}^\top - \boldsymbol{\mu}_{\beta_{\boldsymbol{\theta}}}^\top = \boldsymbol{\mu}_{\beta_{\boldsymbol{\theta}'}}^\top (\boldsymbol{P}_{\beta_{\boldsymbol{\theta}'}} - \boldsymbol{P}_{\beta_{\boldsymbol{\theta}}})(\boldsymbol{I} - \boldsymbol{P}_{\beta_{\boldsymbol{\theta}}} + \mathbf{1}\boldsymbol{\mu}_{\beta_{\boldsymbol{\theta}}}^\top)^{-1}.$$

Therefore, taking norm on each sides,

$$\left\|\boldsymbol{\mu}_{\beta_{\boldsymbol{\theta}'}}^\top - \boldsymbol{\mu}_{\beta_{\boldsymbol{\theta}}}^\top\right\|_1 \leq \left\|(\boldsymbol{I} - \boldsymbol{P}_{\beta_{\boldsymbol{\theta}}} + \mathbf{1}\boldsymbol{\mu}_{\beta_{\boldsymbol{\theta}}}^\top)^{-1}\right\|_1 \left\|\boldsymbol{P}_{\beta_{\boldsymbol{\theta}}} - \boldsymbol{P}_{\beta_{\boldsymbol{\theta}'}}\right\|_1$$

$$= \left\|\sum_{k=0}^\infty (\boldsymbol{P}_{\beta_{\boldsymbol{\theta}}} - \mathbf{1}\boldsymbol{\mu}_{\beta_{\boldsymbol{\theta}}}^\top)^k\right\|_1 \left\|\boldsymbol{P}_{\beta_{\boldsymbol{\theta}}} - \boldsymbol{P}_{\beta_{\boldsymbol{\theta}'}}\right\|_1$$

$$\leq C_{\mu_{\beta_{\boldsymbol{\theta}'}}} \left\|\sum_{k=0}^\infty (\boldsymbol{P}_{\beta_{\boldsymbol{\theta}}} - \mathbf{1}\boldsymbol{\mu}_{\beta_{\boldsymbol{\theta}}}^\top)^k\right\|_u \left\|\boldsymbol{P}_{\beta_{\boldsymbol{\theta}}} - \boldsymbol{P}_{\beta_{\boldsymbol{\theta}'}}\right\|_1$$

$$\leq \frac{C_{\mu_{\beta_{\boldsymbol{\theta}}}}}{1 - \left\|\boldsymbol{P}_{\beta_{\boldsymbol{\theta}}} - \mathbf{1}\boldsymbol{\mu}_{\beta_{\boldsymbol{\theta}}}^\top\right\|_u} \left\|\boldsymbol{P}_{\beta_{\boldsymbol{\theta}}} - \boldsymbol{P}_{\beta_{\boldsymbol{\theta}'}}\right\|_1$$

$$\leq \frac{C_{\mu_{\beta_{\boldsymbol{\theta}}}}}{1 - \left\|\boldsymbol{P}_{\beta_{\boldsymbol{\theta}}} - \mathbf{1}\boldsymbol{\mu}_{\beta_{\boldsymbol{\theta}}}^\top\right\|_u} L \left\|\boldsymbol{\theta} - \boldsymbol{\theta}'\right\|$$

where $\|\cdot\|_u$ is a norm such that $\left\|\boldsymbol{P}_{\beta_{\boldsymbol{\theta}}} - \mathbf{1}\boldsymbol{\mu}_{\beta_{\boldsymbol{\theta}}}^\top\right\|_u < 1$ which exists as $\rho(\boldsymbol{P}_{\beta_{\boldsymbol{\theta}'}} - \mathbf{1}\boldsymbol{\mu}_{\beta_{\boldsymbol{\theta}'}}^\top) < 1$. $C_{\mu_{\beta_{\boldsymbol{\theta}}}}$ is a scalar such that $\|\cdot\| \leq C_{\mu_{\beta_{\boldsymbol{\theta}}}} \|\cdot\|_u$ which exists by equivalence of norm. The second inequality follows from sum of geometric series. Therefore, local liopschitzness of $\mu_{\beta_{\boldsymbol{\theta}}}(a \mid s)$ follows. $\qquad\square$

## 11 STOCHASTIC APPROXIMATION

Let us consider a stochastic approximation scheme (Robbins and Monro, 1951):

$$\boldsymbol{x}_{k+1} = \boldsymbol{x}_k + \alpha_k\left(\boldsymbol{f}(\boldsymbol{x}_k) + \boldsymbol{\epsilon}_k\right)$$

where $f : \mathbb{R}^n \to \mathbb{R}^n$ is continuous function, $\boldsymbol{\epsilon}_k$ is a Martingale difference sequence and $\alpha_k \in [0, 1]$ is the step-size.

**Definition 11.1** (Martingale difference sequence). *Consider a sequence of random variables $\boldsymbol{\epsilon}_0, \boldsymbol{\epsilon}_1, \dots$ and let $\sigma$-fields $\mathcal{F}_k = \sigma(\boldsymbol{x}_0, \boldsymbol{\epsilon}_1, \dots, \boldsymbol{\epsilon}_k)$. If $\mathbb{E}\left[\boldsymbol{\epsilon}_{k+1}|\mathcal{F}_k\right] = 0$ and $\mathbb{E}[\|\boldsymbol{\epsilon}_{k+1}\|^2 |\mathcal{F}_k] < \infty$ almost surely for the $\sigma$-fields $\mathcal{F}_k$, $\boldsymbol{\epsilon}_k$ is called a Martingale difference sequence.*

The ODE counterpart can characterize the stability of stochastic approximation scheme:

$$\dot{\boldsymbol{x}}_t = \boldsymbol{f}(\boldsymbol{x}_t), \quad \boldsymbol{x}_0 \in \mathbb{R}^n, t \geq 0.$$

**Assumption 11.2.** *1. The mapping $\boldsymbol{f} : \mathbb{R}^n \to \mathbb{R}^n$ is globally Lipschitz continuous, and there exists a function $\boldsymbol{f}_\infty : \mathbb{R}^n \to \mathbb{R}^n$ such that*

$$\lim_{c \to \infty} \frac{\boldsymbol{f}(c\boldsymbol{x})}{c} = \boldsymbol{f}_\infty(\boldsymbol{x}), \quad \forall \boldsymbol{x} \in \mathbb{R}^n. \tag{19}$$

*2. The origin in $\mathbb{R}^n$ is a globally asymptotically stable equilibrium for the ODE $\dot{\boldsymbol{x}}_t = \boldsymbol{f}_\infty(\boldsymbol{x}_t)$.*

*3. There exists a globally asymptotically stable equilibrium $\boldsymbol{x}^* \in \mathbb{R}^n$ for the ODE $\dot{\boldsymbol{x}}_t = \boldsymbol{f}(\boldsymbol{x}_t)$, i.e., $\boldsymbol{x}_t \to \boldsymbol{x}^*$ as $t \to \infty$.*

*4. The sequence $\{\boldsymbol{\epsilon}_k, \mathcal{G}_k\}_{k \geq 1}$ where $\mathcal{G}_k$ is sigma-algebra generated by $\{(\boldsymbol{x}_i, \boldsymbol{\epsilon}_i), k \geq i\}$, is a Martingale difference sequence. In addition , there exists a constant $C_0 < \infty$ such that for any initial $\theta_0 \in \mathbb{R}^n$ , we have $\mathbb{E}[\|\boldsymbol{\epsilon}_{k+1}\|^2 |\mathcal{G}_k] \leq C_0(1 + \|\boldsymbol{x}_k\|^2), \forall k \geq 0$.*

*5. The step-sizes satisfies the Robbins-Monro condition (Robbins and Monro, 1951) :*

$$\sum_{k=0}^\infty \alpha_k = \infty, \quad \sum_{k=0}^\infty \alpha_k^2 < \infty.$$

**Lemma 11.3** (Borkar and Meyn Theorem, Borkar and Meyn (2000)). *Suppose Assumption 11.2 holds and there exists unique $\boldsymbol{x}^* \in \mathbb{R}^n$ such that $\boldsymbol{f}(\boldsymbol{x}^*) = \mathbf{0}$. Then, $\boldsymbol{x}_k \to \boldsymbol{x}^*$ with probability one.*

## 12 OMITTED PROOFS IN MAIN MANUSCRIPT

**Lemma 12.1.** *If $\eta > 3$, and $\|\phi(s,a)\|_\infty \leq \frac{1}{\sqrt{p}}$ for all $(s,a) \in \mathcal{S} \times \mathcal{A}$, then $\gamma \boldsymbol{\Phi}^\top \boldsymbol{D}_{\mu_{\beta_{\boldsymbol{\theta}}}} \boldsymbol{P} \boldsymbol{\Pi}_{\pi_{\boldsymbol{\theta}}} \boldsymbol{\Phi} - (\eta \boldsymbol{I} + \boldsymbol{\Phi}^\top \boldsymbol{D}_{\mu_{\beta_{\boldsymbol{\theta}}}} \boldsymbol{\Phi})$ is SNRDD for all $\boldsymbol{\theta} \in \mathbb{R}^p$.*

*Proof.* For $i \in \{1, 2, \ldots, p\}$, let us consider $S_i(\gamma \boldsymbol{\Phi}^\top \boldsymbol{D}_{\mu_{\beta_{\boldsymbol{\theta}}}} \boldsymbol{P} \boldsymbol{\Pi}_{\pi_{\boldsymbol{\theta}}} \boldsymbol{\Phi} - (\eta \boldsymbol{I} + \boldsymbol{\Phi}^\top \boldsymbol{D}_{\mu_{\beta_{\boldsymbol{\theta}}}} \boldsymbol{\Phi}))$ which is defined in Definition 3.1:

$$\left( -\eta - [\boldsymbol{\Phi}^\top \boldsymbol{D}_{\mu_{\beta_{\boldsymbol{\theta}}}} \boldsymbol{\Phi}]_{i,i}^2 + \gamma [\boldsymbol{\Phi}^\top \boldsymbol{D}_{\mu_{\beta_{\boldsymbol{\theta}}}} \boldsymbol{P} \boldsymbol{\Pi}_{\pi_{\boldsymbol{\theta}}^g} \boldsymbol{\Phi}]_{i,i} + \sum_{j \in \{1,2,\ldots,p\} \setminus \{i\}} \left| [-\boldsymbol{\Phi}^\top \boldsymbol{D}_{\mu_{\beta_{\boldsymbol{\theta}}}} \boldsymbol{\Phi} + \gamma \boldsymbol{\Phi}^\top \boldsymbol{D}_{\mu_{\beta_{\boldsymbol{\theta}}}} \boldsymbol{P} \boldsymbol{\Pi}_{\pi_{\boldsymbol{\theta}}} \boldsymbol{\Phi}]_{i,l} \right| \right)$$

$$= -\eta - \sum_{(s,a) \in \mathcal{S} \times \mathcal{A}} \mu_{\beta_{\boldsymbol{\theta}}}(s,a) \left( [\phi(s,a)]_i^2 - \gamma \sum_{s' \in \mathcal{S}} \mathcal{P}(s' \mid s,a) [\phi(s,a)]_i \left[ \sum_{u \in \mathcal{A}} \pi_{\boldsymbol{\theta}}(u \mid s') \phi(s',u) \right]_i \right)$$

$$+ \sum_{j \in \{1,2,\ldots,p\} \setminus \{i\}} \sum_{(s,a) \in \mathcal{S} \times \mathcal{A}} \left| \mu_{\beta_{\boldsymbol{\theta}}}(s,a) \sum_{s' \in \mathcal{S}} \mathcal{P}(s' \mid s,a) [\phi(s,a)]_i \left( [\phi(s,a)]_j - \gamma \left[ \sum_{u \in \mathcal{A}} \pi_{\boldsymbol{\theta}}(u \mid s') \phi(s',u) \right]_j \right) \right|$$

$$\leq -\eta + \sum_{(s,a) \in \mathcal{S} \times \mathcal{A}} \mu_{\beta_{\boldsymbol{\theta}}}(s,a) \left( \|\phi(s,a)\|_\infty^2 + \gamma \sum_{s' \in \mathcal{S}} \mathcal{P}(s' \mid s,a) \|\phi(s,a)\|_\infty \left\| \sum_{u \in \mathcal{A}} \pi_{\boldsymbol{\theta}}(u \mid s') \phi(s',u) \right\|_\infty \right)$$

$$+ \sum_{j \in \{1,2,\ldots,p\} \setminus \{i\}} \sum_{(s,a) \in \mathcal{S} \times \mathcal{A}} \mu_{\beta_{\boldsymbol{\theta}}}(s,a) \sum_{s' \in \mathcal{S}} \mathcal{P}(s' \mid s,a) \|\phi(s,a)\|_\infty \left( \|\phi(s,a)\|_\infty + \gamma \sum_{u \in \mathcal{A}} \pi_{\boldsymbol{\theta}}(u \mid s') \|\phi(s',u)\|_\infty \right)$$

$$\leq -\eta + \sum_{(s,a) \in \mathcal{S} \times \mathcal{A}} \mu_{\beta_{\boldsymbol{\theta}}}(s,a) \left( \frac{1}{p} + \gamma \sum_{s' \in \mathcal{S}} \mathcal{P}(s' \mid s,a) \frac{1}{p} \right)$$

$$+ \sum_{j \in \{1,2,\ldots,p\} \setminus \{i\}} \sum_{(s,a) \in \mathcal{S} \times \mathcal{A}} \mu_{\beta_{\boldsymbol{\theta}}}(s,a) \sum_{s' \in \mathcal{S}} \mathcal{P}(s' \mid s,a) \left( \frac{2}{p} \right)$$

$$\leq -\eta + \frac{2}{p} + 2$$

$$\leq -\eta + 3.$$

The first inequality follows from the fact that $|[\phi(s,a)]_i| \leq \|\phi(s,a)\|_\infty$ for all $(s,a) \in \mathcal{S} \times \mathcal{A}$ and $i \in \{1, 2, \ldots, p\}$ and using triangle inequality. The second inequality follows from the assumption that $\|\phi(s,a)\|_\infty \leq \frac{1}{\sqrt{p}}$ for all $(s,a) \in \mathcal{S} \times \mathcal{A}$. Therefore, $\eta > 3$ is sufficient for our goal.

$\square$

### 12.1 PROOF OF THEOREM 3.2

Now, let us present the proof of Theorem 3.2:

*Proof.* The first statement follows from Lemma 9.3 in the Appendix, which generalizes Theorem 3.2. The proof outline is as follows : We can check that for small enough choice of $\alpha$, $\boldsymbol{x} + \alpha \boldsymbol{F}(\boldsymbol{\theta}, \pi_{\boldsymbol{\theta}}, \mu_{\beta_{\boldsymbol{\theta}}})$ is a self-map, meaning it maps $\mathcal{C}$ onto itself, where $\mathcal{C}$ is a compact set. Moreover, using a continuous behavior and target policy, the function $\boldsymbol{F}$ is continuous. Therefore, we can apply the Brouwer's fixed point theory in Lemma 9.1 in the Appendix.

The second statement directly follows from Lemma 9.5, which applies the result of Davydov et al. (2024a) that for a locally Lipschitz function with SNRDD condition, the uniqueness of the solution is guaranteed. The only condition we need to check is the local Lipschitzness of $\boldsymbol{F}(\boldsymbol{\theta}, \pi_{\boldsymbol{\theta}}, \mu_{\beta_{\boldsymbol{\theta}}})$. The stationary distribution $\mu_{\beta_{\boldsymbol{\theta}}}$ is locally Lipschitz from Lemma 10.14 in the Appendix. As product of locally Lipschitz functions are still locally Lipschitz, which can be verified using Lemma 10.7 in the Appendix, $\boldsymbol{F}(\boldsymbol{\theta}, \pi_{\boldsymbol{\theta}}, \mu_{\beta_{\boldsymbol{\theta}}})$ is a locally Lipschitz function. Therefore, we can now apply Lemma 9.5 in the Appendix. $\square$

## 12.2 PROOF OF PROPOSITION 3.10

*Proof.* The first statement is a specific case of Lemma 9.8 in the Appendix, a generalized version of the first statement. The idea of the proof of Lemma 9.8 is the following : Consider the scenario when (6) holds. Multiplying $\boldsymbol{\Phi}$ on both sides of (5), we get

$$\boldsymbol{\Phi\theta} = \boldsymbol{\Phi}(\boldsymbol{\Phi}^\top \boldsymbol{D}_{\mu_{\beta_\theta}} \boldsymbol{\Phi})^{-1} \left( \gamma \boldsymbol{\Phi}^\top \boldsymbol{D}_{\mu_{\beta_\theta}} \boldsymbol{P}\boldsymbol{\Pi}_{\pi_\theta} \boldsymbol{\Phi\theta} + \boldsymbol{\Phi}^\top \boldsymbol{D}_{\mu_{\beta_\theta}} \boldsymbol{R} \right).$$

Let $\beta_\theta = \beta_{\boldsymbol{\Phi\theta}}$ and $\pi_\theta = \pi_{\boldsymbol{\Phi\theta}}$. Denote $\boldsymbol{y} = \boldsymbol{\Phi\theta}$ and $\boldsymbol{f}(\boldsymbol{y}) := \boldsymbol{\Phi}(\boldsymbol{\Phi}^\top \boldsymbol{D}_{\mu_{\beta_y}} \boldsymbol{\Phi})^{-1} \left( \gamma \boldsymbol{\Phi}^\top \boldsymbol{D}_{\mu_{\beta_y}} \boldsymbol{P}\boldsymbol{\Pi}_{\pi_y} \boldsymbol{y} + \boldsymbol{\Phi}^\top \boldsymbol{D}_{\mu_{\beta_y}} \boldsymbol{R} \right)$. Now, it is sufficient to investigate the solution of the equation $\boldsymbol{y} = \boldsymbol{f}(\boldsymbol{y})$. We can check that $\boldsymbol{y} + \alpha \boldsymbol{f}(\boldsymbol{y})$ is a self-map for some small enough $\alpha$. Moreover, as the policies $\mu_{\beta_y}$ and $\pi_y$ are continuous and $\boldsymbol{f}$ is also a continuous map, we can apply Brouwer's fixed point theorem in Lemma 9.1 in the Appendix. Therefore, we can apply Lemma 9.8 in Appendix Section 9. The same argument holds when we consider the condition (7).

The second statement is a specific case of Lemma 9.9 in the Appendix Section 9. The proof relies on a version of Lebourg's mean value theorem (Lemma 10.6 in the Appendix Section 9), which is applicable as $\pi_\theta$ is Lipschitz and $\beta_\theta$ is a fixed policy. $\qquad\square$

## 12.3 PROOF OF PROPOSITION 3.13

*Proof.* A generalized version of proof is provided in Proposition 9.11 in Appendix Section 9. The proof can be established using the definition of the infinity norm and SNRDD as given in Definition 3.1. $\qquad\square$

## 12.4 PROOF OF LEMMA 4.3

*Proof.* Let us provide the proof of the first statement, the case of asynchronous tabular Q-learning. For $i \in \mathcal{I}_\infty(\boldsymbol{Q} - \tilde{\boldsymbol{Q}})$,

$$[\boldsymbol{Q} - \tilde{\boldsymbol{Q}}]_i [\boldsymbol{F}_{\text{AsyncQ}}(\boldsymbol{Q}) - \boldsymbol{F}_{\text{AsyncQ}}(\tilde{\boldsymbol{Q}})]_i$$
$$= [\boldsymbol{Q} - \tilde{\boldsymbol{Q}}]_i [\gamma \boldsymbol{D}_d \boldsymbol{P} (\boldsymbol{\Pi}_{\pi_{\boldsymbol{Q}}^g} \boldsymbol{Q} - \boldsymbol{\Pi}_{\pi_{\tilde{\boldsymbol{Q}}}^g} \tilde{\boldsymbol{Q}}) - \boldsymbol{D}_d (\boldsymbol{Q} - \tilde{\boldsymbol{Q}})]_i$$

$$= [\boldsymbol{Q} - \tilde{\boldsymbol{Q}}]_i \left( \gamma [\boldsymbol{D}_d]_{i,i} \sum_{j \in [|\mathcal{S}|]} [\boldsymbol{P}]_{i,j} \left( \max_{u \in \mathcal{A}} [\boldsymbol{Q}]_{(j-1)|\mathcal{A}|+u} - \max_{u \in \mathcal{A}} [\tilde{\boldsymbol{Q}}]_{(j-1)|\mathcal{A}|+u} \right) - [\boldsymbol{D}_d]_{i,i} [\boldsymbol{Q} - \tilde{\boldsymbol{Q}}]_i \right)$$

$$\leq - [\boldsymbol{D}_d]_{i,i} \left| [\boldsymbol{Q} - \tilde{\boldsymbol{Q}}]_i \right|^2 + |[\boldsymbol{Q} - \tilde{\boldsymbol{Q}}]_i| \left( \gamma [\boldsymbol{D}_d]_{i,i} \sum_{j \in \mathcal{S}} [\boldsymbol{P}]_{i,j} \max_{u \in \mathcal{A}} |[\boldsymbol{Q}]_{(j-1)|\mathcal{A}|+a} - [\tilde{\boldsymbol{Q}}]_{(j-1)|\mathcal{A}|+a}| \right)$$

$$\leq (\gamma - 1)[\boldsymbol{D}_d]_{i,i} \left| [\boldsymbol{Q} - \tilde{\boldsymbol{Q}}]_i \right|^2$$

$$\leq (\gamma - 1) d_{\min} \left| [\boldsymbol{Q} - \tilde{\boldsymbol{Q}}]_i \right|^2.$$

The second last line follows from the non-expansiveness of the max-operator and $|[\boldsymbol{Q} - \tilde{\boldsymbol{Q}}]_{(s-1)|\mathcal{A}|+a}| \leq |[\boldsymbol{Q} - \tilde{\boldsymbol{Q}}]_i|$ for all $(s,a) \in \mathcal{S} \times \mathcal{A}$ since $i \in \mathcal{I}_\infty(\boldsymbol{Q} - \tilde{\boldsymbol{Q}})$. This proves the first statement.

Now, let us prove the second statement, the case for linear Q-learning. For $\boldsymbol{\theta}, \tilde{\boldsymbol{\theta}} \in \mathbb{R}^p$ and $i \in \mathcal{I}_\infty(\boldsymbol{\theta} - \tilde{\boldsymbol{\theta}})$, from Lebourg's mean value theorem in Lemma 10.5 in the Appendix, we have

$$[\boldsymbol{F}_{\text{linear}}(\boldsymbol{\theta}) - \boldsymbol{F}_{\text{linear}}(\tilde{\boldsymbol{\theta}})]_i = \boldsymbol{a}_i^\top (\boldsymbol{\theta} - \tilde{\boldsymbol{\theta}})$$

where $\boldsymbol{v} \in \{t\boldsymbol{\theta} + (1-t)\tilde{\boldsymbol{\theta}} : t \in [0,1]\}$, $\boldsymbol{a}_i \in \text{conv}\{\lim_{k\to\infty} \nabla \boldsymbol{F}_{\text{linear}}(\boldsymbol{x}_k)^\top \boldsymbol{e}_i : \boldsymbol{x}_k \to \boldsymbol{v}, \ \boldsymbol{x}_k \in \mathcal{D}_{\boldsymbol{F}_{\text{linear}}}\}$ and $\mathcal{D}_{\boldsymbol{F}_{\text{linear}}}$ is the differentiable points of $\boldsymbol{F}_{\text{linear}}$. $\boldsymbol{a}_i$ can be expressed as $\boldsymbol{a}_i = \sum_{j=1}^q \lambda_j \lim_{k\to\infty} \nabla \boldsymbol{F}_{\text{linear}}(\boldsymbol{x}_k^j)^\top \boldsymbol{e}_i$ for some $q \in \mathbb{N}$, $\sum_{j=1}^q \lambda_j = 1$ and for $j \in [q]$, $\lambda_j \geq 0$

and $\{\boldsymbol{x}_k^j\}_{k=1}^\infty$ is a converging sequence to $\boldsymbol{v}$. We have,

$$[\boldsymbol{\theta} - \tilde{\boldsymbol{\theta}}]_i \cdot \boldsymbol{a}_i^\top (\boldsymbol{\theta} - \tilde{\boldsymbol{\theta}})$$

$$= -[\boldsymbol{\Phi}^\top \boldsymbol{D}_d \boldsymbol{\Phi}]_i^2 |[\boldsymbol{\theta} - \tilde{\boldsymbol{\theta}}]_i|^2 + [\boldsymbol{\theta} - \tilde{\boldsymbol{\theta}}]_i \left[ \lim_{k\to\infty} \sum_{j=1}^q \lambda_j \gamma \boldsymbol{\Phi}^\top \boldsymbol{D}_d \boldsymbol{\Phi} \boldsymbol{P} \boldsymbol{\Pi}_{\pi_{\boldsymbol{x}_k^j}^g} \boldsymbol{\Phi} (\boldsymbol{\theta} - \tilde{\boldsymbol{\theta}}) \right]_i$$

$$= \lim_{k\to\infty} \left( -[\boldsymbol{\Phi}^\top \boldsymbol{D}_d \boldsymbol{\Phi}]_i^2 + \gamma \sum_{j=1}^q \lambda_j [\boldsymbol{\Phi}^\top \boldsymbol{D}_d \boldsymbol{P} \boldsymbol{\Pi}_{\pi_{\boldsymbol{x}_k^j}^g} \boldsymbol{\Phi}]_{i,i} \right) \left| [\boldsymbol{\theta} - \tilde{\boldsymbol{\theta}}]_i \right|^2$$

$$+ \lim_{k\to\infty} \gamma \sum_{l\in[p]\backslash\{i\}} \sum_{j=1}^q \lambda_j [\boldsymbol{\Phi}^\top \boldsymbol{D}_d \boldsymbol{P} \boldsymbol{\Pi}_{\pi_{\boldsymbol{x}_k^j}^g} \boldsymbol{\Phi}]_{i,l} \left[ \boldsymbol{\theta} - \tilde{\boldsymbol{\theta}} \right]_i \left[ \boldsymbol{\theta} - \tilde{\boldsymbol{\theta}} \right]_l$$

$$= \lim_{k\to\infty} \sum_{j=1}^q \lambda_j \left( -[\boldsymbol{\Phi}^\top \boldsymbol{D}_d \boldsymbol{\Phi}]_i^2 + \gamma [\boldsymbol{\Phi}^\top \boldsymbol{D}_d \boldsymbol{P} \boldsymbol{\Pi}_{\pi_{\boldsymbol{x}_k^j}^g} \boldsymbol{\Phi}]_{i,i} \right) \left\| \boldsymbol{\theta} - \tilde{\boldsymbol{\theta}} \right\|_\infty^2$$

$$+ \lim_{k\to\infty} \sum_{j=1}^q \lambda_j \gamma \sum_{l\in[p]\backslash\{i\}} [\boldsymbol{\Phi}^\top \boldsymbol{D}_d \boldsymbol{P} \boldsymbol{\Pi}_{\pi_{\boldsymbol{x}_k^j}^g} \boldsymbol{\Phi}]_{i,l} \left[ \boldsymbol{\theta} - \tilde{\boldsymbol{\theta}} \right]_i \left[ \boldsymbol{\theta} - \tilde{\boldsymbol{\theta}} \right]_l$$

$$\leq \lim_{k\to\infty} \sum_{j=1}^q \lambda_j a_{\min} \left\| \boldsymbol{\theta} - \tilde{\boldsymbol{\theta}} \right\|_\infty^2$$

$$= a_{\min} \left\| \boldsymbol{\theta} - \tilde{\boldsymbol{\theta}} \right\|_\infty^2$$

where the second equality follows from simple algebraic decomposition and the last inequality follows from the choice that $|[\boldsymbol{\theta} - \tilde{\boldsymbol{\theta}}]_i| = \left\| \boldsymbol{\theta} - \tilde{\boldsymbol{\theta}} \right\|_\infty$ and from the definition of $a_{\min}$:

$$a_{\min} := \max_{\boldsymbol{x}\in\mathcal{D}_{\boldsymbol{F}_{\text{linear}}}} \max_{i\in[p]} \left( -[\boldsymbol{\Phi}^\top \boldsymbol{D}_d \boldsymbol{\Phi}]_i^2 + \gamma [\boldsymbol{\Phi}^\top \boldsymbol{D}_d \boldsymbol{P} \boldsymbol{\Pi}_{\pi_{\boldsymbol{x}}^g} \boldsymbol{\Phi}]_{i,i} + \gamma \sum_{l\in[p]\backslash\{i\}} \left| [\boldsymbol{\Phi}^\top \boldsymbol{D}_d \boldsymbol{P} \boldsymbol{\Pi}_{\pi_{\boldsymbol{x}}^g} \boldsymbol{\Phi}]_{i,l} \right| \right).$$

The third statement (one-sided Lipschitzness of regularized Q-learning) follows from the same logic as the second statement.

$\square$

### 12.5 PROOF OF PROPOSITION 4.4

*Proof.* The proof follows from applying the Borkar and Meyn Theorem in Lemma 11.3 in Appendix Section 10. Let us verify the items in Assumption 11.2 in Appendix Section 10:

Let us first check item 2 and item 3 of Assumption 11.2. We can see that the ODE counterparts of the Q-learning admit a globally asymptotically stable equilibrium point by one-sided Liipschitzness in Lemma 4.3 and the existence of the solution to PBE in Theorem 3.2.

Now, let us verify the remaining items of Assumption 11.2. Global Lipschitz condition of item 1 follows from the fact that max-operator is a Lipschitz operator. The fourth item can be verified using triangle inequalities and fifth item follows from our assumption on the Robbins-Monro step-size (Robbins and Monro, 1951). $\square$

**Lemma 12.2** (Convergence of AVI). *Consider the update in (11). If (6) or (7) holds, then a unique solution of (1), say $\boldsymbol{\theta}^*$ exists, and $\boldsymbol{\theta}_k \to \boldsymbol{\theta}^*$*

*Proof.* Let us consider the condition in (6). Multiply $\boldsymbol{\Phi}$ on both sides, and then subtracting $\boldsymbol{\Phi}\boldsymbol{\theta}^*$, we get

$$\left\|\mathbf{\Phi}(\boldsymbol{\theta}_{k+1} - \boldsymbol{\theta}^*)\right\|_\infty = \left\|\mathbf{\Phi}(\mathbf{\Phi}^\top \boldsymbol{D}_d \mathbf{\Phi})^{-1}\mathbf{\Phi}^\top \boldsymbol{D}_d(\gamma \boldsymbol{P}\mathbf{\Pi}_{\pi^g_{\mathbf{\Phi}\boldsymbol{\theta}_k}}\mathbf{\Phi}\boldsymbol{\theta}_k - \gamma \boldsymbol{P}\mathbf{\Pi}_{\pi^g_{\mathbf{\Phi}\boldsymbol{\theta}^*}}\mathbf{\Phi}\boldsymbol{\theta}^*)\right\|_\infty$$

$$\leq c \left\|\mathbf{\Phi}\boldsymbol{\theta}_k - \mathbf{\Phi}\boldsymbol{\theta}^*\right\|_\infty$$

where $c := \sup_{\boldsymbol{\theta}\in\mathcal{D}} \gamma \left\|\mathbf{\Phi}(\mathbf{\Phi}^\top \boldsymbol{D}_{\mu_{\beta_{\mathbf{\Phi}\boldsymbol{\theta}}}}\mathbf{\Phi})^{-1}\mathbf{\Phi}^\top \boldsymbol{D}\boldsymbol{P}\mathbf{\Pi}_{\pi_{\mathbf{\Phi}\boldsymbol{\theta}}}\right\|_\infty < 1$, and the second inequality follows the application of Lebourg's mean value theorem from Lemma 10.6 in the Appendix. Therefore, We have $\left\|\mathbf{\Phi}(\boldsymbol{\theta}_{k+1} - \boldsymbol{\theta}^*)\right\|_\infty \to 0$. The same argument holds when (7) holds. $\qquad\square$

## 13  MDP EXAMPLES

We define the TD-fixed point for a policy $\pi$ as $\boldsymbol{\theta}^\pi := (\mathbf{\Phi}^\top \boldsymbol{D}\mathbf{\Phi} - \gamma\mathbf{\Phi}\boldsymbol{D}\boldsymbol{P}\mathbf{\Pi}_\pi\mathbf{\Phi})^{-1}\mathbf{\Phi}^\top \boldsymbol{D}\boldsymbol{R}$. For each greedy policy $\pi \in \Omega$, if the greedy policy $\pi^g_{\boldsymbol{\theta}^\pi}$ induced by the TD-fixed point $\boldsymbol{\theta}^\pi$ differs from $\pi$, then $\boldsymbol{\theta}^\pi$ is not a solution to the PBE.

The step-size for linear Q-learning is set to $0.1$ across all experiments.

**Example 13.1** (Q-learning converges but AVI does not). *Consider an MDP with $|\mathcal{S}| = |\mathcal{A}| = 2$ and $p = 2$:*

$$\mathbf{\Phi} = \begin{bmatrix} 0.34 & -0.59 \\ 0.25 & -0.16 \\ -0.92 & 0.37 \\ 0.83 & 0.19 \end{bmatrix}, \quad \boldsymbol{P} = \begin{bmatrix} 0 & 1 \\ 0.02 & 0.98 \\ 0.99 & 0.01 \\ 0.05 & 0.95 \end{bmatrix}, \quad \boldsymbol{R} = \begin{bmatrix} 0.3 \\ -0.47 \\ -0.87 \\ -1 \end{bmatrix}, \quad \beta(1 \mid 1) = 0.96, \quad \beta(1 \mid 2) = 0.19.$$

*Then, for any $\pi \in \Omega$, where $\Omega$ is the set of deterministic policies, we can check that $-\mathbf{\Phi}^\top \boldsymbol{D}_{\mu_\beta}\mathbf{\Phi} + \gamma\mathbf{\Phi}^\top \boldsymbol{D}_{\mu_\beta}\boldsymbol{P}\mathbf{\Pi}_\pi\mathbf{\Phi}$ is SNRDD. Therefore, by Theorem 3.2, there exists a unique solution to PBE, $\boldsymbol{\theta}^* \approx \begin{bmatrix} -0.67 \\ -1.76 \end{bmatrix}$, and Q-learning will converge to this solution by Proposition 4.4. Moreover, we can check that $\rho(\gamma(\mathbf{\Phi}^\top \boldsymbol{D}_{\mu_\beta}\mathbf{\Phi})^{-1}\mathbf{\Phi}^\top \boldsymbol{D}_{\mu_\beta}\boldsymbol{P}\mathbf{\Pi}_{\pi_{\boldsymbol{\theta}^*}}\mathbf{\Phi}) \approx 1.08 > 1$, and AVI algorithm cannot converge to this solution. Experimental results are given in Figure 2 and Figure 3.*

**Example 13.2** (AVI converges but Q-learning does not ).

$$\mathbf{\Phi} = \begin{bmatrix} 0.37 & 0.99 \\ 0.97 & 1 \\ -1 & -0.95 \\ -0.77 & 0.19 \end{bmatrix}, \quad \boldsymbol{P} = \begin{bmatrix} 0.99 & 0.01 \\ 0.99 & 0.01 \\ 0.89 & 0.11 \\ 0.42 & 0.58 \end{bmatrix}, \quad \boldsymbol{R} = \begin{bmatrix} -0.31 \\ -0.46 \\ -0.35 \\ 0.73 \end{bmatrix}, \quad \beta(1 \mid 1) = 0.59, \ \beta(1 \mid 2) = 0.98.$$

*One can check that the solution to PBE is $\boldsymbol{\theta}^* \approx \begin{bmatrix} -1.26 \\ 0.89 \end{bmatrix}$. The condition (7) is satisfied at this point, and hence AVI converges. Nonetheless, $-\mathbf{\Phi}^\top \boldsymbol{D}_{\mu_\beta}\mathbf{\Phi} + \gamma\mathbf{\Phi}^\top \boldsymbol{D}_{\mu_\beta}\boldsymbol{P}\mathbf{\Pi}_{\pi^g_{\boldsymbol{\theta}^*}}\mathbf{\Phi}$ is not a Hurwitz matrix, and therefore, Q-learning does not converge to $\boldsymbol{\theta}^*$. The experimental results are shown in Figure 2 and Figure 4.*

**Example 13.3** (SNRDD can lead convergence to a point which induces sub-optimal policy).

$$\mathbf{\Phi} = \begin{bmatrix} 0.13 & 0.09 \\ 1 & 0.84 \\ -0.59 & 0.64 \\ -0.94 & -0.28 \end{bmatrix}, \quad \boldsymbol{P} = \begin{bmatrix} 0.99 & 0.01 \\ 0.37 & 0.63 \\ 0.99 & 0.01 \\ 0.99 & 0.01 \end{bmatrix}, \quad \boldsymbol{R} = \begin{bmatrix} -0.48 \\ 0.48 \\ 0.41 \\ 0.18 \end{bmatrix}, \quad \beta(1 \mid 1) = 0.98, \ \beta(1 \mid 2) = 0.96$$

*There are two solutions to PBE, which are $\boldsymbol{\theta}^*_1 \approx \begin{bmatrix} -1.26 \\ -0.27 \end{bmatrix}$ and $\boldsymbol{\theta}^*_2 \approx \begin{bmatrix} -0.45 \\ 0.98 \end{bmatrix}$. One can check that $-\mathbf{\Phi}^\top \boldsymbol{D}_{\mu_\beta}\mathbf{\Phi} + \gamma\mathbf{\Phi}^\top \boldsymbol{D}_{\mu_\beta}\boldsymbol{P}\mathbf{\Pi}_{\pi^g_{\boldsymbol{\theta}^*_1}}\mathbf{\Phi}$ is SNRDD and if we initialize nearby by $\boldsymbol{\theta}^*_1$, then the iterate of the Q-learning will converge to $\boldsymbol{\theta}^*_1$. Meanwhile, the optimal policy corresponds to the greedy policy induced by $\boldsymbol{\theta}^*_2$ whereas $\boldsymbol{\theta}^*_1$ induces a sub-optimal policy, i.e, the expected sum of discounted return is lower. The experimental results can be verified in Figure 1a.*

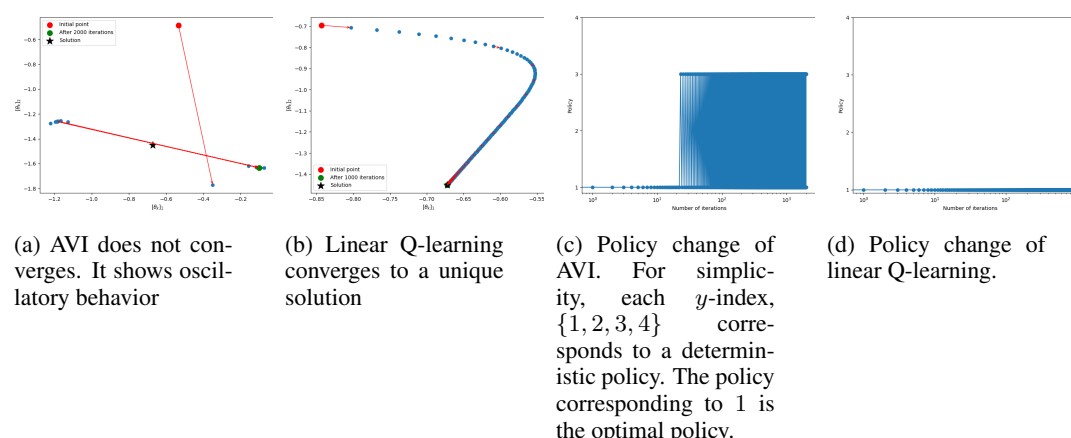

(a) AVI does not converges. It shows oscillatory behavior

(b) Linear Q-learning converges to a unique solution

(c) Policy change of AVI. For simplicity, each $y$-index, $\{1, 2, 3, 4\}$ corresponds to a deterministic policy. The policy corresponding to $1$ is the optimal policy.

(d) Policy change of linear Q-learning.

Figure 3: Experimental results on Example 13.1.

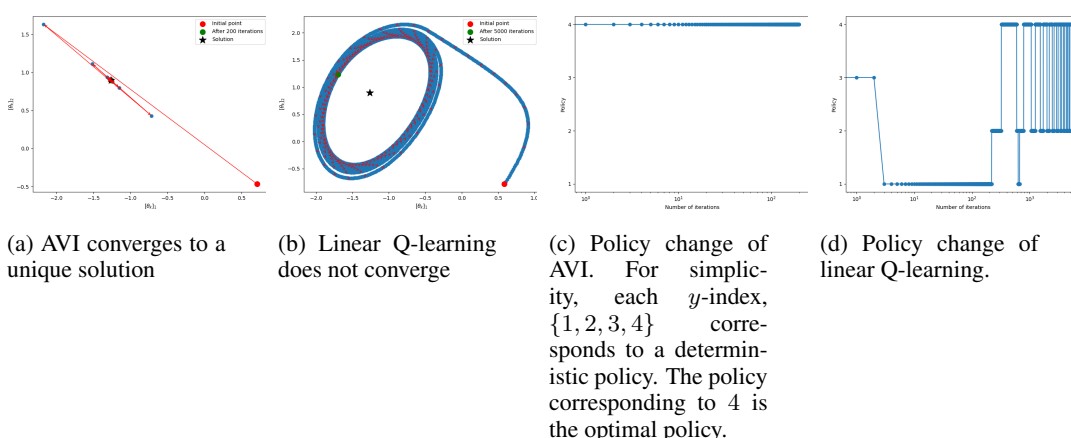

(a) AVI converges to a unique solution

(b) Linear Q-learning does not converge

(c) Policy change of AVI. For simplicity, each $y$-index, $\{1, 2, 3, 4\}$ corresponds to a deterministic policy. The policy corresponding to $4$ is the optimal policy.

(d) Policy change of linear Q-learning.

Figure 4: Experimental results on Example 13.2.

## 14 $\epsilon$-GREEDY SOLUTION EXAMPLE

**Example 14.1.** *Consider a MDP with $|\mathcal{S}| = 1$ and $|\mathcal{A}| = 2$:*

$$\boldsymbol{\Phi} = \begin{bmatrix} 0.45 \\ 0.79 \end{bmatrix}, \quad \boldsymbol{P} = \begin{bmatrix} 1 \\ 1 \end{bmatrix}, \quad \boldsymbol{\Pi}_{\pi_1^\epsilon} = \begin{bmatrix} \epsilon & 1-\epsilon \end{bmatrix}, \quad \boldsymbol{\Pi}_{\pi_2^\epsilon} = \begin{bmatrix} 1-\epsilon & \epsilon \end{bmatrix}, \quad \boldsymbol{R} = \begin{bmatrix} 0.5 \\ -0.78 \end{bmatrix},$$

*where $\pi_1^\epsilon$ and $\pi_2^\epsilon$ are two different $\epsilon$-greedy policies. The corresponding stationary distribution of $\boldsymbol{\Pi}_{\pi_1^\epsilon}$ and $\boldsymbol{\Pi}_{\pi_2^\epsilon}$ is $\boldsymbol{D}_{\mu_{\pi_1^\epsilon}} = \begin{bmatrix} \epsilon & 0 \\ 0 & 1-\epsilon \end{bmatrix}$ and $\boldsymbol{D}_{\mu_{\pi_2^\epsilon}} = \begin{bmatrix} 1-\epsilon & 0 \\ 0 & \epsilon \end{bmatrix}$. From Figure 1b, we can check that once a critical value is crossed over, then the number of solution changes.*

*Bertsekas (2011); Young and Sutton (2020) provided examples that the number of solution changes depending on the value of transition probability or reward. Our example differs as the change is determined by the value of $\epsilon$, which reflects the degree of exploration.*

**Example 14.2** ($\epsilon$-greedy adds stable unstable solution). *Consider the following MDP with $|\mathcal{S}| = 1, |\mathcal{A}| = 2$ and $\gamma = 0.99$:*

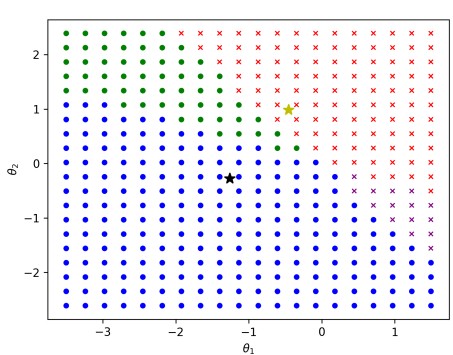
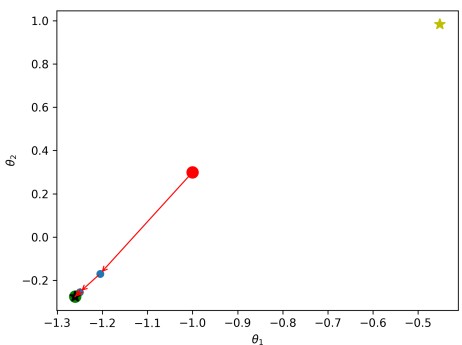

(a) Each color—red, green, purple, and blue—represents the greedy policy induced within each corresponding region. The 'o' and 'x' markers indicate whether the SNRDD condition is satisfied. The black and yellow stars denote the solution of the PBE. As illustrated in Figure 1a, when the initial point lies in the blue region, the trajectory converges locally to the star colored in black located within that region. Moreover, the region where condition motivated from (6) holds coincides with the region which SNRDD condition holds.

(b) This example shows convergence of AVI to the black point when initialized nearby the black point.

Figure 5: The figures show local convergence of linear Q-learning and AVI where the SNRDD condition and the condition motivated from (6) is met only locally in Example 13.3.

$$\mathbf{\Phi} = \begin{bmatrix} x \\ y \end{bmatrix}, \quad \mathbf{P} = \begin{bmatrix} 1 \\ 1 \end{bmatrix}, \quad \mathbf{\Pi}_{\pi_1^\epsilon} = \begin{bmatrix} 1-\epsilon & \epsilon \end{bmatrix}, \quad \mathbf{\Pi}_{\pi_2^\epsilon} = \begin{bmatrix} \epsilon & 1-\epsilon \end{bmatrix}, \quad \mathbf{R} = \begin{bmatrix} r_1 \\ r_2 \end{bmatrix}$$

$$\mathbf{\Pi}_{\pi_1^g} = \begin{bmatrix} 1 & 0 \end{bmatrix}, \quad \mathbf{\Pi}_{\pi_2^g} = \begin{bmatrix} 0 & 1 \end{bmatrix}, \quad \mathbf{D}_{\mu_{\pi_1^\epsilon}} = \begin{bmatrix} 1-\epsilon & 0 \\ 0 & \epsilon \end{bmatrix}, \quad \mathbf{D}_{\mu_{\pi_2^\epsilon}} = \begin{bmatrix} \epsilon & 0 \\ 0 & 1-\epsilon \end{bmatrix}$$

where $\pi_1^g$ and $\pi_2^g$ represent greedy policies that choose the first and second action, respectively, while $\pi_1^\epsilon$ and $\pi_2^\epsilon$ are the corresponding $\epsilon$-greedy policies, respectively.

Then, we can calculate the following quantities:

$$\mathbf{\Phi}^\top \mathbf{D}_{\mu_{\pi_1^\epsilon}} \mathbf{\Phi} = (1-\epsilon)x^2 + \epsilon y^2, \quad \mathbf{\Phi}^\top \mathbf{D}_{\mu_{\pi_2^\epsilon}} \mathbf{\Phi} = \epsilon x^2 + (1-\epsilon)y^2,$$

$$\mathbf{\Phi}^\top \mathbf{D}_{\mu_{\pi_1^\epsilon}} \mathbf{P}\mathbf{\Pi}_{\pi_1^g} \mathbf{\Phi} = \begin{bmatrix} x & y \end{bmatrix} \begin{bmatrix} 1-\epsilon & 0 \\ \epsilon & 0 \end{bmatrix} \begin{bmatrix} x \\ y \end{bmatrix} = (1-\epsilon)x^2 + +\epsilon xy,$$

$$\mathbf{\Phi}^\top \mathbf{D}_{\mu_{\pi_2^\epsilon}} \mathbf{P}\mathbf{\Pi}_{\pi_2^g} \mathbf{\Phi} = \begin{bmatrix} x & y \end{bmatrix} \begin{bmatrix} 0 & \epsilon \\ 0 & 1-\epsilon \end{bmatrix} \begin{bmatrix} x \\ y \end{bmatrix} = (1-\epsilon)y^2 + \epsilon xy,$$

$$\mathbf{\Phi}^\top \mathbf{D}_{\mu_{\pi_1^\epsilon}} \mathbf{R} = (1-\epsilon)xr_1 + \epsilon yr_2, \quad \mathbf{\Phi}^\top \mathbf{D}_{\mu_{\pi_2^\epsilon}} \mathbf{R} = \epsilon xr_1 + (1-\epsilon)yr_2.$$

Now, we can see that

$$A_1 = \mathbf{\Phi}^\top \mathbf{D}_{\mu_{\pi_1^\epsilon}} \mathbf{\Phi} - \gamma \mathbf{\Phi}^\top \mathbf{D}_{\mu_{\pi_1^\epsilon}} \mathbf{P}\mathbf{\Pi}_{\pi_1^g} \mathbf{\Phi} = (1-\epsilon)x^2 + \epsilon y^2 - \gamma((1-\epsilon)x^2 + \epsilon xy)$$

$$= \epsilon(-(1-\gamma)x^2 - \gamma xy + y^2) + (1-\gamma)x^2$$

$$A_2 = \mathbf{\Phi}^\top \mathbf{D}_{\mu_{\pi_2^\epsilon}} \mathbf{\Phi} - \gamma \mathbf{\Phi}^\top \mathbf{D}_{\mu_{\pi_2^\epsilon}} \mathbf{P}\mathbf{\Pi}_{\pi_2^g} \mathbf{\Phi} = \epsilon x^2 + (1-\epsilon)y^2 - \gamma((1-\epsilon)y^2 + \epsilon xy)$$

$$= \epsilon(x^2 - \gamma xy - (1-\gamma)y^2) + (1-\gamma)y^2$$

*Therefore we can now calculate $\theta^{\pi_1^g}$ and $\theta^{\pi_2^g}$, which are the TD-fixed points for the policies $\pi_1^g$ and $\pi_2^g$, respectively:*

$$\theta^{\pi_1^g} = \frac{(1-\epsilon)xr_1 + \epsilon y r_2}{\epsilon(-(1-\gamma)x^2 - \gamma xy + y^2) + (1-\gamma)x^2},$$

$$\theta^{\pi_2^g} = \frac{\epsilon x r_1 + (1-\epsilon)yr_2}{\epsilon(x^2 - \gamma xy - (1-\gamma)y^2) + (1-\gamma)y^2}.$$

*Suppose $y > x > 0$ and $r_1, r_2 < 0$. For $\theta^{\pi_1^g}$ to be a solution, we require $\theta^{\pi_1^g} < 0$ which is satisfied if $A_1 > 0$. Likewise, for $\theta^{\pi_2^g}$ to be a solution, we would require $A_2 < 0$.*

*Let $x = 0.5$ and $y = 1$. Then $A_1 = 0.5\epsilon + 0.0025$, and for $\epsilon > -0.005$, $A_1 > 0$ holds. Therefore, for all $\epsilon \in (0,1)$, $\theta^{\pi_1^g}$ is a solution fo PBE. Meanwhile, $A_2 = -0.255\epsilon + 0.01$ and $A_2 < 0$ holds if $0.04 < \epsilon$. Therefore, $\theta^{\pi_2^g}$ becomes a solution of PBE when $0.04 < \epsilon$.*

*Let us discuss the stability of each point in terms of Q-learning. Note that as $A_1 > 0$, Q-learning will converge to this solution. In contrast, as $A_2 < 0$, Q-learning will not converge to this solution.*

*The optimality of each policy depends on the relative values of $r_1$ and $r_2$. When $r_2 < r_1$, the policy $\pi_1^g$ becomes optimal. Conversely, if $r_1 < r_2$, then the policy $\pi_2^g$ is the optimal policy.*

## 15 RELATED WORKS ON LINEAR Q-LEARNING

This section provides additional literature on linear Q-learning. Several studies have proposed variations of linear Q-learning. Chen et al. (2023) explored the use of target networks and truncation, while Maei et al. (2010); Devraj and Meyn (2017); Carvalho et al. (2020) employed a two-time-scale approach to design a convergent linaer Q-learning algorithm. Although these methods ensure boundedness or convergence, the exact points to which the algorithm converges remain not well understood. In a slightly different setting, Lu et al. (2021) explored a linear programming formulation of Q-learning under deterministic transitions. Furthermore, Che et al. (2024) examined Q-learning with a target network in an overparameterized regime, where the number of features exceeds the size of state-action space.

Lu et al. (2018) provided an example that for a certain regime of $\epsilon$, Q-learning using $\epsilon$-greedy behavior policy can yield a sub-optimal policy compared to possible ones that can be represented by the linear feature while the optimal policy is not realizable. The set of realizable policy by the linear feature set (Lu et al., 2018) is defined as

$$\left\{ \pi \in \Omega : \pi(s) = arg \max_{a \in \mathcal{A}} \phi(s, a)^\top \boldsymbol{\theta}, \boldsymbol{\theta} \in \mathbb{R}^p \right\}.$$

The optimal policy $\pi^*$ may not be in above set, and therefore, the solution of PBE might induce only sub-optimal policies.

## 16 EXTENSION TO NON-LINEAR FUNCTION APPROXIMATION

The contraction theory-based analysis explicitly highlights the challenges in extending these results to the nonlinear function approximation setting. For simplicity let us fix the target policy $\pi$. Let $F_{\text{pbe}}(x) = \nabla f(x)^\top D(R + \gamma P\Pi^\pi f(x))$ and $f : \mathbb{R}^p \to \mathbb{R}^{|\mathcal{S}|}$ approximates the value function, $x \in \mathbb{R}^p$ is the learnable parameter, and $f(x; s)$ denotes the $s$-th element of $f(x)$. The contraction theory [R4] states that, if the Jacobian $\frac{\partial F_{\text{pbe}}}{\partial x}$ is Hurwitz, then every two trajectories of the ODE $\dot{x}_t = F_{\text{pbe}}(x_t)$ are converging. If we calculate the Jacobian $\frac{\partial F_{\text{pbe}}}{\partial x}$, we get

$$\frac{\partial}{\partial x}\left(\nabla f(x)^\top DR + \gamma \nabla f(x)^\top DP^\pi f(x) - \nabla f(x)^\top D\nabla f(x)\right)$$

$$= \underbrace{\sum_s d(s)\nabla^2 f(\boldsymbol{x}; s)\left(\sum_{s'} P^\pi(s' \mid s)\left(r(s, s') + \gamma f(\boldsymbol{x}; s') - f(\boldsymbol{x}; s)\right)\right)}_{I_1}$$

$$+ \underbrace{\sum_s d(s)\left(\gamma \sum_{s' \in \mathcal{S}} P^\pi(s' \mid s)\nabla f(x; s)\nabla f(x; s')^\top - \nabla f(x; s)\nabla f(x; s)^\top\right)}_{I_2}$$

The term $I_1$ appears due to using non-linear function approximation whereas $I_2$ is the term that also appears in the linear function approximation setting.. Consequently, while the $I_2$ can be controlled by the SNRDD approach but it is not clear how to control the $I_1$ term, which is the unique challenge in the analysis. The work by Gallici et al. (2025) considers layer normalization and regularization to ensure that $\frac{\partial F_{\mathrm{pbe}}}{\partial \boldsymbol{x}}$ to be negative definite under the infinite width regime of neural network. It is not clear how the analysis can be extended to the case of finite-width case and the case of Q-learning which includes max-operator.

## 17 PSEUDO CODE

---

**Algorithm 1** (regularized) Q-learning with linear function approximation

---

1: Initialize $\boldsymbol{\theta}_0 \in \mathbb{R}^p,\ \eta \in \mathbb{R}$.
2: **for** iteration step $k \in \{0, 1, \ldots\}$ **do**
3:     Observe $s_k, a_k \sim d(\cdot),\ s'_k \sim \mathcal{P}(\cdot \mid s_k, a_k)$, and $r_k = r(s_k, a_k, s'_k)$.
4:     Update parameters according to

$$\boldsymbol{\theta}_{k+1} = \boldsymbol{\theta}_k + \alpha_k \boldsymbol{\phi}(s_k, a_k)(r_k + \gamma \max_{a \in \mathcal{A}} \boldsymbol{\phi}^\top(s'_k, a)\boldsymbol{\theta}_k - \boldsymbol{\phi}(s_k, a_k)^\top \boldsymbol{\theta}_k - \eta \boldsymbol{\theta}_k).$$

5: **end for**

---

**Algorithm 2** Deterministic (regularized) Q-learning with linear function approximation

---

1: Initialize $\boldsymbol{\theta}_0 \in \mathbb{R}^p,\ \eta \in \mathbb{R},\ d \in \Delta^{\mathcal{S} \times \mathcal{A}}$.
2: **for** iteration step $k \in \{0, 1, \ldots\}$ **do**
3:

$$\boldsymbol{\theta}_{k+1} = \boldsymbol{\theta}_k + \alpha_k(\boldsymbol{\Phi}^\top \boldsymbol{D}_d \boldsymbol{R} + \gamma \boldsymbol{\Phi}^\top \boldsymbol{D}_d \boldsymbol{P}\boldsymbol{\Pi}_{\pi^g_{\boldsymbol{\theta}_k}}\boldsymbol{\Phi}\boldsymbol{\theta}_k - \boldsymbol{\Phi}^\top \boldsymbol{D}_d \boldsymbol{\Phi}\boldsymbol{\theta}_k - \eta\boldsymbol{\theta}_k).$$

4: **end for**

---

