# OpenReview forum: "Understanding the theoretical properties of projected Bellman equation, linear Q-learning, and approximate value iteration"
_ICLR.cc/2026/Conference — Submitted to ICLR 2026_

### Official Review · Reviewer_EKy9 · 2025-10-26

**Soundness:** 2
**Presentation:** 2
**Contribution:** 2
**Rating:** 2
**Confidence:** 3

**Summary:**

The paper studies the projected Bellman equation (PBE) and two algorithms used to solve it: linear Q-learning and approximate value iteration (AVI).

It proposes (i) an SNRDD (strictly negatively row-dominating diagonal) condition as a sufficient condition for existence/uniqueness of PBE solutions and convergence of linear Q-learning, (ii) a second sufficient condition motivated by AVI (two infinity-norm contraction conditions), and (iii) examples highlighting pathological behaviors under $\epsilon$-greedy policies.

The contributions are summarized in the introduction (existence/uniqueness under SNRDD; a second AVI-motivated condition; convergence proofs; examples where AVI converges but linear Q-learning does not, and vice-versa; $\epsilon$-greedy pathologies).

**Strengths:**

1. **Clear examples illustrating ε-greedy pathologies.**

Section 6 presents counterexamples showing how changing ε can lead to the existence of no, one, or multiple PBE solutions. This illustrates that discontinuous policies can undermine fixed-point guarantees even under SNRDD conditions.

2. **Comparative visualization of algorithm behavior.**

The examples and figures demonstrate cases where AVI converges but linear Q-learning diverges, and vice versa. These visual results help highlight subtle differences between projection-based and iterative schemes.

3. **Technically correct foundational results.**

The use of fixed-point theorems and local Lipschitz arguments in proving existence (Theorem 3.2) is mathematically sound.

**Weaknesses:**

1. **Lack of practical motivation for SNRDD and AVI conditions.**

   The paper repeatedly states that the theoretical properties of PBE, linear Q-learning, and AVI are “not well understood” (line 45) but does not clarify *why* such understanding is important. It remains unclear what practical benefit arises from identifying the SNRDD or AVI contraction conditions. Do they guide algorithm design, help diagnose convergence failure, or provide insight into stability under function approximation? Without this link, the results feel detached from practical reinforcement learning. For instance, while SNRDD guarantees uniqueness of the PBE solution (Theorem 3.2), the paper never demonstrates how this insight could be used to construct or modify algorithms in practice.

2. **No discussion of necessity or minimality of conditions (4), (6), and (7).**

   The analysis provides only *global sufficient* conditions for convergence, SNRDD in Eq. (4) and the AVI-motivated inequalities in (6)–(7). However, these are imposed uniformly for all $\theta$ in $\mathcal{D}$, which may be stronger than necessary. For example, Lemma 3.6 follows almost directly from the definition in (4), suggesting that the assumptions are conservative rather than tight. The paper never investigates whether local conditions near a fixed point would suffice, nor does it analyze the borderline cases when these inequalities fail. This omission leaves unclear what the true boundary of convergence is and whether the results could be sharpened.

3. **Lack of empirical or verifiable interpretation.**

   While the paper introduces two mathematical conditions (SNRDD and AVI contraction), it does not discuss how practitioners could verify them in realistic reinforcement learning settings. For instance, Eq. (4) requires checking a row-wise dominance property of a matrix involving the unknown transition structure and feature representation, which is something infeasible to compute in practice. The paper provides no numerical examples demonstrating whether these conditions approximately hold, nor any heuristic indicators that could help identify when a system might violate them.

4. **Unclear takeaway from ε-greedy pathologies (Section 6).**

   Section 6 presents examples where $\epsilon$-greedy policies lead to multiple or nonexistent PBE solutions as $\epsilon$ varies, illustrating discontinuities in the induced operator. While these examples are interesting, the paper does not extract a clear lesson: should $\epsilon$-greedy be avoided, or can the results motivate a modification (e.g., replacing it with softmax or continuous exploration)? Moreover, the examples are disconnected from the main theory: the paper does not explain whether SNRDD or AVI contraction fails in those cases or whether these examples reveal a fundamental limitation of the proposed conditions.

5. **Weak connection to prior convergence literature and unclear technical novelty.**

   The paper positions itself as deepening the theoretical understanding of PBE, yet many results (e.g., Lemma 3.6 and Proposition 3.13) appear to follow directly from standard definitions or fixed-point arguments. It is unclear what the main technical difficulty is or how it compares to earlier analyses such as Li et al. (2024) (*Operations Research*), Tsitsiklis and Van Roy (1997), or Baird (1995). Those works already explore convergence, bias, and sample complexity of Q-learning with general function approximation. In contrast, this paper’s focus on the *existence* of a fixed point lacks discussion on how such existence results affect convergence guarantees, estimation bias, or sample efficiency. Without this comparison, it is hard to see what new theoretical challenge the paper truly addresses.

**Questions:**

1. Beyond “not well understood,” what practical benefit arises from identifying SNRDD or AVI contraction conditions?

2. Are conditions (4)/(6)/(7) necessary near fixed points, or can weaker conditions suffice?

3. How could practitioners empirically verify these conditions?

4. What actionable takeaway should one draw from ε-greedy pathologies (Section 6)?

5. How do these results extend or differ from prior convergence analyses (e.g., Meyn 2024; Li et al. 2024)? What is the technical difficulty?

---

> ### Author Response · Authors · 2025-11-17
>
> **W1.** *Lack of practical motivation for SNRDD and AVI conditions. The paper repeatedly states that the theoretical properties of PBE, linear Q-learning, and AVI are “not well understood” (line 45) but does not clarify why such understanding is important. It remains unclear what practical benefit arises from identifying the SNRDD or AVI contraction conditions. Do they guide algorithm design, help diagnose convergence failure, or provide insight into stability under function approximation? Without this link, the results feel detached from practical reinforcement learning. For instance, while SNRDD guarantees uniqueness of the PBE solution (Theorem 3.2), the paper never demonstrates how this insight could be used to construct or modify algorithms in practice.*
>
>
> **Q1.** *Beyond “not well understood,” what practical benefit arises from identifying SNRDD or AVI contraction conditions?*
>
> **A1.** We thank the reviewer for the comments on the practical motivation for SNRDD and AVI conditions and importance of the theoretical understanding of PBE, linear Q-learning, and AVI.
>
> First, let us explain how the theoretical understanding can be helpful in practice. Our result gives insight on how regularization can be used in practice. In particular, when designing algorithms, we can add an additional $-\eta\theta$ term to PBE or the update of Q-learning to ensure stability--analogous to $\ell_2$ regularization in supervised learning. As discussed in Lemma 3.6 and Remark 3.7 of the manuscript, with appropriate choice of $\eta$, the SNRDD condition can be easily met. The condition on the coefficient $\eta>3$ is sufficient when the feature values scale at the order of $\frac{1}{\sqrt{p}}$, where $p$ is the feature dimension. **This condition does not require knowledge of MDP.** This provides additional insight on how using $l2$-type regularization should be helpful in RL.
>
> Furthermore, let us illustrate how our theoretical findings can indicate where practical improvements may be pursued. Since AVI can be viewed as a primitive form of deep Q-networks (DQN), the fact that we identify regimes in which Q-learning converges while AVI fails highlights meaningful structural differences between the two. This observation encourages further investigation into Q-learning–type updates, consistent with recent efforts to revisit DQN without target networks—essentially reverting to a form of Q-learning [R1, R2].
>
> Lastly, we emphasize the importance of understanding convergence conditions. As the reviewer noted, the conditions we established can be used to determine whether an algorithm is expected to converge or diverge in the **linear function approximation setting**. Moreover, different convergence conditions exist in the literature, and these conditions neither imply one another nor fully overlap, highlighting the need to characterize their distinctions and relationships.
>
> We have clarified this in contribution part, Remark 3.7 and conclusion part of the revised manuscript.
>
>
> [R1] Gallici, Matteo, et al., “Simplifying Deep Temporal Difference Learning,” ICLR 2025.
>
> [R2] Vasan, Gautham, et al., “Deep policy gradient methods without batch updates, target networks, or replay buffers,” NeurIPS 2024, pp. 845–891.
>
> **W2.** *No discussion of necessity or minimality of conditions (4), (6), and (7). The analysis provides only global sufficient conditions for convergence, SNRDD in Eq. (4) and the AVI-motivated inequalities in (6)–(7). However, these are imposed uniformly for all $\theta$ in $\mathcal{D}$, which may be stronger than necessary. For example, Lemma 3.6 follows almost directly from the definition in (4), suggesting that the assumptions are conservative rather than tight. The paper never investigates whether local conditions near a fixed point would suffice, nor does it analyze the borderline cases when these inequalities fail. This omission leaves unclear what the true boundary of convergence is and whether the results could be sharpened.*
>
> **Q2.** *Are conditions (4)/(6)/(7) necessary near fixed points, or can weaker conditions suffice?*
>
> **A2.** We thank the reviewer for the comments on local conditions. Indeed, if a fixed point exists, local conditions can be easily established around the fixed point by replacing the set of differentiable points $\mathcal{D}$ with the neighbourhood of the corresponding fixed point. In contrast, what we tackle is the existence of globally unique fixed point. Moreover, we note that Lemma 3.6 is important yielding the condition in Remark 3.7 which states that $\eta>3$ is required for the feature value scaling at $\frac{1}{\sqrt{p}}$. **This bound is independent on the choice of $\mathcal{D}$, which shows its generality.** Following the discussion, we have added this to Remark 3.7 and the conclusion prat of the revised manuscript.

---

> > ### Author Response · Authors · 2025-11-17
> >
> > **W3.** *Lack of empirical or verifiable interpretation. While the paper introduces two mathematical conditions (SNRDD and AVI contraction), it does not discuss how practitioners could verify them in realistic reinforcement learning settings. For instance, Eq. (4) requires checking a row-wise dominance property of a matrix involving the unknown transition structure and feature representation, which is something infeasible to compute in practice. The paper provides no numerical examples demonstrating whether these conditions approximately hold, nor any heuristic indicators that could help identify when a system might violate them.*
> >
> > **Q3.** *How could practitioners empirically verify these conditions?*
> >
> > **A3.**  We thank the reviewer for the valuable comments on verifiable conditions on the SNRDD condition. As the reviewer correctly pointed out, there are practical scenarios in which the SNRDD condition may be difficult to verify in practice. To address this issue, we can add an additional $-\eta\theta$ term to PBE or the update of Q-learning--analogous to $\ell_2$ regularization in supervised learning. As discussed in Lemma 3.6 and Remark 3.7 of the manuscript, the condition on the coefficient $\eta>3$ is sufficient when the feature values scale at the order of $\frac{1}{\sqrt{p}}$, where $p$ is the feature dimension. **This condition does not require knowledge of MDP.** This provides a simple method to guarantee the SNRDD condition in practice. We have clarified this in Remark 3.7 of the revised manuscript.
> >
> > **W4.** *Unclear takeaway from $\epsilon$-greedy pathologies (Section 6). Section 6 presents examples where $\epsilon$-greedy policies lead to multiple or nonexistent PBE solutions as varies, illustrating discontinuities in the induced operator. While these examples are interesting, the paper does not extract a clear lesson: should $\epsilon$-greedy be avoided, or can the results motivate a modification (e.g., replacing it with softmax or continuous exploration)? Moreover, the examples are disconnected from the main theory: the paper does not explain whether SNRDD or AVI contraction fails in those cases or whether these examples reveal a fundamental limitation of the proposed conditions.*
> >
> > **Q4.** *What actionable takeaway should one draw from $\epsilon$-greedy pathologies (Section 6)?*
> >
> > **A4.** We thank the reviewer for the constructive comments on Section 6. In Example 14.1 in Section 6, which uses $\epsilon$-greedy behavior policy, we show that the existence of solution to PBE is irrelevant with the SNRDD condition. **Therefore, the examples exactly show the fundamental limitation of the proposed conditions to the $\epsilon$-greedy behavior policy.** Nonetheless, we believe that under SNRDD condition, the boundedness of the iterate can be proved using the differential inclusion framework by [R3]. We have included the discussion in the Section 6 and conclusion part of the revised manuscript.
> >
> > [R3] Gopalan, Aditya, and Gugan Thoppe. "Demystifying Approximate RL with $\epsilon $-greedy Exploration: A Differential Inclusion View."

---

> > > ### Author Response · Authors · 2025-11-17
> > >
> > > **W5.** *Weak connection to prior convergence literature and unclear technical novelty. The paper positions itself as deepening the theoretical understanding of PBE, yet many results (e.g., Lemma 3.6 and Proposition 3.13) appear to follow directly from standard definitions or fixed-point arguments. It is unclear what the main technical difficulty is or how it compares to earlier analyses such as Li et al. (2024) (Operations Research), Tsitsiklis and Van Roy (1997), or Baird (1995). Those works already explore convergence, bias, and sample complexity of Q-learning with general function approximation. In contrast, this paper’s focus on the existence of a fixed point lacks discussion on how such existence results affect convergence guarantees, estimation bias, or sample efficiency. Without this comparison, it is hard to see what new theoretical challenge the paper truly addresses.*
> > >
> > > **Q5.** *How do these results extend or differ from prior convergence analyses (e.g., Meyn 2024; Li et al. 2024)? What is the technical difficulty?*
> > >
> > > **A5.** We thank the reviewer for pointing out the connection with prior convergence literature and technical novelty.
> > >
> > > Let us first explain the novelty of the result of Lemma 3.6 and Proposition 3.13
> > >
> > > **Lemma 3.6:** As answered in the previous question, the novelty of Lemma 3.6 is in yielding the result in Remark 3.7 rather than the proof technique. It states that the condition on the coefficient $\eta>3$ is sufficient when the feature values scale at the order of $\frac{1}{\sqrt{p}}$, where $p$ is the feature dimension. This condition does not require knowledge of MDP. This provides additional insight on how using $l2$-type regularization should be helpful in RL. We have clarified this in the revised manuscript.
> > >
> > > **Proposition 3.13:** While the proof can be standard, our result is the first to establish conditions under which both linear Q-learning and AVI converge—an open question not previously addressed in the literature. Moreover, our divergence examples provide complementary insights and also represent novel findings. As noted in the earlier response, these results align with recent empirical progress showing that target networks can be removed in deep Q network (DQN), and our analysis offers theoretical support for these developments.
> > >
> > >
> > > Now, let us highlight the difference with related works:
> > >
> > > **Meyn 2024:** Our work differs from Meyn (2024) in that their analysis relies on a specific behavior policy, the tamed Gibbs policy. In contrast, we consider a distinct setting motivated by the SNRDD and AVI conditions. Notably, the tamed Gibbs policy **requires knowledge of the underlying model parameters, for example a quantity related to minimum of the stationary distribution, whereas our conditions can be satisfied by simply choosing $\eta>3$ when the feature values scale at the order of $\frac{1}{\sqrt{p}}$**. Moreover, we investigate additional relationship with AVI, which was not discussed in Meyn 2024.
> > >
> > > **Li et al., 2024:** Li et al., 2024 only covers the tabular learning scenario whereas we focus on linear function approximation, which covers a different scenario. Tabular scenario differ from linear function approximation setting as the fixed point always exist and is unique, and convergence of algorithm is always guaranteed.
> > >
> > > **Tsitsiklis and Van Roy (1997):** The work in Tsitsiklis and Van Roy (1997) only covers the scenario of TD-learning where a fixed policy is used for both target and behavior policy. In contrast, we consider the case where target and behavior policy keep changes, which causes difficulty in the analysis of solution to PBE and convergence results.
> > >
> > > **Baird (1995):** Baird (1995) develops a residual gradient method which takes gradient on the mean projected Bellman error. This method relies on two independent samples per iteration, which is the reason why the method is not used in practice. In contrast, the algorithms we study—Q-learning and AVI, the primitive form of DQN—do not rely on such a sampling structure and are widely used in practice.
> > >
> > > We have incorporated the above discussion in the related works part of the revised manuscript.

---

> > > > ### Comment · Reviewer_EKy9 · 2025-11-28
> > > >
> > > > Thank you for the authors for the detailed and thoughtful response. I acknowledge the effort to clarify and motivate the theoretical results.
> > > >
> > > > That said, I still view the contributions—particularly the globally sufficient convergence conditions—as somewhat incremental in scope. While the manuscript provides multiple motivations emphasizing the importance of Lemma 3.6, Proposition 3.13, and Remark 3.7, the practical impact of conditions (4), (6), and (7) remains unclear to me beyond the benefits of regularization.
> > > >
> > > > Additionally, regarding my earlier question (Q2), if conditions (4)/(6)/(7) are truly not necessary for convergence, I would expect stronger support such as explicit counterexample–based evidence demonstrating this claim. In my view, the current response (A2) explains how local conditions could be formulated, but it does not fully address whether weaker conditions are sufficient in cases near a fixed point or at the boundary where the proposed inequalities fail.
> > > >
> > > > Based on the current discussion, I am raising this point’s rating to 4.

---

> ### Author Response · Authors · 2025-11-28
>
> We thank the reviewer for the engagement in the discussion, and the time and effort reviewing our manuscript. We clarify the remaining concerns raised by the reviewer in the following:
>
>
> **Q1-1.** *That said, I still view the contributions—particularly the globally sufficient convergence conditions—as somewhat incremental in scope. While the manuscript provides multiple motivations emphasizing the importance of Lemma 3.6, Proposition 3.13, and Remark 3.7, the practical impact of conditions (4), (6), and (7) remains unclear to me beyond the benefits of regularization.*
>
> **A1-1.** We thank the reviewer for providing the chance to make clarification on our contribution. As the reviewer mentioned, the global condition might not easy to hold, and this highlights the importance of using regularization. We exactly derive a condition on the regularization coefficient **which is not dependent on the factors of MDP** in Remark 3.7. The conditions are exactly helpful in understanding the effect of regularization. **Understanding the role of regularization has been an important topic in the deep reinforcement learning community [R1, R3]**. Moving beyond this, we also emphasize our following contributions: we investigate the convergence relationship of linear Q-learning and AVI—**an open question not previously addressed in the literature. This is an important question as recent works try to remove target network from DQN [R1, R2]**. Moreover, our divergence examples provide complementary insights and also represent novel findings between the behavior of the algorithms. Lastly, we show the hardness of extending the analysis to $\epsilon$-behavior policy by presenting an example where the SNRDD condition holds, yet no solution—or multiple solutions—may exist. We have clarified this in the contribution part of our revised manuscript.
>
>
> [R1] Gallici, Matteo, et al., “Simplifying Deep Temporal Difference Learning,” ICLR 2025.
>
> [R2] Vasan, Gautham, et al., “Deep policy gradient methods without batch updates, target networks, or replay buffers,” NeurIPS 2024, pp. 845–891.
>
> [R3] Farebrother, J., Machado, M. C., and Bowling, M. (2018). Generalization and regularization in dqn.
> arXiv preprint arXiv:1810.00123.
>
> **Q2-2.** *Additionally, regarding my earlier question (Q2), if conditions (4)/(6)/(7) are truly not necessary for convergence, I would expect stronger support such as explicit counterexample–based evidence demonstrating this claim. In my view, the current response (A2) explains how local conditions could be formulated, but it does not fully address whether weaker conditions are sufficient in cases near a fixed point or at the boundary where the proposed inequalities fail.*
>
> **A2-2.** We thank the reviewer for the comments on the local conditions. In Example 13.3 of the manuscript, we have exactly provided the case where only SNRDD condition and (6) is met locally. We have illustrated the local convergence in Figure 1 (a) of the manuscript. Moreover, we have newly provided additional details that how SNRDD condition and (6) holds only locally in Figure 5 in the Appendix of the revised manuscript.
>
> Meanwhile, once a fixed point exists, the local behavior of the algorithm is governed by its linearization. In this regime, the analysis reduces to studying the associated linear system, where the SNRDD condition—and consequently conditions (6) or (7) can be replaced with the relevant matrices being Hurwitz and Schur, respectively.
>
>
> We again thank the reviewer for the time, effort, and thoughtful engagement in reviewing our manuscript and participating in the discussion.

---

### Official Review · Reviewer_CbLt · 2025-11-01

**Soundness:** 2
**Presentation:** 2
**Contribution:** 2
**Rating:** 4
**Confidence:** 4

**Summary:**

The paper investigates the projected Bellman equation (PBE) and examines its theoretical connections to linear Q-learning and approximate value iteration (AVI). It establishes sufficient conditions for the existence and uniqueness of PBE solutions—most notably through a strictly negatively row-dominant diagonal (SNRDD) property and an additional condition motivated by AVI. The authors analyze convergence of both AVI and linear Q-learning under these assumptions using contraction mappings and fixed-point arguments. They also present illustrative examples, including cases where convergence fails or yields suboptimal solutions under ε-greedy policies.

**Strengths:**

The paper tackles a fundamental and underexplored theoretical topic: when and why the projected Bellman equation admits a unique solution and how that affects linear Q-learning and AVI. The formal analysis seems to be technically sound, and the examples highlighting pathological convergence behaviors are informative. The paper is well-written and easy to follow in general.

**Weaknesses:**

The existence and uniqueness of PBE solutions under diagonal dominance (SNRDD) are not particularly surprising—such conditions are standard in numerical linear algebra and reinforcement learning theory. Consequently, Section 3, while technically correct, feels incremental and could be condensed substantially. Additionally, some of the technical assumptions, such as requiring relatively large regularization constants for certain guarantees, seem restrictive and analytically convenient rather than broadly insightful. These conditions limit the perceived generality and practical relevance of the results.

Further, given that MDP and RL is a very classical and well-sturdied field, many related works seems missing, to name a few: Convergence of Q-learning with Linear Function Approximation (Melo, Meyn & Ribeiro, 2008), and An Analysis of Linear Models and Value-Function Approximation (Parr et al., 2008). A lack of comparison and acknowledgement of previous works makes it difficult to evaluate how this paper advances beyond well-established results. It remains unclear which aspects are novel relative to known stochastic approximation and ODE-based analyses. The manuscript would benefit from clearer statements of what is newly proven here versus what is already established in the literature.

**Questions:**

See the Weaknesses section

---

> ### Author Response · Authors · 2025-11-17
>
> **W1.** *The existence and uniqueness of PBE solutions under diagonal dominance (SNRDD) are not particularly surprising—such conditions are standard in numerical linear algebra and reinforcement learning theory. Consequently, Section 3, while technically correct, feels incremental and could be condensed substantially. Additionally, some of the technical assumptions, such as requiring relatively large regularization constants for certain guarantees, seem restrictive and analytically convenient rather than broadly insightful. These conditions limit the perceived generality and practical relevance of the results.*
>
> **A1.**  We thank the reviewer for the insightful comments on Section 3 and the regularization coefficient. We would like to emphasize several points regarding the importance of Section 3: Even though the standard technique from fixed point theory is adopted to investigate the solution properties of PBE in (1), the detailed conditions under which these tools apply to the PBE setting have not been thoroughly investigated. Our study contributes to this gap by identifying concrete conditions under which the policy satisfies local Lipschitz continuity or detailed numerical conditions that a feature matrix should satisfy. We study that these conditions are particularly useful when analyzing the regularized PBE, where an additional $\eta\theta$ term—analogous to $\ell_2$ regularization in supervised learning—is added. As we discuss in Lemma 3.6 and Remark 3.7 of the manuscript, with appropriate choice of $\eta$, the SNRDD condition can be easily met. We show that the coefficient $\eta$ only need to meet the condition $\eta>3$ when $||\phi(s,a)||_{\infty}\leq \frac{1}{\sqrt{p}}$, where $p$ is the feature dimension. **This result does not require any knowledge of the MDP.** This provides additional insight on how using $l2$-type regularization should be helpful in RL.
>
> More importantly, we provide necessary and sufficient conditions for the convergence of both AVI and Q-learning based on the SNRDD condition, which has not been considered in the literature. Based on this analysis, we provided an example where only one of the two algorithms converges. This contributes to a deeper understanding of two key algorithmic approaches to solving the PBE. Specifically, AVI can be seen as a simplified form of DQN, where the target network is updated with a sufficiently large interval. Therefore, our results offer additional insights into scenarios where Q-learning may converge while DQN fails, or vice versa. These findings also indicate where practical improvements may be pursued. Since AVI can be viewed as a primitive form of deep Q-networks, the fact that we identify regimes in which Q-learning converges while AVI fails highlights meaningful structural differences between the two. This observation encourages further investigation into Q-learning–type updates, consistent with recent efforts to revisit DQN without target networks—essentially reverting to a form of Q-learning [R1, R2]. We have included the discussion in the contribution part, Remark 3.7 and the conclusion part of our revised manuscript.
>
> [R1] Gallici, Matteo, et al., “Simplifying Deep Temporal Difference Learning,” ICLR 2025.
>
> [R2] Vasan, Gautham, et al., “Deep policy gradient methods without batch updates, target networks, or replay buffers,” NeurIPS 2024, pp. 845–891.

---

> > ### Author Response · Authors · 2025-11-17
> >
> > **W2.** *Further, given that MDP and RL is a very classical and well-sturdied field, many related works seems missing, to name a few: Convergence of Q-learning with Linear Function Approximation (Melo, Meyn \& Ribeiro, 2008), and An Analysis of Linear Models and Value-Function Approximation (Parr et al., 2008). A lack of comparison and acknowledgement of previous works makes it difficult to evaluate how this paper advances beyond well-established results. It remains unclear which aspects are novel relative to known stochastic approximation and ODE-based analyses. The manuscript would benefit from clearer statements of what is newly proven here versus what is already established in the literature.*
> >
> >
> > **A2.** We thank the reviewer for the comments on related work. In the following, we clarify the comparison with the related works:
> >
> > [R3] investigates convergence of Q-learning and imposes a condition on the feature function, requiring $||\phi(s,a)||_{\infty} \leq 1$; however, as noted in its errata, **the proof under this condition is incomplete**. In a follow-up work, [R4] adopts a stronger assumption, $||\phi(s,a)||\_{\infty} = 1$, which is stronger than the condition used in our analysis, i.e., if $||\phi(s,a)||\_{\infty} = 1$, then SNRDD condition is satisfied. We have added this in the Remark 4.6 in the revised manuscript.
> >
> > [R5] studies learning a feature matrix in a **model-based manner**, i.e., requires a $\mathbb{R}^{|\mathcal{S}|\times |\mathcal{S}|}$ space memory. Moreover, **the main focus of [R5] is on the analysis under the policy evaluation scheme** rather than considering policy improvement setting. This contrasts with our work, which centers on the PBE, Q-learning, and AVI. We have added this in the Remark 4.6 and the related works section in the revised manuscript, respectively.
> >
> >
> > Moreover, we do not claim novelty on the new method on ODE based analysis; but rather, our contribution lies in offering a new perspective on analyzing linear Q-learning through the lens of contraction theory. In details, we identify the one-sided Lipschitz condition of linear Q-learning and show a condition on regularization coefficient $\eta$ that does not depend on the knowledge of model parameters. Furthermore, our contribution lies in investigating the convergence relationship with AVI, and showing the hardness of extending the analysis to $\epsilon$-behavior policy by presenting an example where the SNRDD condition holds, yet no solution—or multiple solutions—may exist. We have clarified this in the contribution part of our revised manuscript.
> >
> >
> > [R3] Convergence of Q-learning with Linear Function Approximation, 2008, Melo, Meyn \& Ribeiro
> >
> > [R4] An Analysis of Linear Models, Linear Value-Function Approximation, and Feature Selection for Reinforcement Learning, 2008, Parr et al.
> >
> > [R5] Melo, Francisco S., and M. Isabel Ribeiro. "Q-learning with linear function approximation." International Conference on Computational Learning Theory. Berlin, Heidelberg: Springer Berlin Heidelberg, 2007.

---

### Official Review · Reviewer_DW32 · 2025-11-01

**Soundness:** 3
**Presentation:** 3
**Contribution:** 3
**Rating:** 6
**Confidence:** 3

**Summary:**

This paper studies the projected Bellman equation (PBE) under linear function approximation and analyzes two solvers—linear Q-learning and approximate value iteration (AVI). It introduces a sufficient condition based on “strictly negatively row-dominating diagonals” (SNRDD) to guarantee existence/uniqueness of PBE solutions, connects that condition to AVI-style contraction conditions, and gives convergence proofs for several Q-learning variants via contraction/ODE arguments. It also presents examples showing (i) divergence/convergence mismatches between AVI and linear Q-learning and (ii) pathological behaviors under $\epsilon$-greedy policies (solution multiplicity, non-existence, and emergence of optimal yet unattainable solutions).

**Strengths:**

Strengths

Clear new sufficient condition (SNRDD) for PBE solvability that applies across tabular, linear, and regularized settings, and accommodates off-policy cases and (locally) Lipschitz policies. This is positioned as broader/different from prior on-policy or tamed-Gibbs results.

Unifying convergence view for Q-learning variants (tabular asynchronous, linear, and regularized) via contraction theory/ODE analysis—reducing auxiliary assumptions (e.g., feature positivity/orthogonality; no target network/projection required for their regularized variant).
Explicit comparison to AVI-motivated conditions and a formal relationship result (Proposition 3.13) clarifying when the SNRDD-based criterion aligns with an AVI-style norm bound.

Insightful counter-examples: concrete constructions where AVI converges but linear Q-learning oscillates and vice-versa (Figure 2), and where Q-learning converges to a unique but sub-optimal fixed point. These help map the frontier between the two methods.

**Weaknesses:**

Scope restricted to linear function approximation: While the linear regime is foundational, many modern RL systems are nonlinear; the paper stops short of indicating which parts of the analysis might transfer (even qualitatively) to nonlinear function classes. (Authors briefly note this as future work.)

**Questions:**

Nonlinear approximation outlook: Which pieces of your ODE/contraction proof strategy seem most likely to carry over to nonlinear function classes (e.g., local SNRDD-like Jacobian conditions), and what are the main obstacles?

---

> ### Author Response · Authors · 2025-11-17
>
> **W1 and Q1.** *(W1) Scope restricted to linear function approximation: While the linear regime is foundational, many modern RL systems are nonlinear; the paper stops short of indicating which parts of the analysis might transfer (even qualitatively) to nonlinear function classes. (Authors briefly note this as future work.)  (Q1) Nonlinear approximation outlook: Which pieces of your ODE/contraction proof strategy seem most likely to carry over to nonlinear function classes (e.g., local SNRDD-like Jacobian conditions), and what are the main obstacles?*
>
> **A1.**   We thank the reviewer for the insightful comment regarding the extension to non-linear function approximation. We believe that our analysis can be naturally extended to this setting, following the approach in [R2], which investigates the convergence of Q-learning with neural networks. Specifically, [R2] considers a projection onto a linear subspace of the form ${Q(\theta_0) + \nabla Q(\theta_0)^{\top}(\theta - \theta_0)}$, where $Q(\cdot)$ is the fuction approximator and $\theta_0$ denotes the initialization point. Then, the analysis in [R1] relies on Melo's condition from [R2], originally used to prove convergence under linear function approximation. By replacing Melo's condition with our SNRDD assumption, we believe that our theoretical results can similarly be extended to the non-linear setting.
>
> Moreover, our contraction theory based analysis explicitly highlights the challenges in extending these results to the nonlinear function approximation setting. For simplicity, let us fix the target policy $\pi$ and consider the policy evaluation scheme. Let $F_{\mathrm{pbe}}(x)=\nabla f(x)^{\top}D(R+\gamma P\Pi^{\pi} f(x))$ and $f:\mathbb{R}^p\to\mathbb{R}^{|\mathcal{S}|}$ approximates the value function, $x\in\mathbb{R}^{p}$ is the learnable parameter, and $f(x;s)$ denotes the $s$-th element of $f(x)$. The contraction theory [R3] states that, if the Jacobian $\frac{\partial F_{\mathrm{pbe}}}{\partial x}$ is Hurwitz, then every two trajectories of the ODE $\dot{x}_t=F\_{\text{pbe}}(x_t)$ are converging. If we calculate the Jacobian $\frac{\partial F\_{pbe}}{\partial x}$, we get
>
> $$\begin{aligned}
>      &\frac{\partial }{\partial x}  \left( \nabla f(x)^{\top}DR+  \gamma  \nabla f(x)^{\top}DP^{\pi}f(x) - \nabla f(x)^{\top}D\nabla f(x) \right) \\\\
>      =&  \underbrace{\sum_{s}d(s) \nabla^2 f(x;s) \left( \sum\_{s^{\prime}}P^{\pi}(s^{\prime}\mid s) \left(r(s,s^{\prime})+ \gamma f(x;s^{\prime}) \right)- f(x;s)\right) }\_{I_1}\\
>      &+ \underbrace{\sum\_s d(s) \left(\gamma  \sum\_{s^{\prime}}P^{\pi}(s^{\prime}\mid s) \nabla f(x;s)  \nabla f(x;s^{\prime})^{\top}  - \nabla f(x;s) \nabla f(x;s)^{\top} \right)}\_{I_2}
> \end{aligned}$$
>
> The term $I_1$ appears due to using non-linear function approximation whereas $I_2$ is the term that also appears in the linear function approximation setting. Consequently, while the term $I_2$ can be controlled by the SNRDD approach but it is not clear how to control the $I_1$ term, which is the unique challenge in the analysis.
>
> We have included this discussion in the conclusion part and Section 16 in Appendix of our revised manuscript.
>
> [R1] Xu, Pan, and Quanquan Gu. "A finite-time analysis of Q-learning with neural network function approximation." International conference on machine learning. PMLR, 2020.
>
> [R2] Melo, F. S., Meyn, S. P., and Ribeiro, M. I. (2008). An analysis of reinforcement learning with function approximation. In Proceedings of the 25th international conference on Machine learning,
> pages 664–671.
>
> [R3] Lohmiller, Winfried, and Jean-Jacques E. Slotine. "On contraction analysis for non-linear systems." Automatica 34.6 (1998): 683-696.

---

### Official Review · Reviewer_qNwA · 2025-11-01

**Soundness:** 3
**Presentation:** 3
**Contribution:** 2
**Rating:** 6
**Confidence:** 3

**Summary:**

This paper investigates the existence and uniqueness of solutions for two projected Bellman-equation-based algorithms: linear Q-learning and AVI. The main conditions used in the paper are SNRDD, a matrix condition for determining the stability of dynamical systems, for linear Q-learning, and a matrix norm condition for AVI. The paper also provides a convergence analysis using ODE-based stochastic approximation (Borkar and Meyn, 2000) for the linear Q-learning algorithm, leveraging the SNRDD condition.

**Strengths:**

- The authors present a unifying tool (SNRDD) for analyzing the convergence of tabular, linear, and regularized linear Q-learning.
- An interesting contrast is provided, with conditions showing when AVI converges while linear Q-learning does not, and vice versa, as well as a condition under which both converge (Proposition 3.13).
- An extensive appendix with theoretical rigor is provided that helps guide readers through the definitions and results.

**Weaknesses:**

- My main concern is the novelty of the results provided. The paper centers on the SNRDD condition, which was already used by (Lim and Lee, 2024) for a similar purpose in regularized linear Q-learning. Although (Lim and Lee, 2024) required two additional conditions (e.g., orthogonal and non-negative features), as the authors indicated in Remark 4.5, I believe this is a bit incremental.
- The fixed behavior policy condition, although assumed in related works, is restrictive in my opinion. The authors mention that a replay buffer can be considered a fixed distribution, but the standard use of a replay buffer involves continual updates with recent experience. Therefore, I am not convinced that a replay buffer argument applies here.

**Questions:**

- Why is the fixed-behavior policy $\beta_\theta$ denoted by $\theta$ in Section 2.2?
- In the e-greedy case, how do the existence of an optimal solution and Q-learning’s ability to converge to it relate to the SNRDD condition?
- What are the implications of the conditions for the existence and uniqueness of the PBE solution (SNRDD and the AVI condition)? For example, can we make design choices in linear Q-learning based on the SNRDD condition to ensure convergence?
- (minor) In Definition 3.1 (SNRDD), is there a missing absolute value on the entries $A_{ij}$?

---

> ### Author Response · Authors · 2025-11-17
>
> **W1.**  *My main concern is the novelty of the results provided. The paper centers on the SNRDD condition, which was already used by (Lim and Lee, 2024) for a similar purpose in regularized linear Q-learning. Although (Lim and Lee, 2024) required two additional conditions (e.g., orthogonal and non-negative features), as the authors indicated in Remark 4.5, I believe this is a bit incremental.*
>
>
> **A1.** We thank the reviewer for the constructive comments regarding the comparison with Lim and Lee (2024). Beyond relaxing several assumptions, our work takes a distinct approach by employing the one-sided Lipschitz condition within the framework of contraction theory, whereas Lim and Lee (2024) is based on a switched-system analysis. In addition, we provide a unifying viewpoint through the AVI condition derived from our results, and we present new examples illustrating cases where only one of the two algorithms—Q-learning or AVI—converges. This relationship with AVI was not explored in Lim and Lee (2024). Finally, we present examples demonstrating that the arguments fail to hold under the $\epsilon$-greedy behavior policy showing the hardness of analysis under such scenario, which was not discussed in Lim and Lee (2024). We have modified Remark 4.5 and the contribution part to reflect these points in the revised manuscript.
>
> **W2.** *The fixed behavior policy condition, although assumed in related works, is restrictive in my opinion. The authors mention that a replay buffer can be considered a fixed distribution, but the standard use of a replay buffer involves continual updates with recent experience. Therefore, I am not convinced that a replay buffer argument applies here.*
>
>
> **A2.** We thank the reviewer for the insightful comment regarding the assumptions on the behavior policy and the use of a replay buffer. In our analysis, the replay buffer is introduced to approximate i.i.d. sampling from a fixed distribution, which is a standard assumption in the Q-learning literature. Under a fixed behavior policy, a sufficiently large replay buffer yields samples approximately following the corresponding fixed distribution. Furthermore, the i.i.d. assumption can be relaxed: using the ODE-based approach, the convergence analysis naturally extends to the Markovian observation model generated by a single trajectory under a fixed behavior policy [R1]. We have added this clarification on page seven of the revised manuscript.
>
> [R1] Liu, Shuze Daniel, Shuhang Chen, and Shangtong Zhang. "The ODE method for stochastic approximation and reinforcement learning with markovian noise." Journal of Machine Learning Research 26.24 (2025): 1-76.
>
>
> **Q1.** *Why is the fixed-behavior policy denoted by $\beta_{\theta}$ in Section 2.2?*
>
> **A1.** We thank the reviewer for the comment on the notation. We used it for the consistency of the notation. We have changed it from $\beta_{\theta}$ to $\beta$ for the fixed behavior policy in the revised manuscript.
>
> **Q2.** *In the e-greedy case, how do the existence of an optimal solution and Q-learning’s ability to converge to it relate to the SNRDD condition?*
>
> **A2.** We thank the reviewer for the comment on the case of using $\epsilon$-greedy behavior policy. In Example 14.1 which uses $\epsilon$-greedy behavior policy, we show that the existence of solution to PBE is irrelevant with the SNRDD condition. This example shows hardness of extending the analysis to $\epsilon$-greedy behavior policy. Therefore, it is difficult to prove convergence of Q-learning to a single point under $\epsilon$-greedy policy. Nonetheless, we believe that under SNRDD condition, the boundedness of the iterate can be proved using the differential inclusion framework by [R2]. We have included the discussion in Section 6 and the conclusion part of the revised manuscript.
>
> [R2] Gopalan, Aditya, and Gugan Thoppe. "Demystifying Approximate RL with $\epsilon $-greedy Exploration: A Differential Inclusion View."

---

> > ### Author Response · Authors · 2025-11-17
> >
> > **Q3.** *What are the implications of the conditions for the existence and uniqueness of the PBE solution (SNRDD and the AVI condition)? For example, can we make design choices in linear Q-learning based on the SNRDD condition to ensure convergence?*
> >
> >
> > **A3.** We thank the reviewer for the comments on design choices based on SNRDD condition. We can provide the following guideline for the choice of the regularization coefficient $\eta$ for linear Q-learning: As discussed in Lemma 3.6 and Remark 3.7 of the manuscript, with appropriate choice of $\eta$, the SNRDD condition can be easily met. The condition on the coefficient is $\eta>3$ when $||\phi(s,a)||_{\infty}\leq \frac{1}{\sqrt{p}} $ where $p$ is the feature dimension. This condition does not require any knowledge of the model of the MDP. This provides additional insight on how using $\ell_2$-type regularization should be helpful in RL. We have clarified this in Remark 3.7 of the revised manuscript.
> >
> > Moreover, our theoretical findings also indicate where practical improvements may be pursued. Since AVI can be viewed as a primitive form of deep Q-networks (DQN), the fact that we identify regimes in which Q-learning converges while AVI fails highlights meaningful structural differences between the two. This observation encourages further investigation into Q-learning–type updates, consistent with recent efforts to revisit DQN without target networks—essentially reverting to a form of Q-learning [R3, R4].
> >
> > We have clarified this in Remark 3.7 and in the conclusion part of the revised manuscript.
> >
> > [R3] Gallici, Matteo, et al., “Simplifying Deep Temporal Difference Learning,” ICLR 2025.
> >
> > [R4] Vasan, Gautham, et al., “Deep policy gradient methods without batch updates, target networks, or replay buffers,” NeurIPS 2024, pp. 845–891.
> >
> >
> > **Q4.** *(minor) In Definition 3.1 (SNRDD), is there a missing absolute value on the entries?*
> >
> > **A4.** We thank the reviewer for pointing out the typo. We have corrected this in the revised manuscript.

---

### Author Response · Authors · 2025-11-17
**General Response**

We thank all the reviewers for their constructive feedback, which has greatly helped us improve the quality of the manuscript. All changes in the revised version are highlighted in ${\color{red}red}$. Below, we summarize the common concerns raised by the reviewers and our corresponding responses:

**Technical Novelty and Our Contribution:** Our contribution lies in three fold: First, offering a new perspective on analyzing linear Q-learning through the lens of contraction theory. In details, we identify the one-sided Lipschitz condition of linear Q-learning and show a condition on regularization coefficient $\eta$ that does not depend on the knowledge of model parameters. Second, we investigate the convergence relationship with AVI—an open question not previously addressed in the literature. Moreover, our divergence examples provide complementary insights and also represent novel findings. Lastly, we show the hardness of extending the analysis to $\epsilon$-behavior policy by presenting an example where the SNRDD condition holds, yet no solution—or multiple solutions—may exist. We have clarified this in the contribution part of our revised manuscript.

**Key Practical Insights:** Let us first illustrate how our theoretical understanding can provide help in algorithm design. In particular, our analysis provides explicit guidance on the practical use of regularization. By adding an additional $-\eta\theta$ term to PBE or the update of Q-learning to ensure stability--analogous to $\ell_2$ regularization in supervised learning--we can guarantee stability. As discussed in Lemma 3.6 and Remark 3.7 of the manuscript, with appropriate choice of $\eta$, the SNRDD condition can be easily met. The condition on the coefficient $\eta>3$ is sufficient when the feature values scale at the order of $\frac{1}{\sqrt{p}}$, where $p$ is the feature dimension. **This condition does not require knowledge of MDP.** This provides additional insight on how using $\ell_2$-type regularization should be helpful in RL. We have clarified this in Remark 3.7 of the revised manuscript.


Now, let us illustrate how our theoretical findings can indicate where practical improvements may be pursued. Since AVI can be viewed as a primitive form of deep Q-networks (DQN), the fact that we identify regimes in which Q-learning converges while AVI fails highlights meaningful structural differences between the two. This observation encourages further investigation into Q-learning–type updates, consistent with recent efforts to revisit DQN without target networks—essentially reverting to a form of Q-learning [R1, R2].

[R1] Gallici, Matteo, et al., “Simplifying Deep Temporal Difference Learning,” ICLR 2025.

[R2] Vasan, Gautham, et al., “Deep policy gradient methods without batch updates, target networks, or replay buffers,” NeurIPS 2024, pp. 845–891.


**Implication of Section 6($\epsilon$-greedy behavior):** In Example 14.1 of Section 6, which uses $\epsilon$-greedy behavior policy, we show that existence of solution to PBE is irrelevant with the SNRDD condition. This example shows hardness of extending the analysis to $\epsilon$-greedy behavior policy. Therefore, it is difficult to prove convergence of Q-learning to a single point under $\epsilon$-greedy policy. Nonetheless, we believe that under SNRDD condition, the boundedness of the iterate can be proved using the differential inclusion framework by [R3]. We have included the discussion in the Section 6 and conclusion part of the revised manuscript.

[R3] Gopalan, Aditya, and Gugan Thoppe. "Demystifying Approximate RL with $\epsilon $-greedy Exploration: A Differential Inclusion View."

---

> ### Author Response · Authors · 2025-12-04
> **Summary of Discussion**
>
> Dear AC and reviewers,
>
> Thank you for your time and effort in reviewing our paper. The reviews have helped us significantly improve the quality of our work. Our paper is currently evaluated as borderline (6/6/4/4), with reviewer EKy9 increasing the score from two to four after reading the rebuttal though the reviewer was unable to edit the review. Below, we provide a concise summary of the discussion:
>
>
>
> The reviewers generally acknowledged that the paper offers meaningful contributions in two key aspects. First, the paper presents a unifying perspective on the convergence behavior of linear Q-learning and approximate value iteration (AVI), establishing necessary and sufficient conditions for the convergence of both algorithms and providing counter-examples in which only one of the two algorithms converges. Second, the paper offers illustrative examples involving the $\epsilon$-greedy policy that demonstrate pathological behaviors, thereby clarifying the limitations of the developed theory and emphasizing that the established conditions do not extend to this class of policies.
>
> Reviewer qNwA questioned the novelty of the SNRDD condition compared to Lim and Lee (2024). We clarified that our analysis adopts the contraction theory framework rather than the switched system analysis of Lim and Lee (2024). The adoption of new framework relaxes the condition on feature matrix and regularization coefficient.
>
>
> Reviewer CbLt raised concerns about the scope and significance of using SNRDD condition to study the existence and uniqueness of the solution to PBE. We clarified that while the tools used to establish existence and uniqueness are standard, they lead to new insights when applied in this setting—specifically, they enable a refined analysis of linear Q-learning with regularization and provide foundational results for understanding the convergence relationship between linear Q-learning and AVI.
>
>
> Reviewer DW32 requested clarification on extending the results to non-linear function approximation; we have expanded on this in both the conclusion and Appendix Section 16, emphasizing that the contraction-theoretic approach precisely highlights the challenges involved in extending the analysis to the non-linear setting.
>
>
> Reviewer EKy9 raised concerns on practicality of the convergence conditions provided in the paper. We have addressed this in our general response that with a regularization condition that is not dependent on parameters of MDP, we can meet the desired the conditions. Furthermore, we emphasized that our analysis identifies regimes in which Q-learning converges while AVI fails, revealing meaningful structural differences between the two algorithms. This insight motivates further investigation into Q-learning–type updates and aligns with recent efforts to revisit DQN without target networks—effectively returning to a form of Q-learning [R1, R2].
>
> After the rebuttal, reviewer EKy9 raised questions about the local condition of our analysis. In response, we provided an illustrative example demonstrating how the algorithm behaves under local conditions (Example 13.3) and a more detailed analysis in Figure 5 of the Appendix.
>
>
> We sincerely thank all the reviewers once again for their thoughtful and constructive feedback, which has greatly enhanced the quality of our work.
>
>
> - References
>
> [R1] Gallici, Matteo, et al., “Simplifying Deep Temporal Difference Learning,” ICLR 2025.
>
> [R2] Vasan, Gautham, et al., “Deep policy gradient methods without batch updates, target networks, or replay buffers,” NeurIPS 2024, pp. 845–891.

---

### Meta-Review · Area_Chair_DoMF · 2026-01-05

**Summary:**

This paper identifies the SNRDD condition under which Projected Bellman Equation has a unique solution, and shows that linear Q-learning can converge to this solution under this assumption. They also show that Approximate Value Iteration (AVI) are fundamentally different than linear Q, identifying specific regimes where one converges while the other fails. The research also provides regularization strategy to ensure SNRDD and thus convergence.

Besides novelty concern raised by the reviewers, I remain unconvinced about the practicality of regularization used to ensure SNRDD.  The requirements that the feature magnitude $||\phi(s,a)||\_\infty \leq \frac{1}{\sqrt{p}}$ and the regularization weight $\eta > 3$ are both, in my opinion, not standard or practical.  Notice that the regularization essentially adds a loss $\eta ||\theta||_2^2$ to the loss function.  The main loss (e.g., squared TD loss) the learner care about is $||y- \Phi\theta||_D^2$.  A quick estimation for the magnitude of the TD loss reveals that $ ||\Phi \theta||_D^2 = \sum\_{s,a} \nu(s,a) (\phi(s,a)^\top \theta)^2 \leq \sup\_{s,a} (\phi(s,a)^\top \theta)^2$  $\leq \sup\_{s,a}||\phi(s,a)||\_\infty^2 ||\theta||\_1^2 $ $\leq \frac{1}{p}||\theta||\_1^2 \leq ||\theta||\_2^2$
where we use the assumption $||\phi(s,a)||\_\infty \leq \frac{1}{\sqrt{p}}$ and the Cauchy-Schwarz inequality.

On the other hand, the regularization loss is $\eta ||\theta||\_2^2 > 3||\theta||\_2^2$.  This means that the regularization loss overwhelms the main loss we care about, and thus the algorithm is mainly focusing on minimizing $\eta ||\theta||\_2^2$ than the TD error, and the convergence point could be very far from the optimal $\theta^\star$.  To make the theorem more practical, I think the algorithm has to allow diminishing $\eta$.

I hope that the authors could clarify this scaling in the revised version, and convince that the parameter choice is reasonable.

In the unregularized regime, it would be better to provide concrete practical examples where the conditions hold.

**Reviewer Concerns:**

- Novelty (review qNwA and CbLt) --- addressed: using diagonal dominance to prove existence and uniqueness is a standard tool in numerical linear algebra and RL, and has also been studied in a prior work of Lim and Lee (2024) (though with other assumptions), making the findings feel incremental.  The authors emphasized that the detailed conditions under which these tools apply to the PBE setting have not been thoroughly investigated.

- Applicability of the assumptions (reviewer CbLt) --- partially addressed:  Some of the technical assumptions, such as requiring relatively large regularization constants for certain guarantees, seem restrictive and analytically convenient rather than broadly insightful.  The authors argued that these assumptions can be easily met.

- Lack of practical motivation for SNRDD and AVI conditions (reviewer EKy9) --- partially addressed:  The authors have emphasized that these conditions, while established under linear setting, have implications in, e.g., DQN.  Also, they emphasized their insight on how regularization is helpful in practice.  However, it remains unclear how these conditions relate to scenarios in practice --- are they (without regularization) easily met in real problems?

**Reviewer Scores:**

The original score is 6642.  The score-2 reviewer joined the discussion and raised the score to 4.  Other reviewers remained unengaged so  the scores remained the same.  I think they might keep the same scores even with more discussions.

---

### Decision · Program_Chairs · 2026-01-26

Reject